# NERVE: Noise-Variability-Robust EEG Foundation Model with Electrode-Brain Interactions

## Abstract

Electroencephalography (EEG) is an indispensable modality for measuring and recording brain electrical activity, with broad applications in brain–computer interfaces (BCI) and healthcare. While early EEG models predominantly adopted supervised learning methods due to the scarcity of large-scale datasets and the heterogeneity across tasks and datasets, the recent success of large foundation models has driven increasing efforts to build EEG foundation models. However, most existing studies focus on handling signals with varying formats while overlooking inherent characteristics of EEG signals during acquisition, including low signal-to-noise ratios (SNR), high variability across samples, and spatial dependencies arising from electrode placement within the acquisition system. To address these challenges, we propose NERVE, a novel noise-variability-robust EEG foundation model with electrode-brain interactions. Specifically, pretraining of NERVE begins with learning a noise-robust neural tokenizer that encodes EEG patches into discrete neural tokens. The tokenizer is trained through denoising temporal–spectral prediction to reconstruct temporal and frequency information of the original signal from noise-augmented inputs. NERVE is further pretrained to predict the neural codes of masked EEG patches, integrated with a variability-robust objective that promotes uniform EEG representations. To incorporate spatial structure in EEG, we propose an electrode-position-aware transformer as the backbone for both the tokenizer and the foundation model. It enables the model to capture spatial dependencies among electrodes and brain regions via attention mechanisms. NERVE demonstrates competitive performance across diverse BCI tasks and improved robustness to noise and variability compared to existing EEG foundation models. Our code is available at https://github.com/NERVE-2026/NERVE.

## 1 Introduction

Electroencephalography (EEG) is a method of recording the electrical activity of the brain, which allows evaluating its dynamic functional state (Müller-Putz, 2020). EEG is typically collected in a non-invasive manner, with electrodes placed on the scalp according to the international 10-20 system, providing signals with high temporal resolution. With its non-invasive and portable nature, EEG has become indispensable in brain-computer interface (BCI) and healthcare domains, such as seizure detection (Tzimourta et al., 2019), emotion recognition (Chen et al., 2018), sleep stage detection (Memar & Faradji, 2017), and motor imagery classification (Zhang et al., 2021).

Early studies on BCI tasks relied primarily on machine learning approaches (Edla et al., 2018; Chen et al., 2016). With the rapid advancement of deep learning, numerous neural architectures have been proposed to encode EEG features for downstream BCI tasks, ranging from CNNs (Zhou et al., 2018) and RNNs (Roy et al., 2019) to Transformers (Wan et al., 2023) and GNNs (Zhang et al., 2025a). However, most approaches rely on supervised learning tailored to specific datasets, which limits their generalizability. In addition, collecting and annotating EEG signals is both costly and labor-intensive, making it difficult to acquire large-scale datasets for the task. Moreover, EEG signals exhibit significant heterogeneity, such as mismatched channels and time lengths, arising from

| Noise type | Balanced Accuracy | |
|---|---|---|
| | CBraMod | NERVE |
| - | 0.9189 | **0.9906** |
| EMG | 0.8668 (-5.7%) | **0.9905 (-0.0%)** |
| EOG | 0.9075 **(-0.0%)** | **0.9887** (-0.2%) |
| Environment | 0.9075 (-1.2%) | **0.9906 (-0.0%)** |
| Gaussian | 0.6057 (-34.1%) | **0.8330 (-15.9%)** |

Table 1: Accuracy degradation of EEG foundation models on High-Gamma with noise-augmented inputs.

(a) Full attention (BIOT, LaBraM)   (b) Spatial attention (CBraMod)   (c) EPA attention (NERVE)   Electrode locations

Figure 1: Different spatial dependency modeling strategies.

differences in electrode montages and sampling protocols across acquisition systems, which poses major challenges for developing a universal EEG model.

Recently, inspired by the success of foundation model on natural language processing (NLP) and computer vision (CV), several studies (Yang et al., 2023; Jiang et al., 2024; Wang et al., 2025b) have explored foundation models for EEG. These models are typically pre-trained on large-scale EEG datasets using self-supervised learning and subsequently fine-tuned on downstream tasks. While existing EEG foundation models primarily address variability in channel configurations and signal lengths, they often overlook acquisition-related characteristics that critically affect signal fidelity and representational quality. These factors are particularly important because EEG signals are inherently collected through sensor hardware, and include sensor- and subject-induced noise, variability from recording environments, and differences in hardware composition. We summarize these acquisition-related characteristics as follows:

**1) Low signal-to-noise ratio.** EEG signals inherently suffer from a low signal-to-noise ratio (SNR), and the complexity of brain activity further complicates representation learning under self-supervised paradigms. BIOT (Yang et al., 2023) and CBraMod (Wang et al., 2025b) are pre-trained through masked patch reconstruction, but the low SNR of EEG makes it difficult to reconstruct the original signals accurately. LaBraM (Jiang et al., 2024) introduces masked EEG modeling with neural codebook prediction, but the quantized embeddings do not necessarily ensure robustness to noise (Xiao et al., 2023). As shown in Table 1, we observed that foundation model performance degrades sharply under noise-injected inputs, highlighting the need for noise-robust architectures to improve generalization.

**2) High variability between EEG samples.** EEG is recorded via scalp-mounted electrodes and exhibits substantial variability across subjects, driven by individual differences in functional and anatomical connectivity, head shape, and mental states. Intra-subject variability further arises from time-varying neural dynamics and external environment. Such inter- and intra-subject variability is difficult to capture, as it is not explicitly observable in EEG signals, ultimately undermining model generalization.

**3) Spatial dependency between electrodes during signal acquisition.** Electrodes are positioned on the scalp following the international 10-20 system, which reflects the relationship between electrode placement and the cortical regions. As a result, distinct phases of consciousness produce identifiable electrical patterns across these regions, and signals from spatially adjacent electrodes often capture correlated neural activity. Several BCI studies have demonstrated the importance of incorporating the spatial structure of electrode placement into modeling (Demir et al., 2021; Shen & Namiki, 2025). However, existing EEG foundation models overlook this unique structural characteristic (see Figure 1). BIOT (Yang et al., 2023), LaBraM (Jiang et al., 2024), and NeuroLM (Jiang et al., 2025b) flatten EEG patches before inputting them to transformers, thereby treating dependencies across all patches equally. CBraMod (Wang et al., 2025b) introduces criss-cross attention to capture channel dependencies in EEG, yet it still disregards the intrinsic spatial relationships defined during signal acquisition.

To address these challenges, we propose **NERVE** (**N**oise, **E**lectrode, **R**obust, **V**ariability, **E**EG), a noise- and variability-robust EEG foundation model with brain–electrode interactions. We first train a noise-robust neural tokenizer to generate a discrete neural codebook. Inspired by the denoising autoencoder (Vincent et al., 2008), the tokenizer is optimized to reconstruct the temporal and spectral information of the original signal from noise-augmented inputs, which we term denois-

ing temporal-spectral prediction. NERVE is then pre-trained to predict the neural codes of masked EEG patches from the visible patches, with a variability-robust objective that enforces alignment and uniformity in the representation space, which is crucial for robustness to inter- and intra-subject variability. To incorporate the spatial structure of EEG channels, we further propose an electrode-position–aware (EPA) transformer as the backbone for both the tokenizer and foundation model. The EPA transformer models dependencies among electrodes and their associated cortical regions through attention scores guided by a position router that represents each cortical region. The main contributions of this work are summarized as follows:

- We propose a noise- and variability-robust EEG foundation model with brain-to-electrode interactions, called **NERVE**. This work presents the first foundation model that systematically tackles acquisition-related characteristics of EEG signals, including noise, variability, and spatial dependencies across electrodes. NERVE is pre-trained on a large-scale EEG corpus comprising 27 datasets with 10,956 hours of diverse recordings.

- To address these challenges, NERVE proposes three technical contributions: a noise-robust neural tokenizer to construct a discrete codebook resilient to noise, variability-robust NERVE pre-training for generalizable EEG embeddings, and an electrode-position-aware (EPA) transformer that captures spatial topologies of electrodes across brain regions.

- We evaluate the performance of NERVE on five downstream BCI tasks using 8 public datasets. Experimental results demonstrate that NERVE achieves competitive performance across all tasks, underscoring its strong generalizability and modeling capacity. Furthermore, NERVE demonstrates improved robustness over existing foundation models to noise from EEG signal acquisition, as well as to intra- and inter-subject variability, suggesting its potential for practical application.

## 2 METHOD

The pre-training framework of NERVE is illustrated in Figure 2. It addresses three acquisition-related characteristics, including low SNR, high variability across EEG samples, and spatial dependencies among electrodes through three components: (i) an electrode-position–aware (EPA) transformer as a backbone encoder, (ii) noise-robust neural tokenizer trained via denoising temporal-spectral prediction, (iii) pre-training of NERVE with variability-robust learning.

Prior to a detailed description of NERVE, we first formulate the EEG signals as the multivariate time series $X \in \mathbb{R}^{C \times T}$, where $C$ denotes the number of channels (electrodes) and $T$ represents the signal length. We define the electrode set for $X$ as $\mathbf{C}_X = \{c_{x_1}, c_{x_2}, ...c_{x_C}\}$, where $\mathbf{C}_X \subseteq \mathbf{C} = \{c_1, c_2, ...c_{|\mathbf{C}|}\}$, and $\mathbf{C}$ denotes the universal set of electrodes defined by the international 10–20 system. Note that $\mathbf{C}_X$ may vary across EEG signals $X$, but all follow the 10-20 system convention.

### 2.1 ELECTRODE-POSITION-AWARE TRANSFORMER

Since the electrode-position-aware (EPA) transformer serves as the backbone for both the neural tokenizer and NERVE model, we introduce it first, followed by the noise-robust neural tokenizer and variability-robust NERVE pre-training. We propose the EPA transformer to capture the spatial dependencies of the electrodes and brain regions. To process EEG signals of varying channel numbers and temporal lengths, the EPA transformer first divides each EEG sample into non-overlapping patches with fixed length: $\boldsymbol{x} = \{x_{ij} \in \mathbb{R}^P | i = 1, ..., C, j = 1, ..., N\}$, where $P$ is the patch length and $N = \lfloor \frac{T}{P} \rfloor$ is the total number of patches in each channel.

**Patch Encoding.** Our patch encoder employs several temporal convolution blocks designed to extract local temporal features from EEG signals acquired at high sampling frequencies. The temporal convolution blocks consists of an 1D convolution, a group normalization, and a GELU activation function. Each EEG patch $x_{i,j}$ is fed to the patch encoder to obtain the patch embedding $e_{i,j} \in \mathbb{R}^d$, where $d$ is the embedding dimension.

To encode temporal and spatial information, we add time and channel embeddings to each patch embedding. The time embeddings are defined as $\{e_1^t, ..., e_{t_{\max}}^t\}$, while the channel embeddings $\{e_1^c, ..., e_{|\mathbf{C}|}^c\}$ correspond to the electrodes in the 10–20 system. Both are $d$-dimensional learnable

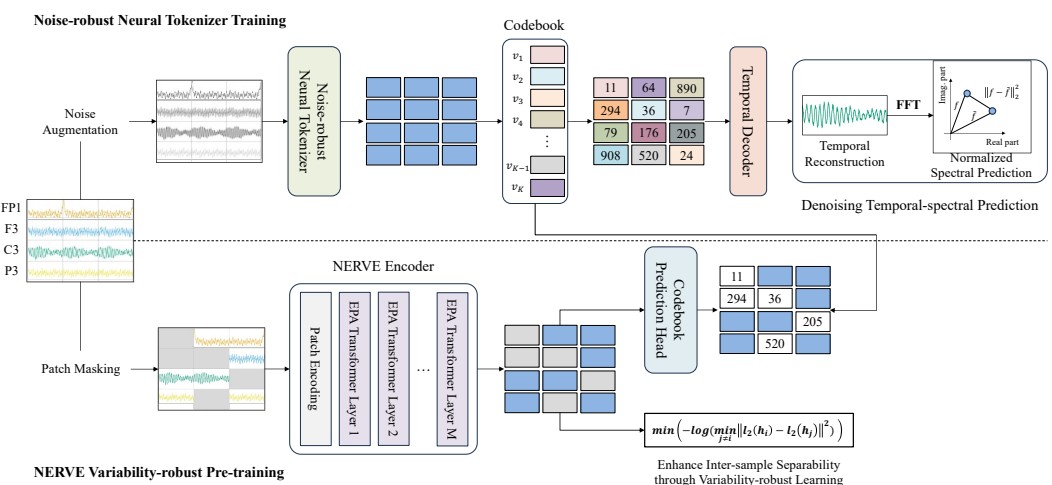

Figure 2: Pre-training architecture of NERVE

vectors. $t_{max}$ is the hyperparameter for maximum number of patches. For $i$'th channel $c_i$, the corresponding channel embedding is selected from the channel embedding set. Then, the final patch embedding of $x_{i,j}$ is formulated as $h_{i,j} = e_{i,j} + e_j^t + e_{c_i}^c$, where $h_{i,j}$ represents the patch embedding. We denote the set of patch embeddings as $H = \{h_{i,j} | i = 1, ..., C, j = 1, ..., N\}$.

**Electrode-position-aware Transformer.** An EPA transformer layer consists of two sequential blocks: a temporal block with layer normalization, temporal attention, residual addition, and a feed-forward network, and a channel block with layer normalization, EPA attention, residual addition, and a feed-forward network. In each block, we adopt a pre-norm strategy, which stabilizes and improves the efficiency of transformer training. We denote the normalized patch embeddings after layer normalization in each block as $\tilde{H} \in \mathbb{R}^{C \times N \times d}$.

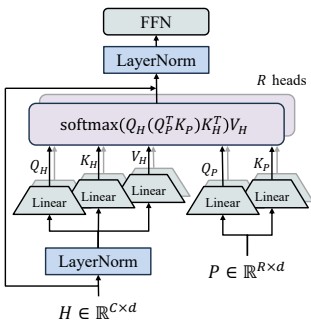

Figure 3: EPA attention

*Temporal Attention.* Temporal attention is used to capture temporal dependencies between EEG patches within each channel. We split $\tilde{H}$ into each channel $\tilde{H} = \{\tilde{H}_{1,:}, ..., \tilde{H}_{C,:}\}$, where $\tilde{H}_{i,:} \in \mathbb{R}^{N \times d}$ is the embeddings of channel $i$ and apply the dot-product attention mechanism:

$$\text{T-Attention}(Q, K, V) = softmax(\frac{QK^T}{\sqrt{d_{head}}})V \tag{1}$$

$$F_i = \text{T-Attention}(\tilde{H}_{i,:}W^Q, \tilde{H}_{i,:}W^K, \tilde{H}_{i,:}W^V) \tag{2}$$

where where $W^Q, W^K, W^V \in \mathbb{R}^{d \times d_{head}}$ are the projection matrices of queries, keys, and values for the each head respectively and $d_{head}$ is the dimension of one head in the multi-head attention.

*Electrode-position-aware Attention.* EPA attention is proposed the model the spatial dependencies between EEG channels based on their positions on the head during signal acquisition. We first split $\tilde{H}$ along the temporal axis as $\tilde{H} = \{\tilde{H}_{:,1}, ..., \tilde{H}_{:,N}\}$, where $\tilde{H}_{:,j} \in \mathbb{R}^{C \times d}$ represents the spatial segments at position $j$. Then, we define a position router $P \in \mathbb{R}^{R \times N \times d}$, a learnable parameter that represents groups of electrodes organized by cortical regions. $R$ is a hyperparameter specifying the number of brain regions, aligned with topographical groupings in EEG research (e.g., frontal, temporal, parietal, occipital). Similarly, the position router is divided along the temporal axis as $P = \{P_{:,1}, ..., P_{:,N}\}$, where $P_{:,j} \in \mathbb{R}^{R \times d}$. Then, EPA attention calculates attention score through the relationship between the electrodes and brain regions, represented by the position router:

$$\text{EPA-Attention}(Q_H, K_H, V_H, Q_P, K_P) = softmax(\frac{Q_H(Q_P^T K_P)K_H^T}{\sqrt{d_{head}}})V_H \tag{3}$$

where $Q_H = \tilde{H}_{:,j} W_H^Q, K_H = \tilde{H}_{:,j} W_H^K, V_H = \tilde{H}_{:,j} W_H^V$ denote the query, key, and value representations of the spatial segments and $Q_P = P_{:,j} W_P^Q, K_P = P_{:,j} W_P^K$ denote the query and key representations of the position router. EPA attention yields attention scores that can be decomposed into two components, electrode-to-brain and brain-to-electrode interactions:

$$Q_H(Q_P^T K_P) K_H^T = (\tilde{H}_{:,j} W_H^Q W_P^{Q^T} P_{:,j}^T)(P_{:,j} W_P^K W_H^{K^T} \tilde{H}_{:,j}^T) \tag{4}$$

$$= (\tilde{H}_{:,j} W^Q P_{:,j}^T)(P_{:,j} W^{K^T} \tilde{H}_{:,j}^T) \tag{5}$$

where $W^Q = W_H^Q W_P^{Q^T}, W^K = W_H^K W_P^{K^T}$. Unlike conventional channel attention, which treats channels as independent tokens or flattens all patches, EPA attention incorporates electrode positions and cortical region groupings, reflecting the spatial relationships inherent to EEG acquisition.

## 2.2 Noise-robust Neural Tokenizer Training

Brain activity emerges as continuous signals over time, providing measurable patterns of underlying neuronal processes. Although neural signals are continuous in form, the internal representation in the brain is inherently discrete (Tee & Taylor, 2020; Perotti et al., 2019). This motivates the need for models that can transform continuous neural recordings into discrete embeddings. VQ-VAE (van den Oord et al., 2017) provides a natural framework for encoding continuous EEG signals into discrete tokens while preserving essential biological information. To achieve strong generalizability, a foundation model should be robust to the diverse sources of noise inherent in signal acquisition. Thus, we train a noise-robust neural tokenizer using denoising temporal-spectral prediction.

**Noise Augmentation.** Motivated by the denoising autoencoder (Vincent et al., 2008), we augment the EEG signals by injecting noise and feed them to our neural tokenizer. We choose Gaussian noise, as it is statistically independent and cannot be removed through rule-based filtering techniques. In BCI systems, although representative noise components are filtered, residual noise inevitably remains in EEG signals (Tanner et al., 2015). Gaussian noise also affects all frequency bands, with different power distributed across frequencies depending on its amplitude. VQ-VAE is prone to codebook degeneration when trained solely on noisy data (Ho et al., 2022). To capture key information in EEG signals while enhancing robustness to noise, we randomly augment half of the samples in each batch. We denote noise-augmented sample as $\tilde{x} = \{\tilde{x}_{ij} = x_{ij} + \sigma \in \mathbb{R}^P | i = 1, ..., C, j = 1, ..., N, \sigma \sim \mathbf{N}(0, \delta^2)\}$.

**Neural Tokenizer.** The neural tokenizer consists of the VQ encoder, the codebook, and the temporal decoder. The codebook $V = \{v_i | i = 1, ..., K\} \in \mathbb{R}^{K \times D}$ contains $K$ discrete tokens of dimension $D$. Given a noise-augmented EEG sample $\tilde{x}$ (or $\mathbf{x}$), the VQ encoder encodes it to patch representations $h_{i,j}$. Then, we select the nearest codes of $h_{i,j}$ from codebook embeddings:

$$z_{i,j} = argmin_k ||l_2(h_{i,j}) - l_2(v_k)||_2, \tag{6}$$

where $j \in \{1, ..., K\}$ and $l_2$ indicates $l_2$ normalization. Consequently, an EEG sample is tokenized to $\mathbf{z} = \{z_{i,j} | i = 1, ..., C, j = 1, ..., N\}$.

**Denoising Temporal-spectral Prediction.** Gaussian noise affects the entire spectrum, with weaker impact on low frequencies and stronger impact on high frequencies. Neural tokenizer–based EEG foundation models struggle to learn frequency-domain information. LaBraM predicts amplitude and phase separately, but its phase prediction loss fails to converge, while NeuroLM predicts temporal and amplitude signals but lacks accurate phase modeling, limiting its ability to represent full spectral content. To address these limitations, we propose predicting the original signals in both real and complex domains to capture full frequency information. Specifically, we apply the Discrete Fourier Transform (DFT) to an EEG patch $x_{i,j} = [x[1], x[2], ..., x[P]]$ of channel $i$ and time $j$:

$$\bar{f}_{i,j}^m = \sum_{t=1}^{P} x[t] \exp(-\frac{2\pi \boldsymbol{j}}{P} mt) \tag{7}$$

c where $m \in [1, P]$ and $\boldsymbol{j}$ is the imaginary unit. We calculate the real and imaginary parts of $x_{i,j}^m$ as $A^m = \sqrt{Re(f_{i,j}^m) + Im(f_{i,j}^m)}$ and $\phi^m = \arctan(\frac{Im(f_{i,j}^m)}{Re(f_{i,j}^m)})$. To emphasize learning phase information, we construct a normalized spectral representation in which the amplitudes are normalized to

sum to one:

$$f_{i,j}^m = norm(\bar{f}_{i,j}^m) = \frac{A^m}{\sum_{m'=1}^{M} A^{m'}} e^{-j\phi} \tag{8}$$

The normalized codebook embeddings are fed into the temporal decoder, which reconstructs the EEG patches in the temporal domain, denoted by $o_{i,j}^t$. Rather than building a decoder for frequency reconstruction, we derive the normalized spectral representation by applying the DFT to the reconstructed EEG patch, denoted by $o_{i,j}^f = norm(DFT(o_{i,j}^t))$. The total loss for training noise-robust neural tokenizer is defined as:

$$L_{NT} = \sum_{x \in \mathcal{D}} \sum_{i,j} ||o_{i,j}^t - x_{i,j}||_2^2 + ||o_{i,j}^f - f_{i,j}||_2^2 + ||\boldsymbol{sg}((l_2(h_{i,j})) - l_2(v_{z_{i,j}})||_2^2 + ||(l_2(h_{i,j}) - \boldsymbol{sg}(l_2(v_{z_{i,j}}))||_2^2 \tag{9}$$

where $\mathcal{D}$ is the EEG dataset and $\boldsymbol{sg}$ denotes the stop-gradient operation that is an identity during forward and has zero gradients. Since the DFT is differentiable, $L_{NT}$ can be optimized using standard stochastic gradient descent methods (Wang et al., 2025a).

## 2.3 NERVE Variability-robust Pre-training

**Masked EEG Modeling.** We pre-train NERVE following the masked EEG modeling strategy of LaBraM. Given an EEG sample $x$, the patch encoder projects each EEG patch into patch embeddings $e_{i,j}$. A subset of patches is then randomly masked with ratio $r$ and replaced by a learnable mask token $e_M \in \mathbb{R}^d$. We have EEG patches $e^{\mathcal{M}} = \{e_{i,j}|(i,j) \notin \mathcal{M}_r\} \cup \{e_M|(i,j) \in \mathcal{M}_r\}$, where $\mathcal{M}_r$ is the set of indices of masked patches. The masked EEG signal is augmented with time and channel embeddings and then fed into the EPA transformer encoder. The encoder output is denoted as $\mathbf{h} = \{h_{i,j}, |, i = 1, \ldots, C, , j = 1, \ldots, N\}$, which is passed through a linear classifier to predict the neural codes of the masked patches. The objective for masked EEG modeling is:

$$L_{MEM} = -\sum_{x \in B} \sum_{(i,j) \in \mathcal{M}} \text{softmax}(\text{Linear}(h_{i,j})) \tag{10}$$

**Variability-robust Learning.** EEG signals, being collected from human subjects, exhibit substantial variability both across and within individuals. To achieve robustness against such variability, the model must distinguish samples from the same subject as well as from different subjects (Cheng et al., 2020). In the representation space, alignment enhances the consistency of samples from the same subject (i.e., intra-subject consistency), whereas uniformity improves the separability of samples from different subjects (i.e., inter-subject separability) (Wang & Isola, 2020). As masked modeling implicitly improves alignment in the representation space (Zhang et al., 2022), we employ Kozachenko–Leonenko (KoLeo) regularization (Sablayrolles et al., 2019) to enforce uniformity and consequently achieve inter-subject separability. For each sample, an embedding $h$ is computed by average pooling over the output of EPA transformer encoder and KoLeo loss is formulated as:

$$L_{KoLeo} = -\frac{1}{|B|} \sum_{i=1}^{|B|} log(min_{j \neq i}||l_2(h_i) - l_2(h_j)||^2) \tag{11}$$

where $B$ is the batch. By enhancing the representation space, NERVE can achieve robustness to variability without relying on subject- or session-specific information. Finally, the overall pre-training loss for variability-robust NERVE is:

$$L = L_{MEM} + \alpha L_{KoLeo} \tag{12}$$

where $\alpha$ is the regularization coefficient for KoLeo loss.

## 3 Experiments

### 3.1 Pre-training

**Pre-training Dataset.** We collected 27 publicly available EEG datasets with a total of 16,595 hours of recordings. To enhance NERVE's generalization capability, we curated the pre-training corpus to cover a broad spectrum of BCI tasks, encompassing 10 distinct use cases, such as seizure detection, gait recognition and emotion recognition. Detailed descriptions of all pre-training datasets and tasks are provided in Table 7 and Appendix B.2.

**Preprocessing.** We adopt the procedures aligned with standard raw EEG processing pipelines applied during sensor acquisition (Kim & Im, 2021; Temko et al., 2011; Kostas et al., 2021; Wang et al., 2025b; Jiang et al., 2024; Ding et al., 2024; Singh & Krishnan, 2023; Apicella et al., 2023). The detailed preprocessing steps and the underlying rationale are provided in Appendix C.3. After preprocessing, we obtain 213,784 EEG samples retained for pre-training and 10,956 hours in total, which is a longer duration than the pre-training datasets from LaBraM (2,534.78 hours) and CBraMod (9,000 hours).

**Pre-training Settings.** We define the NERVE backbone as a 12-layer EPA Transformer with 200 hidden dimensions, a feed-forward network of 800 dimensions, 10-head EPA attention, and a position router with 6 cortical region groups. The patch size $P$ was set to 200 (1 second), and the number of total patches in the EEG sample was limited to 256 to stabilize resource usage. The stride was fixed at 1 second for noiser-robust neural tokenizer and 4 seconds for NERVE model. The noise-robust neural tokenizer and NERVE is pre-trained sequentially on four NVIDIA A100-80G GPUs for about 5 days. More details of the architecture settings can be found in Appendix B.1.

## 3.2 EXPERIMENT SETUP OF DOWNSTREAM BCI TASKS

**Downstream BCI Tasks and Datasets.** To evaluate the performance of NERVE, We conduct downstream experiments on six BCI tasks across 8 datasets. These evaluation datasets are widely recognized for their substantial noise levels and pronounced variability (Obeid & Picone, 2016; Al-Hadithy et al., 2025; Schirrmeister et al., 2017; Zhang et al., 2025b; Soleymani et al., 2011; Nicolas-Alonso & Gomez-Gil, 2012). A summary of all downstream tasks and their corresponding datasets is provided in Table 2. Note that all downstream datasets are not included in the pre-training datasets, so we can evaluate the generalization performance of NERVE. We resampled the EEG signals in the downstream datasets into 200 Hz and set the patch length as 200 (1 second) for the consistency with the pre-training dataset. A detailed description of each dataset is provided in Appendix C.2.

**Baselines & Metrics**. We compare NERVE with both non-foundation model and foundation model baselines to ensure rigorous evaluation: EEGNet (Lawhern et al., 2018), EEGConformer (Song et al., 2022), ContraWR (Yang et al., 2021), CNN-Transformer (Peh et al., 2022), FFCL (Li et al., 2022), and ST-Transformer (Song et al., 2021) for non-foundation model baselines, BIOT (Yang et al., 2023), LaBraM (Jiang et al., 2024), and CBraMod (Wang et al., 2025b), NeuroLM (Jiang et al., 2025b) for foundation model baselines. We re-implemented all baselines using the public code and pre-trained weights in our evaluation setting, as they were reported to be to be difficult to reproduce. For LaBraM, we only fine-tune LaBraM-base, for which pre-trained weights are publicly available. We followed the evaluation metrics from CBraMod (Wang et al., 2025b): balanced accuracy, AUC-PR, and AUROC for binary classification, and balanced accuracy, Cohen's Kappa, and weighted F1 for multi-class classification, and Pearson's correlation, R2 score, and RMSE for regression. We obtain all the results with five different random seeds.

Table 2: Overview of downstream BCI tasks and datasets.

| Tasks | Datasets | # Channels | # Recordings | # Subjects | Sampling Rate | Duration | Label |
|---|---|---|---|---|---|---|---|
| Seizure Detection | TUSL | 20 | 20,750 | 38 | 256 Hz | 3.91 ms | 3-class |
| Emotion Recognition | SEED-V | 62 | 32,706 | 16 | 1000 Hz | 1 ms | 5-class |
| | DEAP | 32 | 30,745 | 32 | 512 Hz | 1.95 ms | 2-class |
| | HCI-Tagging Emotion | 32 | 36,435 | 27 | 256 Hz | 3.9 ms | 2-class |
| Motor Imagery Classification | High-Gamma | 73 | 3,076,357 | 14 | 500 Hz | 180–240 ms | 4-class |
| Event Type Classification | TUEV | 23 | 112,464 | - | 256 Hz | 30 ms | 6-class |
| | BCI-NER Challenge | 56 | 158,264 | 26 | 200 Hz | 5 ms | 2-class |
| Vigilance Prediction | SEED-VIG | 17 | 1,600 | 23 | 200 Hz | 5 ms | Regression |

## 3.3 EXPERIMENTAL RESULTS

We evaluate NERVE and baselines on 8 public datasets for five downstream BCI tasks (see Table 2). In all experiments, we strictly follow the equivalent evaluation setting for every method. In this section, we present the experimental results on four datasets, TUSL, High-Gamma, DEAP, and BCI-NER Challenge. More results are provided in Appendix D. Table 3 presents the evaluation results on TUSL (seizure detection) and High-Gamma (motor imagery classification). NERVE consistently

Table 3: The evaluation results on TUSL and High-Gamma (multi-class classification)

| Methods | TUSL, 3-class | | | High-Gamma, 4-class | | |
| --- | --- | --- | --- | --- | --- | --- |
| | Balanced Accuracy | Cohen's Kappa | Weighted F1 | Balanced Accuracy | Cohen's Kappa | Weighted F1 |
| EEGNet | 0.4167 ± 0.0934 | 0.1265 ± 0.1365 | 0.4040 ± 0.0813 | 0.8231 ± 0.1077 | 0.7641 ± 0.1437 | 0.8255 ± 0.1059 |
| EEGConformer | 0.4933 ± 0.0733 | 0.2501 ± 0.1133 | 0.4458 ± 0.0970 | 0.5776 ± 0.1584 | 0.4368 ± 0.2111 | 0.5451 ± 0.1807 |
| ContraWR | 0.4533 ± 0.0499 | 0.1672 ± 0.0779 | 0.3941 ± 0.0625 | 0.9474 ± 0.0257 | 0.9298 ± 0.0343 | 0.9486 ± 0.0246 |
| CNN-Transformer | 0.4783 ± 0.0676 | 0.1902 ± 0.1039 | 0.3794 ± 0.0591 | 0.8996 ± 0.0139 | 0.8662 ± 0.0185 | 0.9021 ± 0.0137 |
| FFCL | 0.4500 ± 0.1217 | 0.1790 ± 0.1895 | 0.4112 ± 0.1694 | 0.9528 ± 0.0177 | 0.9371 ± 0.0236 | 0.9537 ± 0.0170 |
| ST-Transformer | 0.3400 ± 0.0134 | 0.0079 ± 0.0159 | 0.1525 ± 0.0313 | 0.5017 ± 0.1678 | 0.3356 ± 0.2238 | 0.4562 ± 0.1909 |
| BIOT | 0.5150 ± 0.0883 | 0.2562 ± 0.1385 | 0.4531 ± 0.0912 | 0.4918 ± 0.0234 | 0.3224 ± 0.0312 | 0.4638 ± 0.0313 |
| LaBraM | 0.6806 ± 0.0515 | 0.5020 ± 0.0800 | 0.6386 ± 0.0490 | 0.9855 ± 0.0037 | 0.9807 ± 0.0049 | 0.9856 ± 0.0037 |
| CBraMod | 0.4467 ± 0.0598 | 0.1420 ± 0.0939 | 0.4103 ± 0.0641 | 0.9189 ± 0.0097 | 0.8919 ± 0.0130 | 0.9190 ± 0.0102 |
| NeuroLM | 0.4867 ± 0.0908 | 0.2140 ± 0.1383 | 0.4780 ± 0.0920 | 0.9590 ± 0.0054 | 0.9454 ± 0.0072 | 0.9597 ± 0.0052 |
| NERVE | **0.7000** ± 0.0282 | **0.5327** ± 0.0411 | **0.6507** ± 0.0329 | **0.9906** ± 0.0011 | **0.9875** ± 0.0015 | **0.9906** ± 0.0011 |

Table 4: The evaluation results on DEAP and BCI-NER Challenge (binary classification)

| Methods | DEAP, 2-class | | | BCI-NER Challenge, 2-class | | |
| --- | --- | --- | --- | --- | --- | --- |
| | Balanced Accuracy | AUC-PR | AUROC | Balanced Accuracy | AUC-PR | AUROC |
| EEGNet | 0.4758 ± 0.0242 | 0.5511 ± 0.0161 | 0.4190 ± 0.0387 | 0.5045 ± 0.0025 | 0.2874 ± 0.0094 | 0.5188 ± 0.0098 |
| EEGConformer | 0.5001 ± 0.0121 | 0.5688 ± 0.0256 | 0.4400 ± 0.0252 | 0.4998 ± 0.0003 | 0.2736 ± 0.0105 | 0.5058 ± 0.0117 |
| ContraWR | 0.4581 ± 0.0203 | 0.6242 ± 0.0534 | 0.4491 ± 0.0445 | 0.4995 ± 0.0007 | 0.2873 ± 0.0121 | 0.5371 ± 0.0134 |
| CNN-Transformer | 0.4663 ± 0.0176 | 0.6305 ± 0.0218 | 0.4681 ± 0.0143 | 0.4999 ± 0.0001 | 0.2748 ± 0.0176 | 0.5108 ± 0.0248 |
| FFCL | 0.4598 ± 0.0198 | 0.6101 ± 0.0226 | 0.4490 ± 0.0146 | 0.4995 ± 0.0023 | 0.2762 ± 0.0079 | 0.5164 ± 0.0113 |
| ST-Transformer | 0.4338 ± 0.0266 | 0.5523 ± 0.0199 | 0.3997 ± 0.0273 | 0.5009 ± 0.0008 | 0.2935 ± 0.0131 | 0.5290 ± 0.0078 |
| BIOT | 0.4889 ± 0.0118 | 0.5190 ± 0.0086 | 0.3552 ± 0.0138 | 0.4990 ± 0.0010 | 0.2905 ± 0.0200 | 0.5329 ± 0.0308 |
| LaBraM | 0.5002 ± 0.0003 | 0.5824 ± 0.0155 | 0.4152 ± 0.0161 | 0.5000 ± 0.0001 | 0.2671 ± 0.0149 | 0.4883 ± 0.0306 |
| CBraMod | 0.4511 ± 0.0268 | 0.5666 ± 0.0517 | 0.4068 ± 0.0617 | 0.5115 ± 0.0133 | 0.3004 ± 0.0144 | 0.5305 ± 0.0067 |
| NeuroLM | 0.4734 ± 0.0051 | 0.6129 ± 0.0023 | 0.4734 ± 0.0051 | 0.5000 ± 0.0000 | 0.2729 ± 0.0000 | 0.5000 ± 0.0000 |
| NERVE | **0.5167** ± 0.0131 | **0.6533** ± 0.0156 | **0.5406** ± 0.0267 | **0.5245** ± 0.0189 | **0.3181** ± 0.0159 | **0.5504** ± 0.0181 |

outperforms all baselines on both datasets. NERVE achieves the best performance on High-Gamma, effectively capturing class-wise topographic patterns (see Section H.2 and Figure 13). Despite the performance instability caused by TUSL's small sample size, NERVE still yields improved results. Furthermore, the consistent and higher performance observed on DEAP and BCI-NER Challenge (Table 4) strongly highlights NERVE's capacity and generalizability across diverse BCI tasks.

## 4 ANALYSIS & ABLATION STUDIES

**Robustness to Noise.** We evaluate the noise robustness of foundation models across diverse scenarios. For comparison, we select LaBraM and CBraMod as the two strongest baselines. Models are fine-tuned on clean training and validation sets and evaluated on test sets injected with synthetic noise. The noise types include electromyogram (EMG), electrooculogram (EOG), and environmental noise, commonly arising during EEG acquisition. We include Gaussian noise directly added to the time-domain signals, with values randomly sampled from a Gaussian distribution. We also generate two additional scenarios by controlling SNR of the input signals to 5 dB and 10 dB, as low SNR is the representative challenge of our work. Under these scenarios, we evaluate the performance degradation of foundation models, where smaller degradation indicates greater noise robustness. Additionally, we measure the predictive entropy since a noise-robust model is expected to maintain a stable predictive distribution, where smaller changes indicate greater robustness to noise. Figure 4 displays the analysis result. NERVE exhibited the smallest accuracy degradation and minimal change in predictive entropy change across most noise conditions. Performance remained relatively stable under frequency-specific noise, but Gaussian noise and low SNR scenarios significantly hindered the reliable prediction of EEG foundation models. Since these conditions represent the most commonly observed, unstructured patterns in BCI tasks, NERVE's superior robustness to Gaussian and low SNR conditions highlights its practical utility. The observation holds consistently for other datasets, with further details and analyses available in Appendix G.1.

**Robustness to Variability.** To be robust to intra-subject variability, models must yield consistent predictions across diverse physiological states of the same subject. We quantify this robustness by defining intra-subject distance as the standard deviation of predicted logits across samples of the same class within a subject. A lower distance indicates greater robustness to intra-subject variabil-

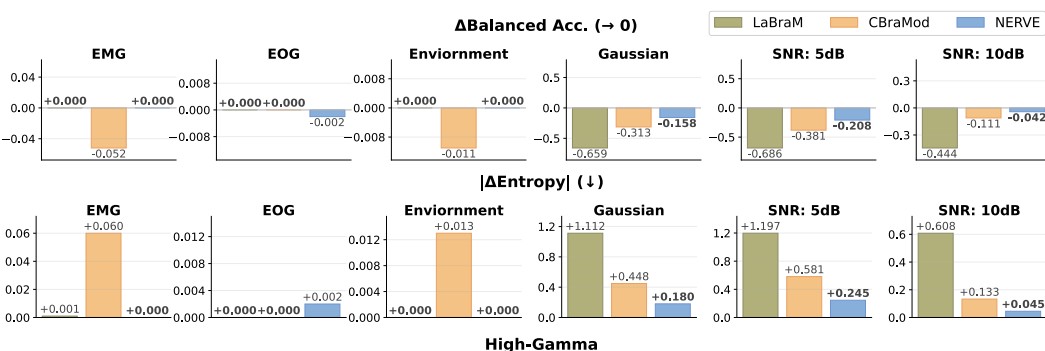

Figure 4: Noise robustness analysis. NERVE shows the minimal accuracy degradation and maintained stable predictions under diverse synthetic noise conditions. Arrows in the title point toward higher robustness.

ity. Figure 5 (a) shows the results on HCI-Tagging Emotion and TUSL, where NERVE consistently achieves lower values than existing foundation models, confirming its superior robustness. To ensure robustness to inter-subject variability, the learned representations of the model must effectively separate subject-specific patterns on each class in the embedding space. To assess the robustness of pre-trained foundation models to inter-subject variability, we define the inter-subject distance for each class as the average pairwise distance between $l_2$-normalized subject embeddings within the class, where each embedding is the mean of all sample embeddings from the corresponding subject. A greater distance indicates higher inter-subject separability and thus greater robustness. As shown in Figure 5 (b), NERVE consistently exhibits the highest inter-subject distances across all classes on High-Gamma, highlighting its ability to capture subject-specific patterns within each class. The robustness to such subject variabilities is attributed to NERVE's variability-robust learning, which encourages embedding uniformity through KoLeo regularization. When pre-trained without KoLeo, NERVE shows a remarkable decline in robustness, validating the importance of variability-robust learning for acquiring generalizable representations independent of subject information.

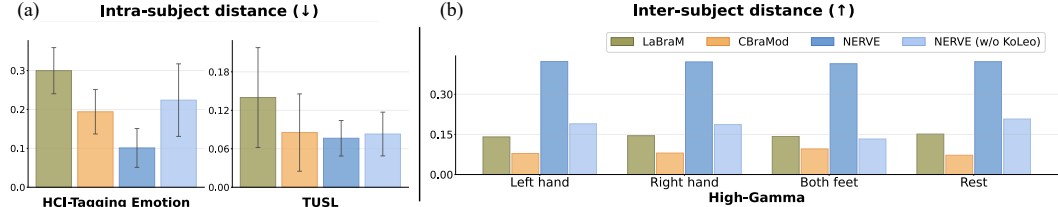

Figure 5: Variability robustness analysis. **Intra-subject variability**: NERVE achieves the lowest intra-subject distance, highlighting the superior robustness to intra-subject variability. (c) **Inter-subject variability**: NERVE exhibits the highest inter-subject distance across all classes, indicating strong robustness to inter-subject variability. Arrows in the title point toward higher robustness.

## 5 CONCLUSION

In this paper, we develop NERVE, the first EEG foundation model explicitly designed to address acquisition-related challenges: low SNR, high variability, and spatial dependencies among electrodes. Through noise-robust neural tokenizer training, variability-robust pre-training, and electrode-position-aware attention, NERVE consistently achieves competitive performance and strong robustness to noise and variability over existing foundation models. Our work presents the necessity of modeling signal-acquisition-specific characteristics and establishes NERVE as a practical and generalizable foundation for future EEG research and BCI applications. Our work has some limitations. Due to resource constraints, we have not explore the scaling laws of EEG pre-training using larger and more diverse pre-training datasets. In addition, real-world clinical validation remains necessary to fully evaluate the practical applicability of EEG foundation models.

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

## A RELATED WORK

### A.1 SUPERVISED EEG MODELS

With the advancement of deep learning, supervised neural networks have shown significant improvements in decoding EEG signals. Zhou et al. (2018) proposed a CNN-based seizure detection model that directly takes raw EEG inputs in both time and frequency domains. To capture sequential dependencies in EEG signals, Roy et al. (2019) introduced ChronoNet, a GRU-based recurrent neural network with multi-scale 1D convolutions and densely connected layers. Wan et al. (2023) introduced EEGformer, a Transformer-based model that applies separate temporal, regional, and synchronous transformer modules. EEGConformer (Peh et al., 2022) integrates CNN-based local feature extraction with self-attention to model global temporal dependencies in EEG signals. Additionally, ACM-GNN (Zhang et al., 2025a) proposed a graph neural network designed to capture

inter-electrode topological relationships and individual-specific brain network modularity through dynamic clustering and hierarchical graph reasoning. These supervised models, although effective for specific tasks, rely heavily on labeled data and often struggle to generalize across subjects, recording conditions, and electrode configurations. Obtaining high-quality EEG annotations is particularly challenging due to the need for expert labeling, long-duration recordings, and ethical or privacy constraints, which further complicates efforts to improve generalization in EEG decoding.

## A.2 EEG FOUNDATION MODELS

Foundation models in NLP and CV have demonstrated strong capability in learning from large-scale unlabeled data and transferring knowledge to a variety of downstream tasks. Building on these successes, foundation model approaches have recently been extended to EEG signals. BIOT (Yang et al., 2023) proposed a generic biosignal transformer that tokenizes heterogeneous EEG signals of varying lengths and formats into unified sequential structures. EEGPT (Wang et al., 2024) introduces a dual self-supervised learning framework that combines spatio-temporal representation alignment with masked reconstruction. LaBraM (Jiang et al., 2024) pre-trains a Transformer on masked EEG token prediction using a vector-quantized tokenizer trained via Fourier spectrum reconstruction. NeuroLM (Jiang et al., 2025b) introduces a unified EEG foundation model that aligns EEG signals with natural language using a vector-quantized tokenizer and instruction tuning. CBraMod (Wang et al., 2025b) leverages a criss-cross Transformer architecture with patch-wise masked reconstruction and asymmetric positional encoding.

Despite their effectiveness in diverse BCI tasks, existing EEG foundation models largely focus on addressing heterogeneity in channel configurations and time-series lengths. Unlike other modalities, EEG signals are recorded from human subjects through scalp-mounted sensor hardware in real-world experiments. Consequently, EEG foundation models should account for structural constraints inherent to hardware-based acquisition. Variability in sensor and subject conditions introduces unavoidable measurement noise and variability. Moreover, the spatial arrangement of electrodes induces dependencies arising from interactions between electrode positions and underlying cortical regions. Therefore, building practical and generalizable EEG foundation models requires explicitly modeling these acquisition-specific characteristics.

## A.3 NOISE AND VARIABILITY IN EEG

EEG signals are inherently contaminated by noise from both external and physiological sources. EEG electrodes detect weak cortical potentials from the scalp surface, making them susceptible to electrical interference and contact instability, while artifacts from eye movements or muscle activity further distort the signals and hinder accurate decoding. Although EEG noise degrades task performance, noise-robust deep learning approaches have shown notable gains in downstream tasks (Xiong et al., 2024; Jiang et al., 2025a). Beyond noise, EEG signals also exhibit high variability across and within individuals, stemming from differences in anatomical and functional brain connectivity, head shape, cognitive states, and environmental conditions. Such variability are difficult to observe directly from EEG signals and poses a significant challenge to model generalization (Ma et al., 2022b). These findings suggest that EEG foundation models must achieve robustness to noise and variability to ensure generalizability and practical applicability. We achieve robustness against such robustness through denoising-based training of the neural tokenizer and variability-robust pre-training of the NERVE foundation model.

## A.4 SPATIAL DEPENDENCY BETWEEN ELECTRODES

EEG recordings inherently contain channel dependencies that arise from the acquisition process. Because EEG captures localized cortical activity through scalp electrodes, the electrode topology defines the spatial dependencies present in EEG signals. Thus, recognizing the spatial pattern in EEG signals contributes to the understanding of the physiological basis of brain states. To exploit the spatial structure, several studies have proposed topology-aware neural architectures. Graph-based attention models, including EEG-GMACN (Ye et al., 2024) and STGATE (Li et al., 2023), leverage adjacency matrices within the attention mechanism to enhance performance on BCI tasks.

Table 5: Hyperparameters for noise-robust neural tokenizer training.

| | Hyperparameters | Values |
|---|---|---|
| Patch encoder (3-layer 1D CNN) | Input channels | {1,8,8} |
| | Output channels | {8,8,8} |
| | Kernel size | {15,3,3} |
| | Stride | {8,1,1} |
| | Padding | {7,1,1} |
| Backbone | EPA Transformer encoder layers | 9 |
| | EPA Transformer decoder layers | 3 |
| | Hidden size | 200 |
| | MLP size | 800 |
| | Attention head number | 10 |
| | Number of position router groups | 6 |
| | Codebook size | $8192 \times 64$ |
| Training | Batch size | 1024 |
| | Peak learning rate | $5e^{-5}$ |
| | Minimal learning rate | $1e^{-5}$ |
| | Learning rate scheduler | Cosine |
| | Optimizer | AdamW |
| | Adam $\beta$ | (0.9, 0.999) |
| | Weight decay | $1e^{-4}$ |
| | Dropout | 0.1 |
| | Total epochs | 20 |
| | Warmup epochs | 2 |
| | Data stride | 200 |
| Noise augmentation | Standard deviation | 0.05 |

However, existing EEG foundation models do not account for the topological structure of EEG signals. They can be categorized into two spatial dependency modeling strategies: (1) full modeling, which applies self-attention to flattened EEG patches and treats dependencies across all patches equally, and (2) criss-cross attention, which disregards the spatial relationships among electrodes and their corresponding brain regions. This limitation motivates our EPA attention, which explicitly leverages electrode topology to model spatial dependencies in EEG signals into the attention score. The EPA Transformer is utilized as the encoder for NERVE foundation model, which captures spatial dependencies emerging from signal acquisition, enhancing both the expressiveness and interpretability of the prediction.

# B DETAILS FOR EXPERIMENTAL SETUP FOR PRE-TRAINING

## B.1 HYPERPARAMETER SETTINGS

Hyperparameters for noise-robust neural tokenizer training and NERVE variability-robust pre-training are presented in Table 5 and 6.

## B.2 PRETRAINING DATASETS DESCRIPTION

- **Resting State EEG Data (Trujillo et al., 2017)**: Resting State EEG Data dataset consists of EEG recordings collected at Texas State University from 22 healthy adult participants during an 8-minute resting-state protocol. Participants alternated between 1-minute blocks of eyes-open and eyes-closed conditions in a darkened room while seated comfortably. EEG was recorded using 64 channels, and the data were analyzed to examine how different reference electrode schemes affect measures of signal complexity and integration (used channels: 64).

- **Neonate (Stevenson et al., 2019)**: Neonate dataset consists of EEG recordings collected from 79 neonates in the Neonatal Intensive Care Unit (NICU) under clinical suspicion of seizures. Each subject contributed one continuous EEG session, recorded using the

Table 6: Hyperparameters for NERVE variability-robust pre-training.

| | Hyperparameters | Values |
|---|---|---|
| Patch encoder (3-layer 1D CNN) | Input channels | {1,8,8} |
| | Output channels | {8,8,8} |
| | Kernel size | {15,3,3} |
| | Stride | {8,1,1} |
| | Padding | {7,1,1} |
| Backbone | EPA Transformer encoder layers | 12 |
| | Hidden size | 200 |
| | MLP size | 800 |
| | Attention head number | 10 |
| | Number of position router groups | 6 |
| Training | Batch size | 512 |
| | Peak learning rate | $3e^{-4}$ |
| | Minimal learning rate | $1e^{-5}$ |
| | Learning rate scheduler | Cosine |
| | Optimizer | AdamW |
| | Adam $\beta$ | (0.9, 0.999) |
| | Weight decay | $5e^{-2}$ |
| | Dropout | 0.1 |
| | Total epochs | 10 |
| | Warmup epochs | 1 |
| | Gradient clipping | 3 |
| | Data stride | 800 |
| | Masking ratio | 0.5 |
| | Alpha | 0.1 |

Table 7: Overview of Pre-training datasets

| Tasks | Datasets | # Channels | # Recordings | # Subjects | Sampling Rate | Duration | Total Time |
|---|---|---|---|---|---|---|---|
| Raw Data | Resting State EEG Data | 64 | 127,174 | 22 | 256 Hz | 4 | 3 hr |
| Seizure Detection | Neonate | 19 | 802,048 | 79 | 256 Hz | 3.91 ms | 57 hr |
| | Siena | 28 | 37,667,840 | 14 | 512 Hz | 1.95 ms | 141 hr |
| | TUSZ | 20 | 7,936 | 675 | 256 Hz | 3.91 ms | 1,474 hr |
| Sleep Stage Detection | 2018 PhysioNet Challenge | 6 | 5,551,404 | 1,985 | 200 Hz | 1800 ms | 14,611 hr |
| Emotion Recognition | SEED | 62 | 226,266 | 15 | 1000 Hz | 1 ms | 42 hr |
| | SEED-IV | 62 | 34,309 | 15 | 200 Hz | 5 ms | 42 hr |
| | DREAMER | 14 | 26,510 | 23 | 128 Hz | 7.81 ms | 24 hr |
| | Emobrain | 64 | 1,147,238 | 5 | 1024 Hz | 1 ms | 3 hr |
| Motor Imagery Classification | BCI IV-2a | 22 | 669,259 | 9 | 250 Hz | 90 ms | 13 hr |
| | BCI IV-2b | 3 | 525,501 | 9 | 250 Hz | 480 ms | 26 hr |
| | BCI IV-1 | 59 | 2,111,625 | 7 | 1000 Hz | 90–480 ms | 14 hr |
| | SHU-MI | 32 | 1,000 | 25 | 250 Hz | 4 ms | 799 hr |
| Event Type Classification | Inria BCI Challenge | 56 | 193,660 | 16 | 600 Hz | 1.7 ms | 14 hr |
| | Target vs Non-target | 32 | 166,663 | 43 | 512 Hz | 2 ms | 4 hr |
| | MoBI | 60 | 135,009 | 8 | 100 Hz | 10 ms | 9 hr |
| | BCIC2020-3 | 64 | 795 | 15 | 256 Hz | 3.91 ms | 272 hr |
| Vigilance Prediction | SPIS Resting State | 64 | 38,401 | 10 | 256 Hz | 4 ms | 1 hr |
| | FatigueSet | 4 | 8,233 | 12 | 256 Hz | 3.91 ms | 11 hr |
| Workload Classification | Stew | 14 | 19,200 | 48 | 128 Hz | 7.81 ms | 4 hr |
| | Raw EEG Data | 64 | 413,568 | - | 256 Hz | 4 ms | 34 hr |
| | Mental Arithmetic | 19 | 651,000 | 36 | 500 Hz | 2 ms | 2 hr |
| | Berlin (dsr) | 28 | 432 | 26 | 200 Hz | 5 ms | 5 hr |
| | Berlin (nback) | 28 | 432 | 26 | 200 Hz | 5 ms | 8 hr |
| | Berlin (wg) | 28 | 6,099 | 26 | 200 Hz | 5 ms | 13 hr |
| Mental Disorder Diagnosis | Mumtaz2016 | 19 | 104,664 | 64 | 256 Hz | 3.91 ms | 10 hr |
| Gal Recognition | Grasp and Lift EEG Challenge | 32 | 176,083 | 12 | 500 Hz | 2 ms | 9 hr |

standard 10–20 system with 19 channels. All data were annotated on a 1-second basis by three independent experts who visually inspected seizure activity (used channels: 19).

- **Siena (Detti, 2020)**: Siena dataset consists of scalp EEG recordings with 28 channels designed for seizure detection, collected at the University of Siena from 14 epilepsy patients. Each subject contributed between 1 to 5 EEG recording sessions conducted under identical experimental conditions while patients were lying in bed, either awake or asleep. Annotations for seizure intervals were provided by clinical experts (used channels: 28).

- **TUSZ (Shah et al., 2018)**: TUSZ dataset consists of 1,643 EEG sessions collected from 675 patients with 20 channels as a subset of the Temple University Hospital (TUH) EEG Corpus. Developed by the Neural Engineering Data Consortium (NEDC), this dataset was created for seizure detection research. Of these sessions, 528 contain annotated seizure events, while the rest include only normal background activity, making it suitable for both training and false alarm evaluation (used channels: 20).

- **2018 PhysioNet Challenge (Ghassemi et al., 2018)**: 2018 PhysioNet Challenge dataset comprises polysomnographic recordings collected from 1,985 subjects at the Massachusetts General Hospital (MGH) Sleep Laboratory for the purpose of sleep disorder diagnosis. The dataset includes multiple physiological signals such as EEG, EOG, and EMG. In this study, we exclusively utilize the EEG modality, which consists of 6 channels. Sleep stages were annotated every 30 seconds according to the AASM standard, covering wakefulness, stages 1–3, REM, and undefined states. Arousals and other sleep-related events were manually annotated by certified technicians (used channels: 6).

- **SEED (Zheng & Lu, 2015)**: SEED dataset consist of EEG and eye movement recordings collected from 15 subjects at the BCMI Lab of Shanghai Jiao Tong University. EEG was recorded using 62 channels at 1000 Hz while participants watched movie clips designed to elicit positive, negative, and neutral emotions. Each subject completed three sessions, and each session contained 15 trials (used channels: 62).

- **SEED-IV (Zheng et al., 2018)**: SEED-IV dataset consists of EEG and eye movement recordings collected from 15 subjects at the BCMI Lab of Shanghai Jiao Tong University. EEG was recorded using 62 channels at 200 Hz while participants watched movie clips designed to elicit happiness, sadness, fear, and neutral emotions. Each subject completed three sessions on separate days, and each session consisted of 24 trials (used channels: 62).

- **DREAMER (Katsigiannis & Ramzan, 2017)**: DREAMER dataset comprises EEG recordings collected from 23 subjects in a controlled laboratory setting at the University of the West of Scotland. EEG was recorded using 14 channels at 128 Hz while participants watched 18 emotion-eliciting video clips. After each clip, participants provided self-ratings of valence, arousal, and dominance. Each subject completed one session (used channels: 14).

- **Emobrain (Savran[1] et al., 2006)**: Emobrain dataset consists of multimodal recordings collected during eNTERFACE'06 Workshop from 5 subjects. It includes EEG recorded with 64 channels at 1024 Hz, fNIRS, facial video, and peripheral physiological signals such as GSR, respiration, and blood pressure. Participants viewed IAPS images across three sessions, each consisting of 30 blocks for a total of 90 blocks and 450 images. Emotional stimuli were categorized into calm, positive-exciting, and negative-exciting conditions, and participants provided self-assessments of valence and arousal after each block (used channels: 64).

- **BCI IV-2a (Brunner et al., 2008)**: BCI IV-2a dataset was collected at Graz University of Technology from 9 subjects using 22 EEG channels. Subjects perform four motor imagery tasks: imagining movements of the left hand, right hand, both feet, and tongue. Each subject completed two sessions on different days, with each session consisting of six runs. Each run contained 48 trials, 12 per class, yielding a total of 288 trials per session (used channels: 22).

- **BCI IV-2b (Leeb et al., 2008)**: BCI IV-2b dataset contains EEG recordings with 3 channels collected at Graz University of Technology from 9 subjects performing motor imagery tasks involving the left and right hands. Each subject participated in five sessions: two screening sessions without feedback and three online feedback sessions. The sessions consisted of multiple runs guided by visual cues (arrows) and auditory signals to direct motor imagery (used channels: 3).

- **BCI IV-1 (Blankertz et al., 2007)**: BCI IV-1 dataset comprises EEG recordings from 7 healthy subjects performing two types of motor imagery tasks: imagining movements of the left or right hand and feet, along with periods of intentional rest (no cue). Continuous EEG was recorded with 59 channels (0.05–200 Hz band, 1000 Hz sampling rate). The experiment included two calibration runs and four evaluation runs, designed for asynchronous BCI classification tasks (used channels: 59).

- **SHU-MI (Ma et al., 2022a)**: SHU-MI dataset comprises 32-channel EEG signals collected from 25 participants performing left- and right-hand motor imagery tasks over five days. It was designed to investigate cross-session variability and supports within-session classification, cross-session classification, and cross-session adaptation (CSA) experiments (used channels: 32).

- **Inria BCI Challenge (Margaux et al., 2012)**: Inria BCI Challenge dataset contains 56-channel EEG recordings from 26 healthy participants engaged in a P300-speller task, where subjects spelled words based on visually flashed stimuli. The experiment included both fast (4 flashes) and slow (8 flashes) flashing conditions. The main objective was to predict error occurrence using only post-feedback EEG signals. Each participant completed five copy-spelling sessions (used channels: 56).

- **Target Versus Non Target (Korczowski et al., 2019)**: Target Versus Non Target dataset was collected using 32 channels in 2015 at GIPSA-lab from 43 participants playing the P300-based BCI game, Brain Invaders. Stimuli consisted of 36 symbols, including one target and 35 non-targets. Three flash duration conditions were used: 50 ms, 80 ms, and 110 ms. EEG signals were recorded while participants interacted with the game under these varying stimulus timings (used channels: 32).

- **MoBI (He et al., 2018)**: MoBI dataset consists of EEG recordings collected from 8 healthy participants walking on a treadmill. Subjects completed repeated sessions under three conditions: standing still, walking, and walking with closed-loop BCI control. Synchronized data include EEG, EOG, and joint angles (hip, knee, ankle). The dataset supports research on neural activity during movement, BCI-mediated motor control, and decoder optimization. In this study, we exclusively utilize the EEG modality, which consists of 60 channels (used channels: 60).

- **BCI2020-3 (Jeong et al., 2022)**: BCI2020-3 dataset contains EEG recordings from 15 participants for multi-class imagined speech classification, and is publicly available via the Open Science Framework. The participants imagined uttering five commonly used words or phrases ('hello', 'help me', 'stop', 'thank you', 'yes') across 70 trials per class, with 60 used for training and 10 for validation, totaling 350 trials. Each trial consisted of a 0.8–1.2 second fixation cue followed by a 2 second imagination period, repeated four times, and concluded with a 3 second rest. EEG was recorded with a 64-channel BrainAmp system under the international 10–20 layout, with electrode impendance maintained below 15 k$\Omega$ (used channels: 64).

- **SPIS Resting State Dataset (Torkamani-Azar et al., 2020)**: SPIS dataset is a resting-state EEG dataset recorded from 10 healthy adult participants at Sabanci University. It includes simultaneous EEG and forehead EOG signals collected while participants performed the Sustained Attention to Response Task (SART), a repetitive stimulus–response paradigm requiring continuous attention. In this study, we exclusively utilize the EEG modality, which consists of 64 channels. Each session lasted over 100 minutes and consisted of 12 blocks with more than 2,700 stimuli in total. Prior to the task, participants completed a mental arithmetic task to normalize attention levels, followed by 2.5 minutes of eyes-open and 2.5 minutes of eyes-closed baseline EEG recordings (used channels: 64).

- **Fatigueset (Kalanadhabhatta et al., 2021)**: FatigueSet dataset is a multimodal dataset collected by the University of Massachusetts Amherst and Nokia Bell Labs to study the interaction between mental fatigue, physical exertion, and physiological responses. EEG was recorded from 12 participants using 4 channels across three sessions involving different levels of physical activity (low, moderate, high) and cognitive tasks. Each session consisted of rest, physical activity, and cognitive phases, during which subjective fatigue ratings and task performance were used to quantify fatigue levels (used channels: 4).

- **Stew (Lim et al., 2018)**: Stew dataset includes EEG recordings with 14 channels from 48 healthy adult male participants performing a multitasking paradigm designed to induce varying levels of mental workload. The experiment consisted of a 3-minute resting state phase and an 18-minute multitasking session, with 2.5 minutes of EEG data from each condition used for analysis. EEG was recorded using a Emotiv EPOC wireless headset. After the task, participants reported subjective workload scores on a 9-point scale to categorize workload levels (used channels: 14).

- **Raw EEG Data (Trujillo, 2021)**: Raw EEG Data dataset contains 64-channel EEG recordings collected at Texas State University and is publicly available via the Texas Data Repository. Subjects performed two categorization tasks: an information-integration task and a multidimensional rule-based task. The dataset is used for the analysis of brain activity during categorization and the understanding of cognitive processing mechanisms (used channels: 64).

- **Mental Arithmetic (Zyma et al., 2019)**: Mental Arithmetic dataset was collected by Igor Sikorsky Kyiv Polytechnic Institute. It includes 19-channel EEG recorded at 500 Hz from 36 subjects. Participants performed repeated subtraction tasks, subtracting two-digit numbers from four-digit numbers, and EEG was recorded for 60 seconds during rest and 60 seconds during task states for each participant (used channels: 19).

- **Berlin_dsr (Shin et al., 2018)**: Berlin_dsr dataset was collected at Technische Universität Berlin. It includes 28-channel EEG recorded at 200 Hz from 26 subjects performing discrimination and selection response tasks. Participants responded to O or X stimuli displayed on screen by pressing buttons. Each session was repeated three times, and a total of 180 trials were performed (used channels: 26).

- **Berlin_nback (Shin et al., 2018)**: Berlin_nback dataset was collected at Technische Universität Berlin. It includes 28-channel EEG recorded at 200 Hz from 26 subjects performing n-back tasks to measure working memory load. Participants completed 0-back, 2-back, and 3-back conditions, with each condition repeated across three sessions (used channels: 26).

- **Berlin_wg (Shin et al., 2018)**: Berlin_wg dataset was collected at Technische Universität Berlin. It includes 28-channel EEG recorded at 200 Hz from 26 subjects performing word generation tasks. Participants were asked to generate as many words as possible starting with given letters, alternating with baseline conditions. Each session included 10 word generation and 10 baseline blocks, for a total of 60 trials (used channels: 26).

- **Mumtaz2016 (Mumtaz, 2016)**: Mumtaz2016 dataset was collected at Universiti Teknologi PETRONAS to study treatment response in patients with major depressive disorder. It includes EEG recorded using 19 channels at 256 Hz from 34 patients and 30 healthy controls. EEG was recorded for five minutes each in eyes-open and eyes-closed resting-state conditions. Some sessions also included visual oddball tasks to collect P300 ERP data. (used channels: 19).

- **Grasp and Lift Challenge (Luciw et al., 2014)**: Grasp and Lift Challenge dataset includes EEG, EMG, kinematic, and force data from 12 participants performing object grasping and lifting tasks. Recordings comprise 32-channel EEG, 5-channel EMG, 3D hand and object positions, and measurements of grip force and torque. Across 3,936 trials, object weight and friction conditions varied randomly. Each trial includes annotations for 16 event timings and 18 behavioral markers, supporting research in sensorimotor decoding and action prediction (used channels: 32).

## C    DETAILS OF EXPERIMENTAL SETUP FOR DOWNSTREAM BCI TASKS

### C.1    HYPERPARAMETER SETTINGS FOR NERVE FINE-TUNING

We load the pre-trained weights of NERVE encoder and attach the task-specific head which is composed of multi-layer perceptrons. NERVE is finetuned to downstream datasets using binary cross-entropy loss for binary classification, cross-entropy loss for multi-class classification, and mean squared error loss for regression. We use a single NVIDIA A100-80G GPU for fine-tuning. Detailed hyperparameters for NERVE fine-tuning are presented in Table 8.

### C.2    FINE-TUNING DATASETS DESCRIPTION

- **TUSL (von Weltin et al., 2017)**: TUSL dataset is a subset of the Temple University Hospital (TUH) EEG Corpus, developed by the Neural Engineering Data Consortium (NEDC) at Temple University. It was specifically designed to support machine learning research for differentiating seizure and slowing events in EEG recordings. The dataset includes 75 EEG

Table 8: Hyperparameters for NERVE fine-tuning.

| Hyperparameters | Values |
|---|---|
| Epochs | 20 |
| Batch size | 128 / 256 |
| Dropout | 0.1 |
| Optimizer | AdamW |
| Adam $\beta$ | (0.9, 0.999) |
| Peak learning rate | $1e^{-4}$ |
| Minimal learning rate | $1e^{-5}$ |
| Weight decay | $1e^{-4}$ |
| Learining rate scheduler | Cosine |
| Warmup epochs | 2 |
| Label smoothing (multi-class) | 0.1 |

sessions with 23 channels from 38 patients, with 10-second segment-based annotations for seizure, slowing, and complex background activity (used channels: 20).

- **SEED-V (Liu et al., 2021)**: SEED-V dataset consists of EEG and eye movement recordings collected from 16 subjects at the BCMI Lab of Shanghai Jiao Tong University. EEG was recorded using 62 channels at 1000 Hz while participants watched video clips designed to elicit five emotions: happiness, sadness, fear, disgust, and neutral. Each subject completed three sessions, and each session consisted of 15 trials (used channels: 62).

- **DEAP (Koelstra et al., 2011)**: DEAP dataset was collected at the University of Twente and at Queen Mary University. It includes EEG recorded using 32 channels at 512 Hz and peripheral physiological signals from 32 participants watching 40 one-minute music videos. After each video, participants rated their emotional responses in terms of valence, arousal, dominance, and like/dislike. (used channels: 32)

- **HCI-Tagging Emotion (Soleymani et al., 2011)**: HCI-Tagging Emotion dataset was collected at the Idiap Research Institute. EEG was recorded using 32 channels at 256 Hz along with ECG, GSR, respiration, skin temperature, eye tracking, and facial videos from 27 subjects. During the experiment, participants watched 20 emotional video clips and annotated with high-arousal and low-arousal labels (used channels: 32).

- **High Gamma (Schirrmeister et al., 2017)**: High Gamma dataset contains EEG recordings from 14 healthy participants performing hand and foot movement tasks, as well as rest, using a 73-channel setup. Each participant completed 13 runs consisting of 4-second trials across four classes: left hand, right hand, both feet, and rest. Visual cues in the form of arrows guided the tasks, and recordings were conducted using the BCI2000 system to capture high gamma activity (used channels: 73).

- **TUEV (Harati et al., 2015)**: TUEV dataset was collected using 23 channels at Temple University Hospital by the Neural Engineering Data Consortium (NEDC), this dataset is a subset of the TUH EEG Corpus and includes EEG sessions labeled with a diverse range of events. These include spike and/or sharp wave discharges (SPSW), periodic lateralized epileptiform discharges (PLED), generalized periodic epileptiform discharges (GPED), eye movements (EYEM), artifacts (ARTF), and background activity (BCKG). The dataset was designed to support event-based EEG classification, noise detection, and clinical signal analysis using machine learning techniques (used channels: 23).

- **BCI-NER Challenge (Mattout et al., 2014)**: BCI-NER Challenge dataset was released as part of the IEEE Neural Engineering Conference (NER2015) BCI Challenge. EEG was recorded using 56 channels at 200 Hz from 26 participants performing copy-spelling tasks under fast and slow flashing conditions to detect error-related potentials (ErrP) (used channels: 55).

- **SEED-VIG (Zheng & Lu, 2017)**: SEED-VIG dataset consists of EEG recordings collected from 23 participants at the BCMI Lab of Shanghai Jiao Tong University using a virtual driving system. EEG was recorded using 17 channels at 200 Hz from participants driving in a simulated driving environment. Vigilance levels were annotated using the PERCLOS metric measured with SMI eye-tracking glasses (used channels: 17).

C.3 DATA PREPROCESSING AND SPLITTING STRATEGIES

**Preprocessing.** Channel selection was designed to maximize spatial resolution and enhance robustness to noise by including as many electrodes as possible. Prior studies have shown that multi-channel configurations are more resilient when some electrodes are corrupted or subject to shifts, thereby providing improved stability under noisy recording conditions (Kim & Im, 2021; Temko et al., 2011). Moreover, empirical evidence consistently indicates that models trained with a larger number of channels outperform those restricted to fewer electrodes (Kim & Im, 2021; Tong et al., 2023; Shan et al., 2012; Temko et al., 2011). Following this evidence, we adopt a comprehensive strategy and include most electrodes available in the pretraining data to achieve broad channel coverage, resulting in a 128-channel configuration. Subsequently, in the presence of missing values, we replace them with zeros, allowing the model to treat such regions as non-signal or to leverage them as an explicit missing-value mask (Kostas et al., 2021). For filtering, we applied a band-pass filter of 0.3 and 75 Hz, following Wang et al. (2025b). The lower cutoff removes slow drifts caused by electrode movement or perspiration, while the upper cutoff suppresses high-frequency muscle and equipment noise, retraining the physiologically relevent EEG spectrum. We adopted a notch filtering of 50 Hz to eliminate power-line interference (Jiang et al., 2024; Ding et al., 2024). The signals were resampled to 200 Hz to reduce computational cost while retaining physiologically relevant frequency components (Singh & Krishnan, 2023; Huang et al., 2022). Finally, we adopted z-score normalization to mitigate inter- and intra-subject variability (Cui et al., 2024; Apicella et al., 2023; Xiao et al., 2025).

**Splitting.** To ensure comparability with prior work, we adopted a subject-wise split with a ratio of 0.8, 0.1, and 0.1 for training, validation, and test sets, respectively. For SEED-V, we applied a trial-based split of 5 trials per subject (Jiang et al., 2024; Wang et al., 2025b). For datasets that already provide an official split, we directly followed the provided configuration. In cases where the dataset only offered training and test sets without a validation set, we further divided the training set into training and validation subsets using a ratio of 0.8 and 0.2.

Table 9: Downstream datasets splitting method.

| Datasets | # Subjects | Train | Validation | Test |
|---|---|---|---|---|
| TUSL | 38 | 1-30 | 31-34 | 35-38 |
| SEED-V | 16 | | 5 trials per subject | |
| DEAP | 32 | 1-25 | 26-29 | 30-32 |
| HCI-Tagging Emotion | 27 | 1-22 | 23-24 | 25-27 |
| High-Gamma | 14 | 1-11 | 12 | 13-14 |
| TUEV | 38 | official split | - | official split |
| BCI-NER Challenge | 26 | official split | - | official split |
| SEED-VIG | 21 | 1-15 | 16-19 | 20-23 |

C.4 BASELINES

Here, We introduce the baselines for performance evaluation, which consist of non-foundation model and foundation model baselines.

**Non-foundation Model Baselines:**

- **EEGNet** (Lawhern et al., 2018) employs a lightweight convolutional architecture that leverages depthwise and separable convolutions to efficiently model EEG signals.

- **EEGConformer** (Song et al., 2022) integrates convolutional layers for capturing localized temporal patterns and self-attention modules to model long-range temporal dependencies within EEG data.

- **ContraWR** (Yang et al., 2021) processes biosignals by converting them into multi-channel spectrograms, which are then analyzed using a ResNet-based 2D CNN to extract discriminative features.

- **CNN-Transformer** (Peh et al., 2022) combines convolutional operations to extract local representations with transformer blocks to capture global contextual relationships.

Table 10: The evaluation results on HCI-Tagging Emotion and SEED-V

| Methods | HCI-Tagging Emotion, 2-class | | | SEED-V, 5-class | | |
| | Balanced Accuracy | AUC-PR | AUROC | Balanced Accuracy | Cohen's Kappa | Weighted F1 |
|---|---|---|---|---|---|---|
| EEGNet | 0.5518 ± 0.0052 | 0.4065 ± 0.0299 | 0.5746 ± 0.0140 | 0.3046 ± 0.0083 | 0.1282 ± 0.0096 | 0.3064 ± 0.0079 |
| EEGConformer | 0.5289 ± 0.0489 | 0.4566 ± 0.0293 | 0.5567 ± 0.0348 | 0.3370 ± 0.0098 | 0.1758 ± 0.0136 | 0.3481 ± 0.0105 |
| ContraWR | 0.5793 ± 0.0343 | 0.4999 ± 0.0701 | 0.6282 ± 0.0553 | 0.3685 ± 0.0121 | 0.2115 ± 0.0178 | 0.3723 ± 0.0179 |
| CNN-Transformer | 0.5808 ± 0.0133 | 0.4714 ± 0.0374 | 0.5940 ± 0.0240 | 0.3491 ± 0.0351 | 0.1912 ± 0.0417 | 0.3538 ± 0.0412 |
| FFCL | 0.5782 ± 0.0480 | **0.5200** ± 0.0616 | 0.6099 ± 0.0542 | 0.3731 ± 0.0137 | 0.2185 ± 0.0180 | 0.3771 ± 0.0190 |
| ST-Transformer | 0.5050 ± 0.0111 | 0.3482 ± 0.0184 | 0.4981 ± 0.0173 | 0.2994 ± 0.0237 | 0.1205 ± 0.0307 | 0.2886 ± 0.0362 |
| BIOT | 0.5667 ± 0.0253 | 0.4455 ± 0.0310 | 0.6130 ± 0.0244 | 0.3201 ± 0.0073 | 0.1541 ± 0.0093 | 0.3283 ± 0.0073 |
| LaBraM | 0.5813 ± 0.0185 | 0.4299 ± 0.0230 | 0.6202 ± 0.0231 | **0.4009** ± 0.0087 | **0.2518** ± 0.0094 | **0.4055** ± 0.0071 |
| CBraMod | 0.5675 ± 0.0403 | 0.4802 ± 0.0530 | 0.6325 ± 0.0601 | 0.3642 ± 0.0054 | 0.2123 ± 0.0066 | 0.3756 ± 0.0052 |
| NeuroLM | 0.5000 ± 0.0000 | 0.3549 ± 0.0000 | 0.5000 ± 0.0000 | 0.3494 ± 0.0031 | 0.1915 ± 0.0039 | 0.3579 ± 0.0043 |
| NERVE | **0.5943** ± 0.0177 | 0.5103 ± 0.0381 | **0.6691** ± 0.0238 | 0.3788 ± 0.0031 | 0.2305 ± 0.0034 | 0.3908 ± 0.0028 |

Table 11: The evaluation results on TUEV and SEED-VIG

| Methods | TUEV, 6-class | | | SEED-VIG, regression | | |
| | Balanced Accuracy | Cohen's Kappa | Weighted F1 | Pearson's Correlation | R2 Score | RMSE↓ |
|---|---|---|---|---|---|---|
| EEGNet | 0.4534 ± 0.0135 | 0.3315 ± 0.0367 | 0.6492 ± 0.0217 | 0.2397 ± 0.0633 | -0.2193 ± 0.1778 | 0.3440 ± 0.0253 |
| EEGConformer | 0.3607 ± 0.0168 | 0.2678 ± 0.0449 | 0.6347 ± 0.0228 | 0.3001 ± 0.1186 | -0.3539 ± 0.1765 | 0.3627 ± 0.0236 |
| ContraWR | 0.4178 ± 0.0412 | 0.3814 ± 0.0856 | 0.6859 ± 0.0419 | 0.4291 ± 0.1471 | 0.1156 ± 0.1181 | 0.2931 ± 0.0197 |
| CNN-Transformer | 0.3670 ± 0.0138 | 0.3150 ± 0.0383 | 0.6535 ± 0.0212 | 0.1805 ± 0.1399 | -0.0803 ± 0.0656 | 0.3245 ± 0.0098 |
| FFCL | 0.4012 ± 0.0247 | 0.3255 ± 0.0263 | 0.6555 ± 0.0138 | 0.3836 ± 0.0954 | 0.1228 ± 0.0832 | 0.2923 ± 0.0136 |
| ST-Transformer | 0.4432 ± 0.0295 | 0.4356 ± 0.0313 | 0.7130 ± 0.0156 | 0.0761 ± 0.1890 | -0.3170 ± 0.3244 | 0.3560 ± 0.0422 |
| BIOT | 0.4792 ± 0.0130 | 0.4529 ± 0.0214 | 0.7195 ± 0.0111 | 0.4195 ± 0.0324 | -0.2334 ± 0.1261 | 0.3465 ± 0.0176 |
| LaBraM | 0.5519 ± 0.0206 | **0.5311** ± 0.0295 | **0.7605** ± 0.0144 | **0.5498** ± 0.0099 | 0.1515 ± 0.0289 | 0.2875 ± 0.0048 |
| CBraMod | 0.5098 ± 0.0130 | 0.4208 ± 0.0233 | 0.7013 ± 0.0134 | 0.5089 ± 0.0335 | 0.0113 ± 0.1657 | 0.3096 ± 0.0260 |
| NeuroLM | 0.4653 ± 0.0091 | 0.3940 ± 0.0179 | 0.6867 ± 0.0116 | - | - | - |
| NERVE | **0.5595** ± 0.0106 | 0.4810 ± 0.0146 | 0.7249 ± 0.0095 | 0.4158 ± 0.0177 | **0.1591** ± 0.0142 | **0.2865** ± 0.0024 |

- **FFCL** (Li et al., 2022) employs a parallel architecture where convolutional layers extract spatial features and LSTM modules model temporal dynamics.
- **ST-Transformer** (Song et al., 2021) utilizes a transformer-based design that simultaneously learns spatial and temporal dependencies within EEG signals through attention mechanisms.

**Foundation Model Baselines:**

- **BIOT** (Yang et al., 2023) introduces a unified Transformer-based architecture for diverse biosignals with mismatched channels and variable lengths that tokenizes different biosignals into unified sentence structure. Since BIOT supports up to 18 predefined channel embeddings, we omit channel embeddings when processing EEG signals with more than 18 channels or unmatched channel configurations.
- **LaBraM** (Jiang et al., 2024) is a large EEG foundation model, which learns EEG representations by predicting discrete EEG codes of neural tokenizer from masked EEG signal.
- **CBraMod** (Wang et al., 2025b) is a foundation model for EEG that captures diverse spatial dependencies using assymetric positional encoding and a criss-cross attention mechanism within a transformer architecture.
- **NeuroLM** (Jiang et al., 2025b) is a multi-task foundation model that combines text-aligned EEG tokenizer and large language model. NeuroLM can perform diverse EEG tasks within a single model through instruction tuning.

# D  MORE EXPERIMENTAL RESULTS ON OTHER DATASETS

In this section, we provide experimental results on four datasets omitted from the main text due to space constraints. Table 10 presents the performance on HCI-Tagging Emotion and SEED-V for emotion recognition. On HCI-Tagging Emotion, NERVE outperforms all foundation model baselines, although non-foundation models such as FFCL and ContraWR exhibit competitive performance on this dataset. NERVE shows the second best performance on SEED-V, following LaBraM, while outperforming all non-foundation model baselines. Table 11 presents the evaluation results

on TUEV and SEED-VIG, corresponding to event type classification and vigilance prediction tasks, respectively. As SEED-VIG is a regression task, NeuroLM is not evaluated because its instruction tuning framework does not support regression tasks. On TUEV, NERVE achieves the highest accuracy and ranks second in Cohen's Kappa and weighted F1, following LaBraM, while outperforming all non-foundation model baselines. On SEED-VIG, NERVE exhibited strong performance in terms of R2 score and RMSE, while showing a relatively lower correlation compared to other models.

## E    VISUALIZATION OF NOISE-ROBUST NEURAL TOKENIZER TRAINING

We visualize training results of noise-robust neural tokenizer. As depected in Figure 7, given a noise-augmented signal, tokenizer could successfully capture the trend in the original EEG signal. Although Gaussian noise has a marginal visual effect on the signal, it leads to significant performance degradation due to its influence across all frequency bands (see Sections 4 and G.1). As observed in Figure 6, the steady decrease in both temporal reconstruction loss and normalized frequency loss may contribute to NERVE's robustness against Gaussian noise as well as other frequency-specific noises.

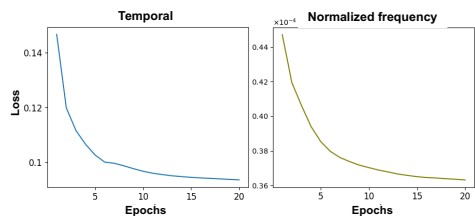

Figure 6: Temporal reconstruction and normalized frequency prediction loss curves.

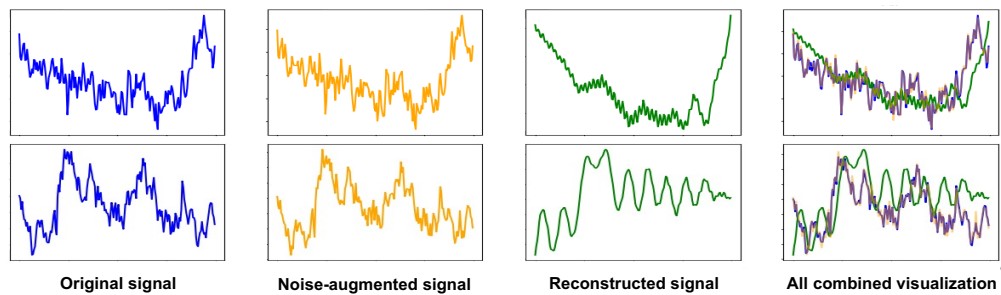

| Original signal | Noise-augmented signal | Reconstructed signal | All combined visualization |

Figure 7: Visualization of noise-robust neural quantizer pre-training results

## F    CONNECTION BETWEEN EPA ATTENTION AND EXISTING STUDIES

In this section, we analyze the formulation of the EPA attention and investigate the connection between the EPA attention and attention algorithms proposed in previous studies. The EPA transformer is composed of two sequential sub-blocks, the temporal and channel blocks, where temporal attention and EPA attention mechanisms are applied. EPA attention defines a position router $P \in \mathbb{R}^{R \times N \times d}$ as a learnable parameter, representing groups of electrodes organized by cortical regions. Then, EPA attention calculates attention score by multiplying query and key projections of EEG channel embeddings with query and key projections of the position router, as formulated in Equation 3.

**Connection to CBraMod.** EPA transformer constitutes as a sequential formulation of generalized criss-cross transformer proposed in CBraMod(Wang et al., 2025b). Criss-cross attention is defined as parallel-head formulation of standard temporal and spatial attentions. To derive the connection between ours and criss-cross attention, we denote $S_j = (Q_H M_j K_H^T)/\sqrt{d}$ and $M_j = Q_P^T K_P = (PW_Q)^T (PW_K) \in \mathbb{R}^{d \times d}$. If $M_j \equiv I$, EPA attention reduces to the canonical spatial attention, which is the spatial attention found in the cross-cross attention block. Thus, the EPA transformer represents a sequential conversion of the generalized criss-cross transformer with arbitrary $M_j$, and criss-cross transformer can be regarded as a special case of EPA transformer. Furthermore, given that $PW_Q \in \mathbb{R}^{d \times R}$ with smaller $R$ than the number of EEG channels (in our work

$R = 6$), the rank of $M_j$ is inherently bounded by $R$. Since $rank(M_j) \leq R$, so EPA attention constrains the channel-wise score matrix $S_j$ to lie in a low-dimensional, router-defined subspace. This low-rank and topology-aligned regularization encourages robust electrode–region–electrode dependencies while suppressing spurious correlations among channels. When deriving the gradient of the loss function $L$ with respect to the attention score $S_j$ (see the Equation 13), $M_j^T$ acts as a multiplier to the gradient of the standard channel attention. Given that $rank(M_j) \leq R$, this operation constrains the gradient flow, imposing an implicit regularization effect on the learned parameters.

$$\frac{\partial L}{\partial S_j} = \frac{\partial L}{\partial Q_H}\left(\frac{\partial S_j}{\partial Q_H}\right)^{-1} = \frac{\partial L}{\partial Q_H}(M_j K_H^T)^{-1} = \frac{\partial L}{\partial Q_H}(K_H^T)^{-1}(M_j^T)^{-1} \tag{13}$$

**Connection to Linformer.** Since the position router and projection layers are basically linear layers, this formulation shares similarities with Linformer (Wang et al., 2020) in utilizing linear layers. However, their objectives and detailed mechanisms are fundamentally distinct. The primary objective of Linformer is to utilize linear layers to reduce the computational complexity of the self-attention mechanism. In contrast, EPA attention defines the position router as learnable parameters to explicitly model the interaction between EEG channels and their spatial positions. As a result, although EPA attention structurally resembles an attention operation with linear layers, its fundamental purpose is spatial modeling rather than complexity reduction. In Linformer attention, linear layers are applied to the key and value embeddings to reduce dimensionality. In contrast, EPA attention applies linear transformations (via the position router) to the query and key embeddings. Due to this structural difference, Linformer operates with $O(C)$ complexity, whereas EPA attention maintains an $O(C^2)$ complexity.

# G ROBUSTNESS ANALYSIS

## G.1 ROBUSTNESS TO NOISE

We assessed the robustness of existing foundation models against various types of noise inherent in EEG signal acquisition. To achieve this, we meticulously designed six distinct noise scenarios: four fundamental types including Electromyogram (EMG), Electrooculogram (EOG), environmental noise (from surrounding electronics), and Gaussian noise; and two additional scenarios generated by controlling signal-to-noise ratio (SNR) of the input signals to 5 dB and 10dB. These conditions were specifically created to reflect real-world artifacts, as the high sensitivity of scalp electrodes makes EEG signals inherently vulnerable to artifacts from muscular and ocular activities, alongside ambient environmental interference. Furthermore, because SNR serves as the standard metric for quantifying signal quality (Braun et al., 2017; Galeotti & Scully, 2018), addressing low SNR, which represents the most critical acquisition-related challenge of our work, is essential. These frequently encountered noise sources significantly degrade both signal quality and downstream performance in real-world EEG acquisition settings.

Muscle activity generates EMG noise, prominent in the beta (13–30 Hz) and gamma (> 30 Hz) bands. EOG artifacts are commonly observed in the low frequency bands, mainly between 0.5

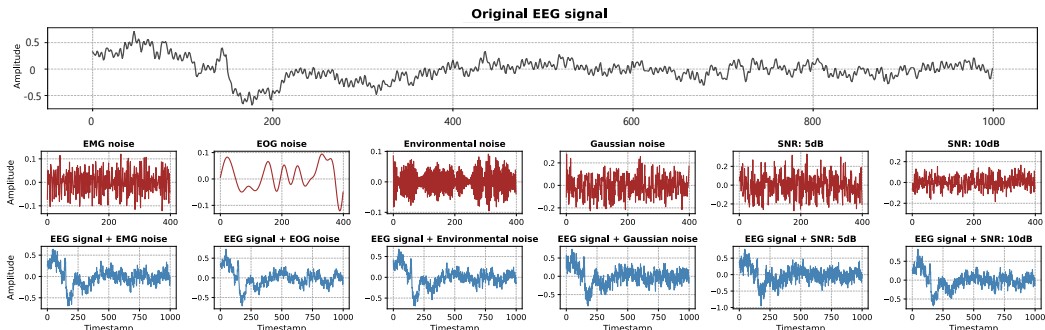

Figure 8: Case visualization of noise scenarios in noise robustness analysis

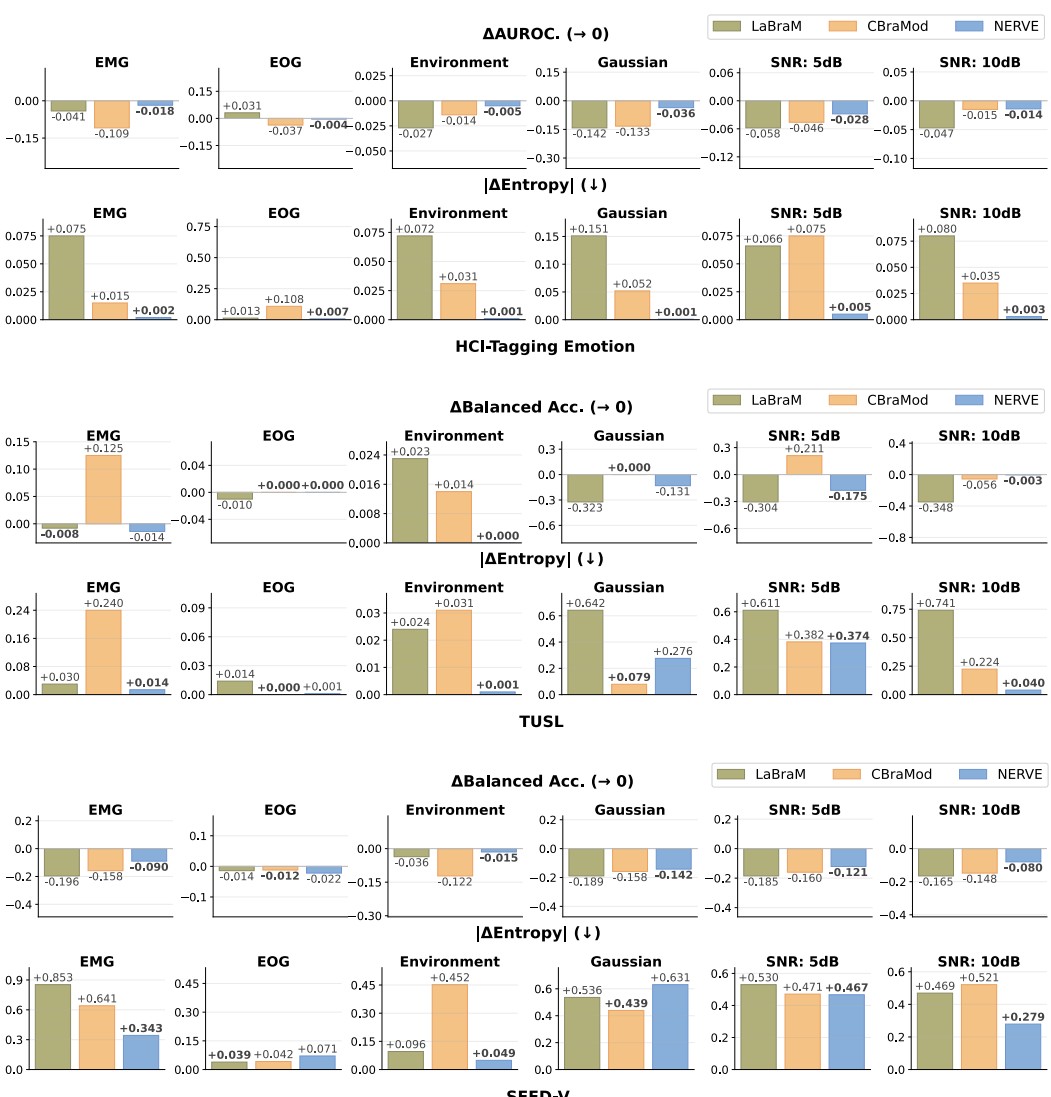

Figure 9: Noise robustness analysis results

and 12 Hz. Environmental noise (e.g., power line interference) contaminates the 50–60 Hz band. To simulate these, we applied the Discrete Fourier Transform (DFT) to the EEG signals and then injected Gaussian noise with a standard deviation of 10 into specific frequency bands: EOG (5–12 Hz), EMG (50–70 Hz), and Environmental (50–60 Hz). Gaussian noise is simulated by adding zero-mean Gaussian noise with a standard deviation of 0.5 to the raw EEG signals in the time domain. In SNR scenarios, we control the signal-to-noise ratio (SNR) of the input signal, which is defined as $SNR_{DB} = 10log_{10}(\frac{P_{signal}}{P_{noise}})$, where the power $P = lim_{T \to \infty} \frac{1}{T} \int_{-\frac{T}{2}}^{\frac{T}{2}} x^2(t)dt$. The example of corrupted signal with synthetic noises is presented in Figure 8.

Figure 9 presents the analysis results on the HCI-Tagging Emotion, TUSL, and SEED-V datasets. We use AUROC as the primary performance metric for HCI-Tagging Emotion, as it better captures performance than balanced accuracy in class-imbalanced datasets. For predictive entropy, we visualize the absolute change from the clean signal to assess the consistency of predictions under noise-injected inputs. NERVE exhibited the lowest performance drop and the smallest entropy change across all noise conditions for HCI-Tagging Emotion, and in the majority of cases for TUSL and SEED-V. Meanwhile, LaBraM and CBraMod often exhibited higher accuracy on TUSL when noise was applied to the raw EEG signal. We argue that EEG foundation models take clinically meaningless signals as useful evidence and utilize them for prediction. This misinterpretation of

noise, which we term noise hallucination, is frequent across datasets and poses a critical challenge for deploying EEG foundation models. In real-world scenarios where noisy signals are pervasive, noise hallucination may undermine both the validity and reliability of their predictions. Conversely, NERVE did not exhibit observable noise hallucination in our experiments, consistently producing robust and clinically meaningful predictions under noisy conditions.

## G.2 ROBUSTNESS TO VARIABILITY.

The analysis on robustness to variability was conducted on HCI-Tagging Emotion, TUSL, High-Gamma, and SEED-V. The remaining four datasets were unsuitable for this evaluation for the following reasons: TUEV lacks subject identifiers; SEED-VIG is a regression dataset, making it impossible to calculate the logits; and on DEAP and BCI-NER-Challenge, the LaBraM baseline exhibited degenerate predictions across certain seeds, which made reliable variability measurement impossible.

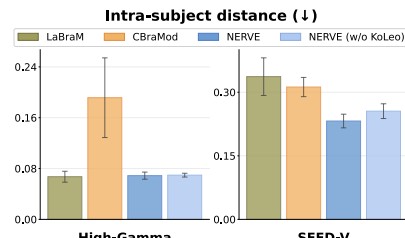

Figure 10: Anaysis results on robustness to intra-subject variability

Figure 10 illustrates the average intra-subject distances of test subjects on the High-Gamma and SEED-V datasets for different EEG foundation models. We measure the average intra-subject distance for each foundation model fine-tuned on the respective dataset. A lower average intra-subject distance indicates that the model can consistently distinguish subtly varying patterns arising from diverse physiological states and environmental factors during EEG signal acquisition from each subject, suggesting strong robustness to intra-subject variability. On both datasets, NERVE exhibited the lowest average intra-subject distance and the smallest variance across subjects, demonstrating superior robustness to intra-subject variability. Compared to NERVE without KoLeo loss, the full model exhibits a lower intra-subject distance, highlighting the effectiveness of embedding uniformity through KoLeo loss in achieving robustness to intra-subject variability. On the one hand, LaBraM and NERVE show similar intra-subject distances on the High-Gamma dataset. This is attributed to the small number of test subjects on High-Gamma, which is two. The small number of subjects accidently suggests comparable robustness to intra-subject variability, but NERVE exhibits superior robustness to intra-subject variability on other datasets over three test subjects.

Figure 11 presents the inter-subject distance for each class on the BCI-NER Challenge and SEED-V datasets. We evaluate the inter-subject distance on each dataset using the pre-trained weights of the foundation models without fine-tuning. NERVE consistently exhibits the highest inter-subject distance across all classes on both datasets, indicating strong robustness to inter-subject variability. Similarly, NERVE pre-trained without KoLeo loss exhibits reduced inter-subject distance, confirming that embedding uniformity enhances robustness to inter-subject variability even without subject-identifiable information.

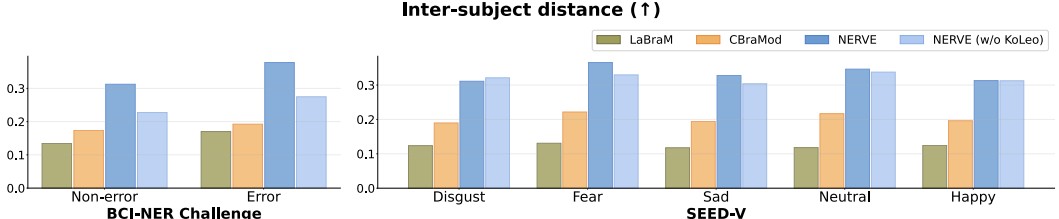

Figure 11: Analysis results on robustness to inter-subject variability

## H INTERPRETABILITY ANALYSIS

### H.1 CHANNEL EMBEDDING VISUALIZATION

We visualized the inter-channel cosine similarity heatmap of pre-trained NERVE embeddings using the BCIC2020-3 dataset from our pre-training corpus. To compute these channel embeddings, the

output of the final EPA transformer layer was extracted, and the embedding for each channel was defined by taking the mean of these output representations across the patch dimension. As shown in Figure 12, We observe high similarity among specific groups of electrodes, forming a distinct block-wise similarity structure. This observation underscores the necessity of region-wise channel modeling for EEG signals and supports the validity of EPA attention.

## H.2 CLASS TOPOGRAPHY VISUALIZATION

We conducted a visualization analysis to compare the topography of the raw sample with class activation topography of NERVE on the High-Gamma dataset. To visualize raw EEG topography, we computed the maximum on the signal for each channel and drew the results on a topographic map. o visualize the class activation topography of NERVE, Grad-GAM was utilized to compute the contribution of each channel to the resulting prediction, which was then formulated as a class activation topographic map (Selvaraju et al., 2017). Figure 13 displays the class-wise topographies of raw EEG signals and NERVE embeddings. We observed that electrodes associated with the left hand exhibit patterns relatively symmetric to those of the right hand, while both cortical regions are activated for the 'both feet' class. In contrast, minimal activation is observed during the resting state. Although NERVE's class activation topography is not identical to the raw EEG signals, its representations capture broadly similar spatial patterns, demonstrating distinct differences across classes.

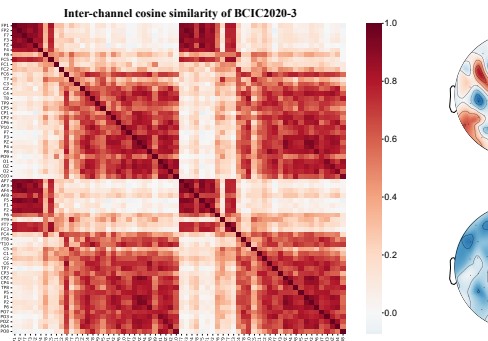

Figure 12: Inter-channel cosine similarity heatmap on BCIC2020-3

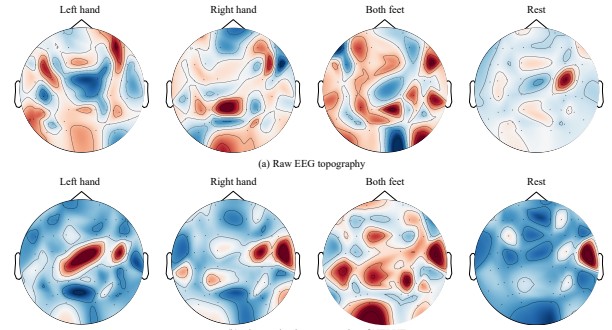

Figure 13: Topography visualization on High-Gamma (motor imagery classification)

# I MORE ABLATION STUDIES

## I.1 ABLATION STUDY ON NOISE-ROBUST NEURAL TOKENIZER TRAINING

We evaluated the impact of noise-robust training for the neural tokenizer on downstream BCI tasks and its robustness to noise. To validate this, we trained the standard neural tokenizer as follows: we removed the Gaussian noise augmentation and directly fed the original EEG signal. The neural tokenizer was then trained to reconstruct the EEG sample in the temporal domain and full spectrum, identical to the process in denoising temporal-spectral prediction without noise augmentation. Subsequently, the NERVE model was trained to predict the codebook indices of the standard neural tokenizer for masked EEG signals, which we term as NERVE (standard).

Table 12 shows the comparative evaluation results between NERVE and NERVE (standard) on the SEED-V, TUSL, DEAP, and HCI-Tagging Emotion datasets. NERVE (standard) exhibits the degraded performance on these datasets, which indicates pre-training with noise-robust neural tokenizer enhances the generalization performance to various BCI tasks. Our noise-robust neural tokenizer resembles a denoising autoencoder (Vincent et al., 2008). The denoising framework serves as a training criterion for learning to extract useful information that constitute high-level semantic information (Vincent et al., 2010). In particular, when Gaussian noise is added, the denoising autoencoder training criterion is equivalent to score matching objective, which indicates that optimizing this criterion approximates the gradient of the log density Vincent (2011). Thus, our noise-

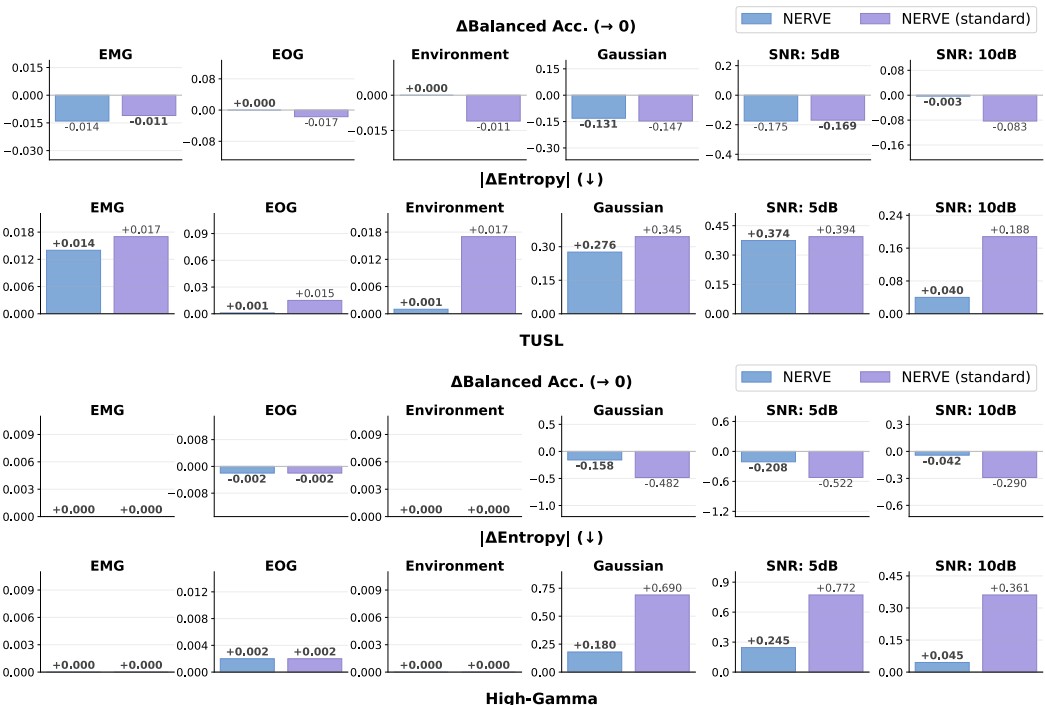

Figure 14: Noise robustness analysis results of NERVE trained with standard neural tokenizer

robust training can be theoretically regarded as learning the useful representations that effectively approximate the data manifold.

We also evaluated noise robustness of NERVE (standard) on various noise conditions. Following the same setup of robustness analysis in Section G.1, we computed the accuracy degradation and the absolute change of predictive entropy from the clean signal. Figure 14 presents the analysis results on TUSL and High-Gamma. NERVE trained without noise augmentation exhibited greater accuracy degradation and more significantly perturbed predictive entropy than NERVE utilizing a noise-robust neural tokenizer. This ablation study underscores that noise-robust tokenizer training is essential for achieving both generalizability and robustness.

### I.2 REPRESENTATION ANALYSIS

To illustrate the behavior of NERVE on various noise scenarios, we provide a representation analysis in Table 13. We computed the Centered Kernel Alignment (CKA) similarity (Kornblith et al., 2019) between embeddings of raw EEG signals and signals corrupted with synthetic noises for NERVE and NERVE (standard). We can find the following two observations: First, NERVE demonstrates higher CKA similarity between raw and corrupted EEG signals in most noise conditions on High-Gamma, TUSL, BCI-NER Challenge, and SEED-VIG. This result indicates that NERVE produces stable representations in the presence of noise artifacts, which accounts for the superior robustness reported in the noise robustness analysis. Through noise augmentation and denoising temporal-

Table 12: The ablation results on noise-robust neural tokenizer training

| Methods | SEED-V, 5-class | | | TUSL, 3-class | | |
| --- | --- | --- | --- | --- | --- | --- |
| | Balanced Accuracy | Cohen's Kappa | Weighted F1 | Balanced Accuracy | Cohen's Kappa | Weighted F1 |
| NERVE | **0.3788** ± 0.0031 | **0.2305** ± 0.0034 | **0.3908** ± 0.0028 | **0.7000** ± 0.0282 | **0.5327** ± 0.0411 | **0.6507** ± 0.0329 |
| NERVE (standard) | 0.3764 ± 0.0014 | 0.2291 ± 0.0016 | 0.3096 ± 0.0012 | 0.6050 ± 0.0452 | 0.3748 ± 0.0736 | 0.5742 ± 0.0320 |

| Methods | DEAP, 2-class | | | HCI-Tagging Emotion, 2-class | | |
| --- | --- | --- | --- | --- | --- | --- |
| | Balanced Accuracy | AUC-PR | AUROC | Balanced Accuracy | AUC-PR | AUROC |
| NERVE | **0.5167** ± 0.0131 | **0.6533** ± 0.0156 | 0.5406 ± 0.0267 | **0.5943** ± 0.0177 | **0.5103** ± 0.0381 | **0.6691** ± 0.0238 |
| NERVE (standard) | 0.5140 ± 0.0124 | 0.6510 ± 0.0202 | **0.5442** ± 0.0278 | 0.5912 ± 0.0298 | 0.4641 ± 0.0519 | 0.6491 ± 0.0377 |

spectral prediction, the noise-robust neural tokenizer learns EEG codebooks that are resilient to noise, thereby providing a prediction target containing high-level semantic information for NERVE pre-training. Second, both models exhibit low CKA similarity between raw and corrupted samples under Gaussian noise and low SNR scenarios. This indicates that unstructured noises affecting the entire spectrum highly perturb the output representations, suggesting these noises pose a more significant challenge for regression tasks.

Table 13: Analysis of CKA similarity between embeddings of raw and noise-corrupted EEG signals

| Methods | EOG | EMG | Environment | Gaussian | SNR: 5dB | SNR: 10dB |
|---|---|---|---|---|---|---|
| **High-Gamma, 4-class** | | | | | | |
| NERVE | **0.9986** ± 0.0002 | **0.9996** ± 0.0001 | **0.9999** ± 0.0000 | **0.7052** ± 0.0051 | **0.6536** ± 0.0162 | **0.9069** ± 0.0038 |
| NERVE (standard) | 0.9941 ± 0.0004 | 0.9989 ± 0.0001 | **0.9999** ± 0.0001 | 0.1395 ± 0.0019 | 0.1323 ± 0.0048 | 0.2416 ± 0.0052 |
| **TUSL, 3-class** | | | | | | |
| NERVE | **0.9980** ± 0.0008 | **0.9987** ± 0.0002 | **0.9998** ± 0.0000 | 0.7788 ± 0.0125 | **0.8644** ± 0.0089 | **0.9558** ± 0.0061 |
| NERVE (standard) | 0.9956 ± 0.0012 | 0.9967 ± 0.0006 | 0.9993 ± 0.0002 | **0.8740** ± 0.0145 | 0.8362 ± 0.0227 | 0.9385 ± 0.0123 |
| **BCI-NER Challenge, 2-class** | | | | | | |
| NERVE | **0.9995** ± 0.0000 | 0.9998 ± 0.0001 | **1.0000** ± 0.0000 | **0.8957** ± 0.0005 | **0.8632** ± 0.0006 | **0.9590** ± 0.0001 |
| NERVE (standard) | **0.9995** ± 0.0000 | **0.9999** ± 0.0000 | **1.0000** ± 0.0000 | 0.8283 ± 0.0003 | 0.8077 ± 0.0002 | 0.8860 ± 0.0010 |
| **SEED-VIG, regression** | | | | | | |
| NERVE | **0.9953** ± 0.0002 | 0.9582 ± 0.0010 | 0.9921 ± 0.0003 | **0.3225** ± 0.0019 | **0.3014** ± 0.0027 | **0.4725** ± 0.0043 |
| NERVE (standard) | 0.9944 ± 0.0002 | **0.9854** ± 0.0006 | **0.9986** ± 0.0001 | 0.2738 ± 0.0094 | 0.2687 ± 0.0083 | 0.3403 ± 0.0068 |

## I.3 ABLATION STUDY ON VARIABILITY-ROBUST PRE-TRAINING

We evaluated the impact of variability-robust learning in NERVE pre-training on downstream BCI tasks. Since variability robustness is encouraged via the KoLeo loss by promoting sample separability, we compared NERVE with its variant pre-trained without the KoLeo loss, which we term as NERVE (w/o KoLeo). Table 14 presents the evaluation results on the SEED-V, TUSL, DEAP and HCI-Tagging Emotion. NERVE achieved superior performance across all datasets on SEED-V, TUSL, DEAP, and HCI-Tagging Emotion, suggesting that a uniformly distributed embedding space enhances downstream BCI task performance. However, as shown in Table 15, the effect of variability-robust pre-training was marginal on the High-Gamma dataset. We attribute this observation to the limited number of test subjects (only two), which indicates that robustness to intra- and inter-subject variabilities is less critical in this specific experimental setting. For SEED-VIG, which is a regression dataset, NERVE trained without KoLeo loss outperformed the standard NERVE. This suggests that enhancing the uniformity of EEG embeddings does not necessarily improves downstream performance for regression task.

Table 14: The ablation results on variability-robust pre-training (effective cases)

| Methods | Balanced Accuracy | Cohen's Kappa | Weighted F1 | Balanced Accuracy | Cohen's Kappa | Weighted F1 |
|---|---|---|---|---|---|---|
| | **SEED-V, 5-class** | | | **TUSL, 3-class** | | |
| NERVE | **0.3788** ± 0.0031 | **0.2305** ± 0.0034 | **0.3908** ± 0.0028 | **0.7000** ± 0.0282 | **0.5327** ± 0.0411 | **0.6507** ± 0.0329 |
| NERVE (w/o KoLeo) | 0.3764 ± 0.0011 | 0.2280 ± 0.0021 | 0.3894 ± 0.0020 | 0.6717 ± 0.0625 | 0.4936 ± 0.0952 | 0.6378 ± 0.0794 |
| | **DEAP, 2-class** | | | **HCI-Tagging Emotion, 2-class** | | |
| | Balanced Accuracy | AUC-PR | AUROC | Balanced Accuracy | AUC-PR | AUROC |
| NERVE | **0.5167** ± 0.0131 | **0.6533** ± 0.0156 | **0.5406** ± 0.0267 | **0.5943** ± 0.0177 | **0.5103** ± 0.0381 | **0.6691** ± 0.0238 |
| NERVE (w/o KoLeo) | 0.5001 ± 0.0075 | 0.6340 ± 0.0142 | 0.5121 ± 0.0217 | 0.5664 ± 0.0414 | 0.4202 ± 0.0033 | 0.6041 ± 0.0343 |

Table 15: The ablation results on variability-robust pre-training (ineffective cases)

| Methods | Balanced Accuracy | Cohen's Kappa | Weighted F1 | Pearson's Correlation | R2 Score | RMSE↓ |
|---|---|---|---|---|---|---|
| | **High-Gamma, 4-class** | | | **SEED-VIG, regression** | | |
| NERVE | **0.9906** ± 0.0011 | **0.9875** ± 0.0015 | **0.9906** ± 0.0011 | 0.4158 ± 0.0177 | 0.1591 ± 0.0159 | 0.2865 ± 0.0181 |
| NERVE (w/o KoLeo) | 0.9904 ± 0.0013 | 0.9871 ± 0.0017 | 0.9904 ± 0.0013 | **0.4427** ± 0.0198 | **0.1694** ± 0.0251 | **0.2847** ± 0.0044 |

### I.4 ABLATION STUDY ON ATTENTION MECHANISM

We conducted an ablation study to directly assess EPA attention's effectiveness. Specifically, we evaluated alternative attention mechanisms: (1) Temporal transformer, which applies self-attention along the temporal axis for each EEG channel; (2) Temporal-channel transformer, which sequentially applies self-attention along the temporal and channel axes; (3) Criss-cross transformer, which adopts the criss-cross attention proposed in CBraMod (Wang et al., 2025b); and (4) EPA transformer, which employs our proposed EPA mechanism.

These attention mechanisms were evaluated in two configurations: a task-specific supervised model and a foundation model. First, we constructed transformer-based prediction models for downstream BCI tasks using four types of attention mechanisms: For a fair comparison, all models utilized a uniform 3-layer transformer architecture with a hidden dimension of 200 and were trained for 20 epochs. Table 16 presents the evaluation results of different transformer-based models across the SEED-V, TUSL, DEAP and HCI-Tagging Emotion datasets. The EPA transformer outperforms other attention mechanisms on TUSL and HCI-Tagging Emotion, and achieves second-best performance on SEED-V and DEAP. The superior results of both the temporal-channel and EPA transformers over the temporal transformer suggest that modeling spatial dependencies between electrodes enhances downstream BCI task performance. The EPA transformer demonstrated best or second-best performance in all datasets across the task-specific supervised evaluations, confirming its strong efficacy for BCI applications.

Table 16: The ablation results on attention mechanisms across task-specific supervised evaluations.

| Methods | SEED-V, 5-class | | | TUSL, 3-class | | |
|---|---|---|---|---|---|---|
| | Balanced Accuracy | Cohen's Kappa | Weighted F1 | Balanced Accuracy | Cohen's Kappa | Weighted F1 |
| Temporal | $0.2905 \pm 0.0003$ | $0.3008 \pm 0.0006$ | $0.1193 \pm 0.0009$ | $0.5350 \pm 0.1025$ | $0.2828 \pm 0.1547$ | $0.4697 \pm 0.1125$ |
| Temporal-channel | $\mathbf{0.3660} \pm 0.0004$ | $\mathbf{0.3749} \pm 0.0012$ | $\mathbf{0.2122} \pm 0.0008$ | $\underline{0.6217} \pm 0.0272$ | $\underline{0.4284} \pm 0.0427$ | $\underline{0.6145} \pm 0.0342$ |
| Criss-cross | $0.3355 \pm 0.0058$ | $0.3473 \pm 0.0062$ | $0.1753 \pm 0.0076$ | $0.5700 \pm 0.0455$ | $0.3374 \pm 0.0637$ | $0.4907 \pm 0.0483$ |
| EPA | $\underline{0.3626} \pm 0.0033$ | $\underline{0.3724} \pm 0.0035$ | $\underline{0.2086} \pm 0.0040$ | $\mathbf{0.6317} \pm 0.1123$ | $\mathbf{0.4388} \pm 0.1732$ | $\mathbf{0.6220} \pm 0.1113$ |
| Methods | DEAP, 2-class | | | HCI-Tagging Emotion, 2-class | | |
| | Balanced Accuracy | AUC-PR | AUROC | Balanced Accuracy | AUC-PR | AUROC |
| Temporal | $0.4921 \pm 0.0085$ | $0.5828 \pm 0.0337$ | $0.4520 \pm 0.0361$ | $\underline{0.5699} \pm 0.0338$ | $0.4481 \pm 0.0337$ | $0.6248 \pm 0.0344$ |
| Temporal-channel | $\mathbf{0.4955} \pm 0.0129$ | $\underline{0.6223} \pm 0.0181$ | $\mathbf{0.4878} \pm 0.0214$ | $0.5619 \pm 0.0366$ | $\underline{0.4620} \pm 0.0293$ | $\underline{0.6303} \pm 0.0350$ |
| Criss-cross | $0.4928 \pm 0.0057$ | $0.5643 \pm 0.0484$ | $0.4221 \pm 0.0555$ | $0.5480 \pm 0.0140$ | $0.4421 \pm 0.0280$ | $0.6146 \pm 0.0427$ |
| EPA | $\underline{0.4943} \pm 0.0092$ | $\mathbf{0.6256} \pm 0.0172$ | $\underline{0.4864} \pm 0.0217$ | $\mathbf{0.5933} \pm 0.0421$ | $\mathbf{0.4775} \pm 0.0459$ | $\mathbf{0.6462} \pm 0.0423$ |

We also replaced the EPA transformer with alternative attention mechanisms and pre-trained these NERVE variants on our corpus, following the same pre-training configurations. The resulting NERVE models were then fine-tuned on the downstream BCI datasets. Table 17 presents the evaluation results on the four listed datasets. As a NERVE encoder, EPA attention consistently outperforms the other variants on all datasets. This superiority stems from the position router's ability to robustly learn electrode dependencies across large-scale and diverse pre-training data, strengthening its efficacy as a foundational encoder.

Table 17: The ablation results on attention mechanisms for NERVE encoder.

| Methods | SEED-V, 5-class | | | TUSL, 3-class | | |
|---|---|---|---|---|---|---|
| | Balanced Accuracy | Cohen's Kappa | Weighted F1 | Balanced Accuracy | Cohen's Kappa | Weighted F1 |
| Temporal | $0.3563 \pm 0.0020$ | $0.2020 \pm 0.0031$ | $0.3671 \pm 0.0029$ | $\underline{0.6722} \pm 0.0483$ | $\underline{0.4965} \pm 0.0698$ | $\underline{0.6266} \pm 0.0480$ |
| Temporal-channel | $\underline{0.3746} \pm 0.0008$ | $\underline{0.2263} \pm 0.0021$ | $\underline{0.3870} \pm 0.0018$ | $0.6139 \pm 0.0307$ | $0.4030 \pm 0.0454$ | $0.5791 \pm 0.0252$ |
| Criss-cross | $0.3581 \pm 0.0066$ | $0.2050 \pm 0.0065$ | $0.3695 \pm 0.0047$ | $0.6028 \pm 0.0142$ | $0.3850 \pm 0.0257$ | $0.5732 \pm 0.0064$ |
| EPA | $\mathbf{0.3788} \pm 0.0031$ | $\mathbf{0.2305} \pm 0.0034$ | $\mathbf{0.3908} \pm 0.0028$ | $\mathbf{0.7000} \pm 0.0282$ | $\mathbf{0.5327} \pm 0.0411$ | $\mathbf{0.6507} \pm 0.0329$ |
| Methods | DEAP, 2-class | | | HCI-Tagging Emotion, 2-class | | |
| | Balanced Accuracy | AUC-PR | AUROC | Balanced Accuracy | AUC-PR | AUROC |
| Temporal | $0.4939 \pm 0.0188$ | $0.6223 \pm 0.0088$ | $0.4971 \pm 0.0214$ | $0.5451 \pm 0.0034$ | $0.4079 \pm 0.0653$ | $0.5685 \pm 0.0719$ |
| Temporal-channel | $\underline{0.5086} \pm 0.0075$ | $\underline{0.6314} \pm 0.0216$ | $\underline{0.5044} \pm 0.0277$ | $0.5671 \pm 0.0407$ | $\underline{0.4960} \pm 0.0256$ | $0.6201 \pm 0.0549$ |
| Criss-cross | $0.4967 \pm 0.0081$ | $0.6267 \pm 0.0145$ | $0.4982 \pm 0.0217$ | $\underline{0.5866} \pm 0.0106$ | $0.4436 \pm 0.0212$ | $\underline{0.6305} \pm 0.0180$ |
| EPA | $\mathbf{0.5167} \pm 0.0131$ | $\mathbf{0.6533} \pm 0.0156$ | $\mathbf{0.5406} \pm 0.0267$ | $\mathbf{0.5943} \pm 0.0177$ | $\mathbf{0.5103} \pm 0.0381$ | $\mathbf{0.6691} \pm 0.0238$ |

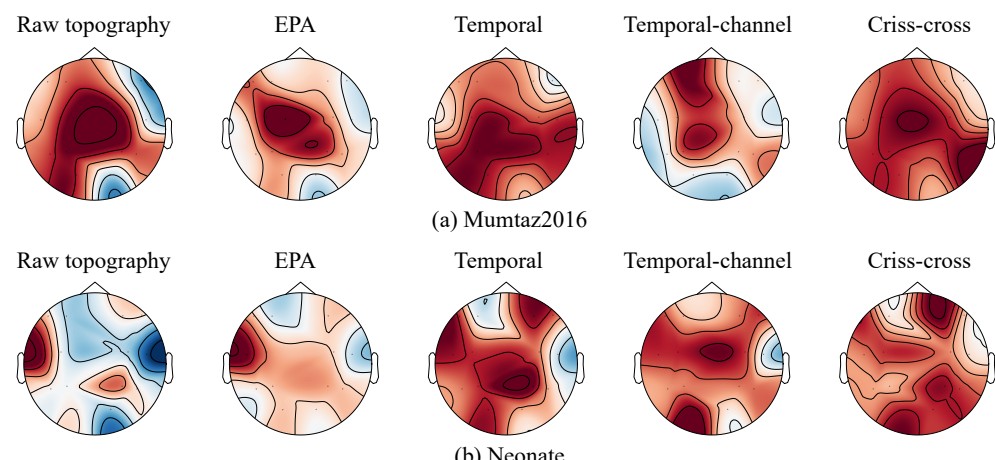

Figure 15: Topography comparison between EEG signals and channel embeddings of different attention mechanisms.

### I.5 TOPOGRAPHY COMPARISON BETWEEN DIFFERENT ATTENTION MECHANISMS

We conducted a comparative analysis between the topography of raw EEG samples obtained from our pre-training datasets and the topography of pre-trained channel embeddings learned by various attention architectures. To visualize the raw EEG signals, we computed the maximum on the signal for each channel and represented the results on a topographic map. Similarly, for the channel embeddings, we visualized the topography after extracting the output of the final encoder layer and taking the maximum value along the patch dimension for each respective channel. For this comparison, we compared our proposed NERVE (using EPA attention) against three variants equipped with different attention mechanisms as in Section I.4 and Table 17. These NERVE variants are pre-trained on our corpus, following the same pre-training configurations. Figure 15 depicts the topographic maps on Mumtaz2016 and Neonate in the pre-training corpus. We observed that the channel embedding topography of NERVE with EPA attention resembles the raw signal topography more closely than those of other architectures. This indicates that EPA attention effectively captures the inherent topological structure of the electrodes.

### I.6 ABLATION STUDY ON FINE-TUNING

We conduct an ablation study on fine-tuning method to evaluate the impact of fine-tuning. We compare the downstream performance of foundation models using linear probing, which freezes the pre-trained backbone parameters and trains only the task-specific head. For NERVE, we additionally evaluate partial fine-tuning, where the final few layers and the task head are trained while the remaining layers are kept frozen. In this experiment, we perform partial fine-tuning on NERVE with training only 3, 6, and 9 layers, respectively. Table 18 presents the various fine-tuning results on the TUSL and DEAP datasets. Compared to foundation models trained with linear probing, NERVE shows superior performance on both datasets; however, linear probing generally yields lower results than end-to-end fine-tuning. However, as shown in the performance of NERVE with partial fine-tuning on TUSL, the model's performance improves significantly when the last few layers are fine-tuned along with the task head. This suggests that while a uniformly distributed embedding space in the frozen state may not be an optimal representation for downstream BCI tasks, it serves as an effective initialization for fine-tuning-based transfer learning.

### I.7 PRE-TRAINED EMBEDDING VISUALIZATION

In this section, we visualize the pre-trained EEG embeddings of downstream datasets across different foundation models. Using the pre-trained weights, we extract embeddings of test-set samples by average pooling over EEG patches, normalize them to unit norm, and project the embedding space with UMAP (McInnes et al., 2018). Figure 16 depicts the UMAP visualization plots on the DEEP

Table 18: The ablation results on fine-tuning methods.

| Methods | TUSL, 3-class | | | DEAP, 2-class | | |
|---------|---------------|---|---|---------------|---|---|
| | Balanced Accuracy | Cohen's Kappa | Weighted F1 | Balanced Accuracy | AUC-PR | AUROC |
| BIOT (LP) | 0.4111 ± 0.0956 | 0.1040 ± 0.1539 | 0.3720 ± 0.0703 | 0.4972 ± 0.0052 | 0.6014 ± 0.0306 | 0.4379 ± 0.0234 |
| LaBraM (LP) | 0.4026 ± 0.0674 | 0.0232 ± 0.0629 | 0.3160 ± 0.0932 | 0.5000 ± 0.0000 | 0.5778 ± 0.0001 | 0.4236 ± 0.0001 |
| CBraMod (LP) | 0.4028 ± 0.0336 | 0.0726 ± 0.0512 | 0.3631 ± 0.0246 | 0.4627 ± 0.0200 | 0.5564 ± 0.0181 | 0.3896 ± 0.0233 |
| NERVE (LP) | **0.4584** ± 0.0312 | **0.1775** ± 0.0252 | **0.4050** ± 0.0420 | **0.5000** ± 0.0000 | **0.6420** ± 0.0126 | **0.5059** ± 0.0128 |
| NERVE (PT3) | 0.6445 ± 0.0336 | 0.5857 ± 0.0272 | 0.4436 ± 0.0508 | 0.4958 ± 0.0062 | 0.6285 ± 0.0159 | 0.4974 ± 0.0176 |
| NERVE (PT6) | 0.6305 ± 0.0336 | 0.5759 ± 0.0284 | 0.4249 ± 0.0503 | 0.5103 ± 0.0086 | 0.6332 ± 0.0067 | 0.5157 ± 0.0082 |
| NERVE (PT9) | **0.6528** ± 0.0039 | **0.5953** ± 0.0019 | **0.4602** ± 0.0009 | **0.5122** ± 0.0159 | **0.6412** ± 0.0042 | **0.5266** ± 0.0168 |

and High-Gamma datasets. Each color represents a subject ID in the test set. NERVE produces an embedding space that is evenly distributed, with distinct inter-subject separation and cohesive intra-subject grouping. Such embedding structure enhances robustness to inter- and intra-subject variability without utilizing subject identifiable information.

**DEAP**

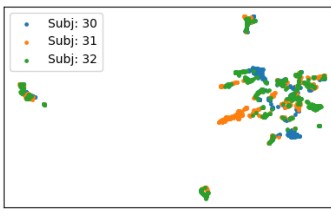 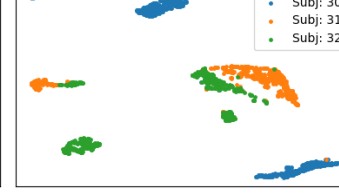 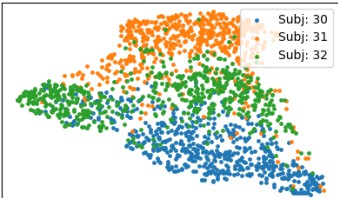

**High-Gamma**

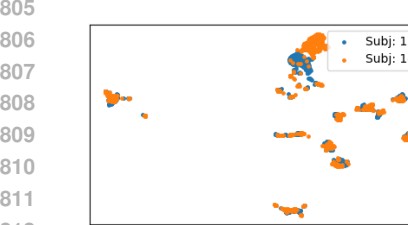 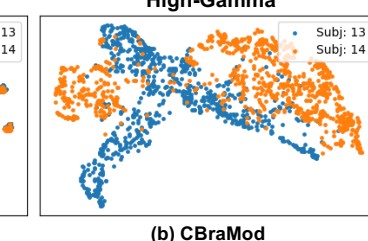 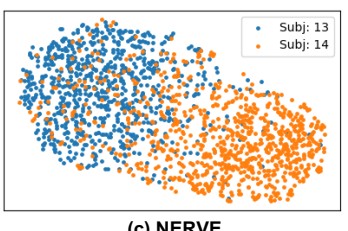

(a) LaBraM          (b) CBraMod          (c) NERVE

Figure 16: Pre-trained embedding visualization

## J LIMITATIONS AND FUTURE WORKS

Our work has some limitations. In the current architecture, explicit physical priors are not strictly incorporated into the position router. While this lack of structural regularization allows NERVE the flexibility to learn signal-driven soft cortical groupings and latent spatial patterns optimal for the task, it may pose challenges regarding interpretability. To address this, we plan to construct anatomical priors on the position router to better consider neural interaction mechanisms and design enhanced EEG signal encoding methods that are both flexible and interpretable.

This work does not yet explore the scaling laws of EEG pre-training on larger datasets, primarily due to resource limitations. Furthermore, real-world clinical validation is still required to fully assess the practical utility of EEG foundation models. To address these limitations, we plan to expand our pre-training corpus by incorporating the TUEG dataset (Obeid & Picone, 2016), a large-scale clinical EEG collection with a total duration of 27.063 hours (previously used in CBraMod pre-training). We will then rigorously investigate the scaling behavior of NERVE. Finally, we aim to collaborate with EEG sensor research institutes and clinical institutions to evaluate NERVE on in-the-wild EEG recordings and assess its robustness to previously unseen acquisition conditions.

