# OpenReview forum: "NERVE: Noise-Variability-Robust EEG Foundation Model with Electrode-Brain Interactions"
_ICLR.cc/2026/Conference — Submitted to ICLR 2026_

### Official Review · Reviewer_1XLF · 2025-10-23

**Soundness:** 2
**Presentation:** 3
**Contribution:** 2
**Rating:** 6
**Confidence:** 4

**Summary:**

The authors propose **NERVE**, a noise- and variability-robust EEG foundation model designed to address key challenges in EEG analysis, including low signal-to-noise ratios (SNR), high inter-sample variability, and spatial dependencies arising from electrode placement in acquisition systems. The proposed framework consists of three core components. First, a **noise-robust neural tokenizer** encodes EEG patches into discrete neural tokens. Second, a **variability-robust pretraining strategy** enforces alignment and uniformity in the representation space to improve robustness against distributional shifts. Third, an **electrode-position–aware (EPA) transformer** serves as the backbone for both the tokenizer and the foundation model, explicitly modeling the spatial structure of EEG channels.

**Strengths:**

1.The topic of EEG foundation models (EFMs) is highly significant and relevant to the advancement of brain–computer interface (BCI) technologies.

2.The overall logical flow of the introduction and methodology sections is clear and easy to follow.

3.The work appears to be technically complete and systematically presented.

**Weaknesses:**

1.**Section 2.1** introduces the EPA transformer, which is intended to capture spatial dependencies among electrodes and brain regions. However, the logical flow of this section is not tightly connected to the stated goal of modeling such spatial dependencies.

2.**Section 2.2** does not directly address the low SNR problem in EEG signals. Merely citing the inspiration behind the method does not sufficiently justify its validity or effectiveness.

3.In **Section 3.3**, the attribution of the model’s strong performance lacks clarity. While Table 3 demonstrates overall improvement, there is insufficient evidence to link the observed gains to the specific factors claimed by the authors.

**Questions:**

1.Time embeddings and channel embeddings are defined as *d*-dimensional learnable vectors. How do the authors ensure that the learned parameters indeed correspond to temporal and channel-specific information, respectively?

2.The discussion of electrode-to-brain and brain-to-electrode interactions is overly brief. Could the authors provide a more formal justification or theoretical explanation for these interactions?

---

> ### Author Response · Authors · 2025-11-21
> **Response to Reviewer 1XLF (1)**
>
> > [W1] Section 2.1 introduces the EPA transformer, which is intended to capture spatial dependencies among electrodes and brain regions. However, the logical flow of this section is not tightly connected to the stated goal of modeling such spatial dependencies.
>
> - Thank you for your valuable comment. We propose NERVE, an EEG foundation model that addresses three challenges arising from EEG signal acquisition: low signal-to-noise ratio, high variability between EEG samples, and spatial dependencies stemming from electrode topology. The EPA transformer is designed to model the relationship between electrodes (EEG channels) and their associated cortical regions via attention scores. Each EPA transformer layer consists of sequentially aligned temporal and channel blocks. Within the channel block, EPA attention is employed to capture spatial dependencies arising from electrode topology.
>
> - Specifically, we define a position router $P\in\mathbb{R}^{R\times N \times d}$ to map the positions of electrodes to $R$ distinct brain regions, aligning with standard topographical groupings in EEG research (e.g., frontal, parietal, or occipital). Then, EPA attention calculates attention score via the matrix multiplication of query and key representations of EEG channel-wise representations $H$ and position routers:
>
> $$EPA-Attention(Q_H, K_H, V_H, Q_P, K_P) = softmax(\frac{Q_HQ_P^TK_PK_H^T}{\sqrt{d}})V_H$$
>
> - EPA attention yields attention scores decomposable into two distinct components: electrode-to-brain ($Q_HQ_P^T$) and brain-to-electrodes ($K_PK_H^T$). While spatial attention in existing EEG foundation models typically models channel dependency through direct matrix multiplication of EEG channels ($Q_HK_H^T$), EPA attention structures the spatial dependencies by bridging electrodes and cortical regions using position routers.
>
> - We investigated the formulation of EPA attention and demonstrated that the EPA transformer is a sequential formulation of the generalized criss-cross transformer proposed in CBraMod (refer to our responses to Q2). To substantiate the motivation for EPA attention and its link to the observed superior performance, we conducted various analyses on benchmark datasets, including interpretability analysis and an ablation study on attention mechanisms (refer to our responses to W3). These experiments confirm that EPA attention effectively captures the raw EEG signal topography, leading to superior performance on BCI tasks compared to other attention mechanisms.

---

> ### Author Response · Authors · 2025-11-21
> **Response to Reviewer 1XLF (2)**
>
> > [W2] Section 2.2 does not directly address the low SNR problem in EEG signals. Merely citing the inspiration behind the method does not sufficiently justify its validity or effectiveness.
>
> - Thank you for your constructive feedback. We acknowledge that simply citing the inspiration may not fully elucidate the method's effectiveness. To clarify, the low SNR inherent in EEG signals complicates representation learning and degrades generalization (see Table 1 and Section 4 in the manuscript). Even after standard filtering, residual noise inevitably remains [1]. To directly address this, our neural tokenizer training does not merely adopt the concept of a Denoising Autoencoder (DAE) [2] but is specifically engineered to perform denoising temporal-spectral prediction. This objective compels the model to extract high-level semantic representations that are invariant to noise, rather than simply memorizing the noisy input [3]. Geometrically, this process functions as projecting a corrupted sample back onto the intrinsic uncorrupted data manifold [3]. Therefore, this mechanism is not just an auxiliary task but an essential component that directly mitigates the low SNR problem by separating the true signal from noise, thereby enhancing both robustness and generalization.
>
> - Our denoising temporal-spectral prediction is meticulously designed to learn noise-invariant representations of EEG signals. Merely reconstructing the signal in the time domain is insufficient for capturing inherent semantic information due to the low SNR of EEG. Therefore, we propose a dual reconstruction objective covering both the time domain and the full frequency spectrum. Since noise artifacts often perturb specific frequency bands, simultaneous reconstruction in temporal and spectral spaces facilitates a deeper extraction of the intrinsic knowledge within EEG signals.
>
> - To practically justify the validity of our work, we additionally designed synthetic noise scenarios by directly controlling the signal-to-noise ratio (SNR) of the input signal, which is defined as $SNR_{DB}=10log_{10}(\frac{{P_{signal}}}{P_{noise}})$, where the power $P=lim_{T\rightarrow \infty} \frac{1}{T} \int_{-\frac{T}{2}}^{\frac{T}{2}}s^2(t)dt$. We defined several noise levels using SNR values of 20, 10, 5 dB (ranging from clean to noisy signals) and corrupted the input signal with these artificial noises. As shown in Table 2 below, NERVE demonstrates the superior robustness on these corrupted signals, achieving smallest accuracy drop and prediction entropy change compared to existing EEG foundation models. These analyses demonstrates the robustness of NERVE to complex and unstructured noises as well as physiological noises, emphasizing the practical relevance and generalizability of our work. To faciliate your understanding, we provided a case visualization of synthetic noise scenarios in Figure 8 of the revised manuscript. Additional analysis results can also be found in Figure 9 and Section G.1 in the Appendix.
>
> (Table 1) Noise robustness analysis on corrupted EEG samples from High-Gamma and HCI-Tagging Emotion
>
> |High_Gamma|  | $\triangle$ Acc |  |  | \|$\triangle$ Entropy\| | |HCI-Tagging Emotion|  | $\triangle$ AUROC |  |  | \|$\triangle$ Entropy\| | |
> |:---:|:---:|:---:|:---:|:---:|:---:|:---:|:---:|:---:|:---:|:---:|:---:|:---:|:---:|
> ||LaBraM|CBraMod|NERVE|LaBraM|CBraMod|NERVE||LaBraM|CBraMod|NERVE|LaBraM|CBraMod|NERVE|
> |SNR 5|-0.686|-0.381|**-0.208**|1.197|0.581|**0.245**|SNR 5|-0.058|-0.046|**-0.028**|0.066|0.075|**0.005**|
> |SNR 10|-0.444|-0.111|**-0.042**|0.608|0.133|**0.045**|SNR 10|-0.047|-0.015|**-0.014**|0.080|0.035|**0.003**|
> |SNR 20|-0.061|-0.006|**-0.001**|0.065|0.007|**0.001**|SNR 20|-0.039|0.001|**-0.002**|0.068|0.007|**0.001**|

---

> ### Author Response · Authors · 2025-11-21
> **Response to Reviewer 1XLF (3)**
>
> - We also conducted the ablation studies on noise-robust neural tokenizer. Specifically, we compared NERVE trained with the noise-robust neural tokenizer against NERVE trained with a standard neural tokenizer (without noise augmentation), which we term as NERVE (standard). Then, we analyzed the robustness of both models to diverse noise scenarios. As shown in Table 2, the original NERVE exhibits superior robustness to noise, characterized by small accuracy degradation and predictive entropy changes. The ablation study underscores that noise-robust tokenizer training is essential for achieving robustness to physiological noise artifacts and low SNR signals. More experimental results are presented in Section I.1 in the Appendix of the revised manuscript.
>
> (Table 2) Analysis on robustness to noises of NERVE trained with noise-robust and standard neural tokenizers
>
> |High_Gamma||$\triangle$ Acc &emsp;&emsp;&emsp;&emsp;&emsp;|| \|$\triangle$ Entropy\|&emsp;&emsp;&emsp; | DEAP||$\triangle$ Acc &emsp;&emsp;&emsp;&emsp;&emsp;||\|$\triangle$ Entropy\|&emsp;&emsp;&emsp;|
> |:---:|:---:|:---:|:---:|:---:|:---:|:---:|:---:|:---:|:---:|
> |Noise|NERVE|NERVE (standard)|NERVE|NERVE (standard)|Noise|NERVE|NERVE (standard)|NERVE|NERVE (standard)|
> |EMG|-0.014|**-0.011**|**0.014**|0.017|EMG|**0.000**|-0.001|**0.001**|0.002|
> |EOG|**0.000**|-0.017|**0.001**|0.015|EOG|**-0.001**|-0.002|**0.006**|0.008|
> |Environment|**0.000**|-0.011|**0.001**|0.017|Environment|**0.000**|**0.000**|**0.001**|**0.001**|
> |Gaussian|**-0.131**|-0.147|**0.276**|0.345|Gaussian|**-0.025**|-0.029|**0.035**|0.061|
> |SNR 5|-0.175|**-0.169**|**0.374**|0.394|SNR 5|**-0.019**|-0.026|**0.030**|0.060|
> |SNR 10|**-0.003**|-0.083|**0.040**|0.188|SNR 10|**-0.008**|-0.028|**0.020**|0.048|
> |SNR 20|**-0.001**|-0.005|**0.001**|0.0015|SNR 20|**-0.002**|-0.007|**0.005**|0.012|
>
> - Once again, We genuinely appreciate your constructive feedback. Your insightful advice has been instrumental in guiding us to sharpen our research motivation and strengthen the validity of our method.
>
> [1] Darren Tanner, Kara Morgan-Short, and Steven J Luck. How inappropriate high-pass filters can produce artifactual effects and incorrect conclusions in erp studies of language and cognition. Psychophysiology, 52(8):997–1009, 2015.
>
> [2] Vincent, P., Larochelle, H., Bengio, Y., & Manzagol, P. A. (2008, July). Extracting and composing robust features with denoising autoencoders. In Proceedings of the 25th international conference on Machine learning (pp. 1096-1103).
>
> [3] Vincent, P., Larochelle, H., Lajoie, I., Bengio, Y., Manzagol, P. A., & Bottou, L. (2010). Stacked denoising autoencoders: Learning useful representations in a deep network with a local denoising criterion. Journal of machine learning research, 11(12).

---

> ### Author Response · Authors · 2025-11-21
> **Response to Reviewer 1XLF (4)**
>
> > [W3] In Section 3.3, the attribution of the model’s strong performance lacks clarity. While Table 3 demonstrates overall improvement, there is insufficient evidence to link the observed gains to the specific factors claimed by the authors.
>
> - Thank you for your thorough feedback. Following your advice, we have conducted various analyses and ablation studies to investigate the factors contributing to the observed superior performance. The observed improvements can be primarily attributed to three aspects:
>
> - Effectiveness of EPA Attention
>     - First, we conducted a visualization analysis to compare the topography of the raw sample with class activation topography of NERVE on the High-Gamma dataset. The visualization results and the setup of visualization analysis are provided in Figure 13 and Section H.2 of the Appendix in the revised manuscript. We observed that electrodes associated with the left hand exhibit patterns relatively symmetric to those of the right hand, while both cortical regions are activated for the 'both feet' class. In contrast, minimal activation is observed during the resting state. Although NERVE's class activation topography is not identical to the raw EEG signals, its representations capture broadly similar spatial patterns, demonstrating distinct differences across classes.
>
>     - Second, we compared the topography of raw samples from the pre-training datasets with the channel embedding topographies learned by different attention architectures. Specifically, we analyzed NERVE (using EPA attention) against variants equipped with temporal attention (attention across the time dimension), temporal-channel attention (sequential blocks of attention in time and channel dimensions), and criss-cross attention (the mechanism proposed by CBraMod). The visualization results are presented in Figure 15 of the revised manuscript. We observed that the channel embedding topography of NERVE with EPA attention resembles the raw signal topography more closely than those of other architectures. This indicates that EPA attention effectively captures the inherent topological structure of the electrodes.
>
>     - Third, we visualized the cosine similarity heatmap using the Mumtaz2016 dataset from our pre-training corpus. As shown in Figure 12 of the revised manuscript, we observe high similarity among specific groups of electrodes, forming a distinct block-wise similarity structure. This observation underscores the necessity of region-wise channel modeling for EEG signals and supports the validity of EPA attention.
>
>     - Lastly, we conducted an ablation study to evaluate various attention mechanisms within the NERVE backbone. We replaced EPA attention with temporal, temporal-channel, and criss-cross attention modules. As shown in Table 3 below, NERVE with EPA attention outperforms other attention-based architectures on the TUEV and DEAP datasets, highlighting the effectiveness of EPA attention for EEG modeling. Additional experimental results are provided in Tables 17 and Section I.4 of the revised manuscript.
>
> (Table 3) The ablation results on attention mechanism for NERVE backbone.
> |||TUEV|||DEAP||
> |:---:|:---:|:---:|:---:|:---:|:---:|:---:|
> ||Acc|Kappa|F1|Acc|AUPRC|AUROC|
> |Temporal|0.5386 $\pm$ 0.0084|0.4570 $\pm$ 0.0110 | 0.7112 $\pm$ 0.0075 |0.4939 $\pm$ 0.0188|0.6223 $\pm$ 0.0088|0.4971 $\pm$ 0.0214|
> |Temporal-channel|0.5503 $\pm$ 0.0169|0.4753 $\pm$ 0.0043|0.7235 $\pm$ 0.0049|0.5086 $\pm$ 0.0075|0.6314 $\pm$ 0.0216|0.5044 $\pm$ 0.0277|
> |Criss-cross|0.5270 $\pm$ 0.0346|0.4615 $\pm$ 0.0587|0.7227 $\pm$ 0.0315|0.4967 $\pm$ 0.0081|0.6267 $\pm$ 0.0145|0.4982 $\pm$ 0.0217|
> |EPA|**0.5595** $\pm$ 0.0106|**0.4810** $\pm$ 0.0146|**0.7249** $\pm$ 0.0095|**0.5167** $\pm$ 0.0131|**0.6553** $\pm$ 0.0156|**0.5406** $\pm$ 0.0267|

---

> ### Author Response · Authors · 2025-11-21
> **Response to Reviewer 1XLF (5)**
>
> > [W3] In Section 3.3, the attribution of the model’s strong performance lacks clarity. While Table 3 demonstrates overall improvement, there is insufficient evidence to link the observed gains to the specific factors claimed by the authors.
>
> - Effectiveness of noise-robust neural tokenizer training
>     - We compared NERVE trained with the noise-robust neural tokenizer against NERVE trained with a standard neural tokenizer (without noise augmentation). As shown in Table 4 NERVE utilizing the noise-robust neural tokenizer outperforms the model trained without noise augmentation in BCI tasks, indicating the noise-robust training enhances the generalization capacity of NERVE and thereby enhancing the performance on downstream BCI tasks.
>
>     - It has been theoretically proven that training with noise enhances generalization performance, even on clean data. Training with noise improves generalization performance by acting as training with generalized Tikhonov regularization [4]. Our noise-robust neural tokenizer training closely resembles a denoising autoencoder [2]. The goal of denoising autoencoder is not the task of denoising itself. Denoising serves as a training criterion for learning to extract useful representations that constitute higher-level semantic information [3]. In particular, when Gaussian noise is added, the denoising autoencoder training criterion is equivalent to score matching objective, which indicates that optimizing this criterion approximates the gradient of the log density [5]. Thus, our noise-robust training can be theoretically regarded as learning the useful representations that effectively approximate the data manifold.
>
> (Table 4) The ablation results on noise-robust training of neural-tokenizer
>
> |Methods|  |SEED-V| | |TUSL| |
> |:---:|:---:|:---:|:---:|:---:|:---:|:---:|
> |NERVE|**0.3788** $\pm$ 0.0031|**0.2305** $\pm$ 0.0034|**0.3908** $\pm$ 0.0028|**0.7000** $\pm$ 0.0282|**0.5327** $\pm$ 0.0411|**0.6507** $\pm$ 0.0329|
> |NERVE (standard)|0.3764 $\pm$ 0.0014|0.2291 $\pm$ 0.0016|0.3096 $\pm$ 0.0012|0.6050 $\pm$ 0.0452|0.3748 $\pm$ 0.0736|0.5742 $\pm$ 0.0320|
>
>
> [4] Bishop, C. M. (1995). Training with noise is equivalent to Tikhonov regularization. Neural computation, 7(1), 108-116.
>
> [5] Vincent, P. (2011). A connection between score matching and denoising autoencoders. Neural computation, 23(7), 1661-1674.
>
> ----
>
>
> - Effectiveness of variability-robust pre-training
>     - We evaluate the impact of variability-robust learning in NERVE pre-training on downstream BCI tasks. Since variability robustness is encouraged via the KoLeo loss by promoting sample separability, we compare NERVE with its variant pre-trained without the KoLeo loss, which we term as NERVE (w/o KoLeo). As shown in Table 5 below, NERVE achieves superior performance across all datasets, suggesting that a uniformly distributed embedding space enhances downstream BCI task performance.
>
> (Table 5) The ablation results on variability-robust pre-training (KoLeo loss)
>
> |||TUEV|||DEAP||
> |:---:|:---:|:---:|:---:|:---:|:---:|:---:|
> ||Acc|Kappa|F1|Acc|AUPRC|AUROC|
> |NERVE|**0.5595** $\pm$ 0.0106|**0.4810** $\pm$ 0.0146|**0.7249** $\pm$ 0.0095|**0.5167** $\pm$ 0.0131|**0.6533** $\pm$ 0.0156|**0.5406** $\pm$ 0.0267|
> |NERVE (w/o KoLeo)|0.5495 $\pm$ 0.0105|0.4517 $\pm$ 0.0217|0.7120 $\pm$ 0.0217|0.5001 $\pm$ 0.0075|0.6340 $\pm$ 0.0142|0.5121 $\pm$ 0.0217|
>
> - To summarize, the improved performance of NERVE on diverse BCI tasks can be systematically attributed to our three main contributions to address acquisition-related challenges: noise-robust neural tokenizer training, variability-robust pre-training, and EPA attention for capturing the spatial structure between electrodes. We sincerely appreciate your comment, as it has enabled us to sharpen the technical validity of our work.

---

> ### Author Response · Authors · 2025-11-21
> **Response to Reviewer 1XLF (6)**
>
> > [Q1] Time embeddings and channel embeddings are defined as d-dimensional learnable vectors. How do the authors ensure that the learned parameters indeed correspond to temporal and channel-specific information, respectively?
>
> - Thank you for your detailed comment. Time embeddings indicate the position of each patch and are commonly employed in transformer-based architectures [6, 7]. Since the self-attention mechanism is permutation-invariant, time embeddings encode the sequential structure of the EEG signals. Channel embeddings are designed to align with the electrodes of the standard 10-20 system. Accordingly, the specific channel embedding for each channel is selected and added to the corresponding patch. Then, the sequence of patch embeddings combined with time and channel embeddings are fed to the EPA Transformer encoder.
>
> - To investigate whether the learned embeddings of NERVE capture the inter-channel structure, we visualized the inter-channel cosine similarity heatmap using the BCIC2020-3 dataset, a subset of our pre-training corpus. As shown in Figure 12 of the revised manuscript, we observe high similarity among specific groups of electrodes, forming a distinct block-wise similarity structure. This suggests that while each channel embedding provides electrode-specific information to the corresponding patches, the final NERVE representations effectively encode the region-wise topological structure of the EEG signals.
>
> [6] Vaswani, A., Shazeer, N., Parmar, N., Uszkoreit, J., Jones, L., Gomez, A. N., ... & Polosukhin, I. (2017). Attention is all you need. Advances in neural information processing systems, 30.
> [7] Weibang Jiang, Liming Zhao, and Bao liang Lu. (2024). Large brain model for learning generic representations with tremendous EEG data in BCI. In The Twelfth International Conference on Learning Representations,

---

> ### Author Response · Authors · 2025-11-21
> **Response to Reviewer 1XLF (7)**
>
> > [Q2] The discussion of electrode-to-brain and brain-to-electrode interactions is overly brief. Could the authors provide a more formal justification or theoretical explanation for these interactions?
>
> - Thank you for your detailed comment. We propose EPA attention to model the relationship between EEG channels arising from the electrode placement during signal acqusition. Specifically in EPA attention, we define a position router $P\in\mathbb{R}^{R\times N \times d}$ to map the positions of electrodes to $R$ distinct brain regions, aligning with standard topographical groupings in EEG research (e.g., frontal, parietal, or occipital). Then, EPA attention calculates attention score via the matrix multiplication of query and key representations of EEG channel-wise representations $H$ and position routers:
>
> $$EPA-Attention(Q_H, K_H, V_H, Q_P, K_P) = softmax(\frac{Q_HQ_P^TK_PK_H^T}{\sqrt{d}})V_H$$
>
> - EPA attention yields attention scores decomposable into two distinct components: electrode-to-brain ($Q_HQ_P^T$) and brain-to-electrodes ($K_PK_H^T$). While spatial attention in existing EEG foundation models typically models channel dependency through direct matrix multiplication of EEG channels ($Q_HK_H^T$), EPA attention structures the spatial dependencies by bridging electrodes and cortical regions using position routers.
>
> - Furthermore, we denote $S_j=(Q_H M_j K_H^T)/\sqrt{d}$ and $M_j=Q_P^T K_P=(PW_Q)^T(PW_K)\in\mathbb{R}^{d\times d}$. Since $PW_Q \in \mathbb{R}^{d\times R}$, the rank of $M_j$ is inherently bounded by $R$. Given that $rank(M_j)\leq R$, so EPA attention constrains the channel-wise score matrix $S_j=(Q_H M_j K_H^T)/\sqrt{d}$ to lie in a low-dimensional, router-defined subspace. This low-rank and topology-aligned regularization encourages robust electrode–region–electrode dependencies while suppressing spurious correlations among channels.
>
> - EPA attention is not merely a sequential adaptation of criss-cross attention; rather, it constitutes a sequential formulation of 'generalized' criss-cross attention. We denote $S_j=(Q_H M_j K_H^T)/\sqrt{d}$ and $M_j=Q_P^T K_P=(PW_Q)^T(PW_K)\in\mathbb{R}^{d\times d}$. If $M_j\equiv I$, EPA attention is reduced to the sequential conversion of criss-cross attention. Thus, criss-cross attention can be regarded as a special case of EPA attention.
>
> - Furthermore, given that $PW_Q \in \mathbb{R}^{d\times R}$, the rank of $M_j$ is inherently bounded by $R$. Since $rank(M_j)\leq R$, so EPA attention constrains the channel-wise score matrix $S_j=(Q_H M_j K_H^T)/\sqrt{d}$ to lie in a low-dimensional, router-defined subspace. This low-rank and topology-aligned regularization encourages robust electrode–region–electrode dependencies while suppressing spurious correlations among channels.
>
>
> $$\frac{\partial L}{\partial S_j} = \frac{\partial L}{\partial Q_H} \left(\frac{\partial S_j}{\partial Q_H}\right)^{-1}=\frac{\partial L}{\partial Q_H}(M_jK_H^T)^{-1}=\frac{\partial L}{\partial Q_H}(K_H)^{-1}(M_j^T)^{-1}$$
>
>
> - When deriving the gradient of the loss function $L$ with respect to the attention score $S_j$ (see the equation above), $M_j^T$ acts as a multiplier to the gradient of the standard channel attention. Given that $rank(M_j)\leq R$, this operation constrains the gradient flow, imposing an implicit regularization effect on the learned parameters.
>
> - We deeply appreciate your valuable feedback, as it has enabled us to sharpen the technical contribution of our work. We have incorporated this discussion into Section F of the revised manuscript.

---

> > ### Author Response · Authors · 2025-11-30
> >
> > Dear Reviewer 1XLF,
> >
> > We appreciate your positive comments and insightful feedback, which led to constructive discussions and meaningful improvements in the revised manuscript.
> >
> > Thanks to your feedback, we were able to **enhance the logical flow of EPA attention for capturing spatial dependencies** (W1), **substantiate how NERVE addresses the low SNR problem** and demonstrate **its superior robustness in low SNR scenarios** (W2). Lastly, we could **substantiate the link between our technical components and improved performance** through various ablation studies and analyses (W3), and **formally justify the interactions between electrodes and brain regions in our EPA mechanism** (Q2).
> >
> > We sincerely thank you for helping us strengthen the technical contribution of our work in various aspects. We have uploaded our meta responses for this rebuttal phase, summarizing the key improvements resulting from our discussions with you and other reviewers. Please review our meta responses, and we would be happy to engage further should you have additional questions.
> >
> > Sincerely,
> >
> > The authors

---

### Official Review · Reviewer_c7uQ · 2025-10-24

**Soundness:** 3
**Presentation:** 2
**Contribution:** 2
**Rating:** 6
**Confidence:** 3

**Summary:**

The paper proposes NERVE, a novel EEG foundation model designed to address key acquisition-related challenges of EEG signals: low signal-to-noise ratio, high inter- and intra-subject variability, and spatial dependencies among electrodes. By introducing a noise-robust neural tokenizer, a variability-robust pretraining objective, and an electrode-position-aware transformer architecture, NERVE demonstrates competitive performance across multiple BCI tasks and improved robustness compared to existing foundation models.

**Strengths:**

The paper is well-organized, with each component clearly explained and logically presented. NERVE demonstrates strong empirical performance, achieving state-of-the-art or competitive results on the majority of downstream tasks compared to several recent EEG foundation models. The motivation is clear and well-grounded, and the method is thoughtfully designed to directly address the three key challenges highlighted in the paper: low signal-to-noise ratio, high intra- and inter-subject variability, and spatial dependencies among electrodes.

**Weaknesses:**

1. The criss-cross attention mechanism depicted in Figure 1(b) appears to be inaccurately represented.

2. While the core contribution of this work lies in proposing a noise-robust foundation model, the evaluation of noise robustness is insufficient. The authors have conducted comparative anti-noise experiments only on a limited number of datasets (Fig. 4, Fig. 7). Moreover, the experimental results indicate that the differences between NERVE and the baseline models on EMG, EOG, and environmental noise are marginal. The Gaussian condition, under which NERVE demonstrates its best noise robustness, is less common in practical scenarios, as most EEG signals undergo preprocessing—thereby diminishing the practical impact of NERVE’s contribution. Additionally, the authors should provide visual examples under varying noise injection conditions to better illustrate model behavior.

3. The authors present extensive experiments to demonstrate the stability of NERVE’s performance in intra-subject settings. However, in my own opinion, this emphasis appears misaligned with the intended role of a foundation model for EEG. Such models are expected to prioritize generalization across diverse scenarios, tasks, and multicenter environments, rather than focusing on individual-specific adaptations. Robustness in both intra-subject and inter-subject contexts has already been extensively studied in dedicated literature.

**Questions:**

1. How did the authors verify that the Neural Tokenizer had been adequately trained for noise robustness prior to the masked reconstruction pre-training?

2.  Could the authors provide an ablation study on the Neural Tokenizer? Since the noise robustness of NERVE appears to stem primarily from the pre-trained Neural Tokenizer, would the anti-noise performance deteriorate significantly if this component were removed?

---

> ### Author Response · Authors · 2025-11-21
> **Response to Reviewer c7uQ (1)**
>
> > [w1] The criss-cross attention mechanism depicted in Figure 1(b) appears to be inaccurately represented.
>
> - Thank you for your detailed comment. We acknowledge that while our original intention in Figure 1 was to compare attention mechanisms in the channel dimension, the caption 'criss-cross attention' for Figure 1(b) did not clearly represent the spatial attention aspect of the criss-cross attention mechanism. Following your advice, we have refined the captions in Figure 1 to enhance clarity and better highlight the technical contributions of NERVE. Specifically, we renamed them to: 'Full attention (BIOT, LaBraM)' for Figure 1(a), 'Spatial attention (CBraMod)' for Figure 1(b), and 'EPA attention (NERVE)' for Figure 1(c). Please refer to the updated Figure 1 in the revised manuscript.

---

> ### Author Response · Authors · 2025-11-21
> **Response to Reviewer c7uQ (2)**
>
> > [W2] While the core contribution of this work lies in proposing a noise-robust foundation model, the evaluation of noise robustness is insufficient. The authors have conducted comparative anti-noise experiments only on a limited number of datasets (Fig. 4, Fig. 7). Moreover, the experimental results indicate that the differences between NERVE and the baseline models on EMG, EOG, and environmental noise are marginal. The Gaussian condition, under which NERVE demonstrates its best noise robustness, is less common in practical scenarios, as most EEG signals undergo preprocessing—thereby diminishing the practical impact of NERVE’s contribution. Additionally, the authors should provide visual examples under varying noise injection conditions to better illustrate model behavior.
>
> 1. While the core contribution of this work lies in proposing a noise-robust foundation model, the evaluation of noise robustness is insufficient. The authors have conducted comparative anti-noise experiments only on a limited number of datasets (Fig. 4, Fig. 7).
>
> - Thank you for your detailed review. We acknowledge that the results on noise robustness are not sufficiently presented due to the page limit. We additionally present the noise robustness analysis results on the TUSL and SEED-V datasets (see Table 1 below). NERVE consistently demonstrates superior robustness across most noise scenarios on the evaluated datasets compared to existing EEG foundation models. Following your advice, we include the extended results on noise robustness analysis in Figure 9 of the revised manuscript.
>
> (Table 1) Noise robustness analysis on corrupted EEG samples from TUSL and SEED-V
>
> |TUSL|  | $\triangle$ Acc |  |  | \|$\triangle$ Entropy\| | |SEED-V|  | $\triangle$ Acc |  |  | \|$\triangle$ Entropy\| | |
> |:---:|:---:|:---:|:---:|:---:|:---:|:---:|:---:|:---:|:---:|:---:|:---:|:---:|:---:|
> ||LaBraM|CBraMod|NERVE|LaBraM|CBraMod|NERVE||LaBraM|CBraMod|NERVE|LaBraM|CBraMod|NERVE|
> |EMG|**-0.008**|0.125|-0.014|0.030|0.240|**0.014**|EMG|-0.196|-0.158|**-0.090**|0.853|0.641|**0.343**|
> |EOG|-0.010|**0.000**|**0.000**|0.014|**0.000**|0.001|EOG|-0.014|**-0.012**|-0.022|**0.039**|0.042|0.071|
> |Environment|0.023|0.014|**0.000**|0.024|0.031|**0.001**|Environment|-0.036|-0.122|**-0.015**|0.096|0.452|**0.049**|
> |Gaussian|-0.323|**0.000**|-0.131|0.642|**0.079**|0.276|Gaussian|-0.189|-0.158|**-0.142**|0.536|**0.439**|0.631|
> |SNR 5|-0.304|0.211|**-0.175**|0.611|0.382|**0.374**|SNR 5|-0.185|-0.160|**-0.121**|0.530|0.471|**0.467**|
> |SNR 10|-0.348|-0.056|**-0.003**|0.741|0.224|**0.040**|SNR 10|-0.165|-0.148|**-0.080**|0.469|0.521|**0.279**|
> |SNR 20|-0.206|0.025|**-0.011**|0.414|0.072|**0.019**|SNR 20|-0.090|-0.102|**-0.017**|0.254|0.321|**0.051**|

---

> > ### Author Response · Authors · 2025-11-21
> > **Response to Reviewer c7uQ (3)**
> >
> > - To deeply validate the robustness of NERVE to complex and unstructured noises, we control the signal-to-noise ratio (SNR) of the input signal, which is defined as $SNR_{DB}=10log_{10}(\frac{{P_{signal}}}{P_{noise}})$, where the power $P=lim_{T\rightarrow \infty} \frac{1}{T} \int_{-\frac{T}{2}}^{\frac{T}{2}}s^2(t)dt$. SNR serves as the standard metric for quantifying signal quality and evaluating model robustness across varying noise levels [1, 2]; low SNR of EEG signals represents the most critical acquisition-related challenge we aim to address. We define several noise levels using SNR values of 20, 10, 5 dB (ranging from clean to noisy signals) and corrupt the input signal with these artificial noises. As shown in Table 2 below, NERVE demonstrates the superior robustness on these corrupted signals, achieving smallest accuracy drop and prediction entropy change compared to existing EEG foundation models. These analyses demonstrates the robustness of NERVE to complex and unstructured noises as well as physiological noises, emphasizing the practical relevance and generalizability of our work. More analysis results are provided in Section G.1 in the revised manuscript.
> >
> > [1] Braun, S., Neil, D., & Liu, S. C. (2017, August). A curriculum learning method for improved noise robustness in automatic speech recognition. In 2017 25th European Signal Processing Conference (EUSIPCO) (pp. 548-552). IEEE.
> >
> > [2] Galeotti, L., & Scully, C. G. (2018). A method to extract realistic artifacts from electrocardiogram recordings for robust algorithm testing. Journal of electrocardiology, 51(6), S56-S60.
> >
> > (Table 2) Noise robustness analysis on corrupted EEG samples from High-Gamma and HCI-Tagging Emotion
> >
> > |High_Gamma|  | $\triangle$ Acc |  |  | \|$\triangle$ Entropy\| | |HCI-Tagging Emotion|  | $\triangle$ AUROC |  |  | \|$\triangle$ Entropy\| | |
> > |:---:|:---:|:---:|:---:|:---:|:---:|:---:|:---:|:---:|:---:|:---:|:---:|:---:|:---:|
> > ||LaBraM|CBraMod|NERVE|LaBraM|CBraMod|NERVE||LaBraM|CBraMod|NERVE|LaBraM|CBraMod|NERVE|
> > |SNR 5|-0.676|-0.302|**-0.167**|1.167|0.569|**0.262**|SNR 5|-0.046|-0.046|**-0.033**|0.156|0.075|**0.003**|
> > |SNR 10|-0.437|-0.124|**-0.042**|0.596|0.148|**0.045**|SNR 10|-0.051|**-0.015**|**-0.015**|0.142|0.035|**0.002**|
> > |SNR 20|-0.061|-0.009|**0.000**|0.065|0.011|**0.001**|SNR 20|-0.072|**-0.001**|-0.002|0.101|0.007|**0.001**|

---

> ### Author Response · Authors · 2025-11-21
> **Response to Reviewer c7uQ (4)**
>
> > [W2] While the core contribution of this work lies in proposing a noise-robust foundation model, the evaluation of noise robustness is insufficient. The authors have conducted comparative anti-noise experiments only on a limited number of datasets (Fig. 4, Fig. 7). Moreover, the experimental results indicate that the differences between NERVE and the baseline models on EMG, EOG, and environmental noise are marginal. The Gaussian condition, under which NERVE demonstrates its best noise robustness, is less common in practical scenarios, as most EEG signals undergo preprocessing—thereby diminishing the practical impact of NERVE’s contribution. Additionally, the authors should provide visual examples under varying noise injection conditions to better illustrate model behavior.
>
> 2. Moreover, the experimental results indicate that the differences between NERVE and the baseline models on EMG, EOG, and environmental noise are marginal. The Gaussian condition, under which NERVE demonstrates its best noise robustness, is less common in practical scenarios, as most EEG signals undergo preprocessing—thereby diminishing the practical impact of NERVE’s contribution.
>
> - Thank you for your insightful comment. We acknowledge the point that the performance differences between NERVE and the baseline models on EMG, EOG, and environmental noise conditions might appear marginal. The High-Gamma dataset is a relatively easy dataset where most baselines already demonstrate high performance. Consequently, the designed noise simulation may not have been strong enough to induce a significant degradation in performance and stability across all noise types. However, as demonstrated in Figure 9 in the revised manuscript, the gap between NERVE and other baselines is not marginal and represents a meaningful improvement.
>
> - While representative noise artifacts are often filtered through rule-based filtering techniques, residual noise inevitably remains in the processed signals [3, 4]. Because of that, the robustness to Gaussian noise is especially significant. Gaussian noise is statistically independent and corrupts all frequency bands, making its complete removal extremely difficult [5]. Baselines exhibited significant degradation in performance and prediction stability under the Gaussian condition. This indicates that residual Gaussian noise significantly hinders the reliable prediction of an EEG foundation model. Gaussian noise is the most commonly observed, unstructured pattern in many BCI tasks. An EEG foundation model must guarantee superior robustness to this fundamental noise type to ensure its practical utility. Therefore, our experimental results demonstrating superior robustness to the Gaussian condition critically highlight the practical contribution of NERVE.
>
> [3] Darren Tanner, Kara Morgan-Short, and Steven J Luck. How inappropriate high-pass filters can produce artifactual effects and incorrect conclusions in erp studies of language and cognition. Psychophysiology, 52(8):997–1009, 2015.
>
> [4] Whitham, E. M. et al. (2007). Scalp electrical recording during paralysis: quantitative evidence that EEG frequencies above 20 Hz are contaminated by EMG. Clinical neurophysiology : official journal of the International Federation of Clinical Neurophysiology, 118(8), 1877–1888. https://doi.org/10.1016/j.clinph.2007.04.027
>
> [5] Vaseghi, S. V. (2008). Advanced digital signal processing and noise reduction. John Wiley & Sons.

---

> ### Author Response · Authors · 2025-11-21
> **Response to Reviewer c7uQ (5)**
>
> > [W2] While the core contribution of this work lies in proposing a noise-robust foundation model, the evaluation of noise robustness is insufficient. The authors have conducted comparative anti-noise experiments only on a limited number of datasets (Fig. 4, Fig. 7). Moreover, the experimental results indicate that the differences between NERVE and the baseline models on EMG, EOG, and environmental noise are marginal. The Gaussian condition, under which NERVE demonstrates its best noise robustness, is less common in practical scenarios, as most EEG signals undergo preprocessing—thereby diminishing the practical impact of NERVE’s contribution. Additionally, the authors should provide visual examples under varying noise injection conditions to better illustrate model behavior.
>
> 3. Additionally, the authors should provide visual examples under varying noise injection conditions to better illustrate model behavior.
>
> - Thank you for your detailed feedback. Following your feedback, we provided a case visualization of the simulated noises used in the noise robustness analysis in Figure 8 of the revised manuscript. Also, to better illustrate the behavior of NERVE on various noise scenarios, we conducted a representation analysis. We computed the Centered Kernel Alignment (CKA) similarity between embeddings of raw EEG signals and signals corrupted with synthetic noises for NERVE and NERVE trained with standard neural tokenizer (not noise-robust variant trained without noise augmentation), which we term as NERVE (standard). Table 3 below shows the representation analysis results on High-Gamma and TUSL. NERVE demonstrated higher CKA similarity between raw and corrupted EEG signals on most noise conditions. This result indicates that NERVE produces stable representations in the presence of noise artifacts, which accounts for the superior robustness reported in the noise robustness analysis. Through noise augmentation and denoising temporal-spectral prediction, the noise-robust neural tokenizer learns EEG codebooks that are resilient to noise, thereby providing a prediction target containing high-level semantic information for NERVE pre-training. Additional analyses are provided in Section I.2 in the revised manuscript.
>
> (Table 3) Analysis of CKA similarity between embeddings of raw and noise-corrupted EEG signals
>
> |Noises||High-Gamma&emsp;&emsp;||TUSL&emsp;&emsp;&emsp;&emsp;&emsp;&emsp;&emsp;&emsp;&emsp;&emsp;|
> |:---:|:---:|:---:|:---:|:---:|
> ||NERVE|NERVE (standard)|NERVE| NERVE (standard)|
> |EOG|**0.9986** $\pm$ 0.0002|0.9941 $\pm$ 0.0004|**0.9980** $\pm$ 0.0008|0.9956 $\pm$ 0.0012|
> |EMG|**0.9996** $\pm$ 0.0001|0.9989 $\pm$ 0.0001|**0.9987** $\pm$ 0.0002|0.9967 $\pm$ 0.0006|
> |Environment|**0.9999** $\pm$ 0.0000|**0.9999** $\pm$ 0.0001|**0.9998** $\pm$ 0.0000|0.9993 $\pm$ 0.0002|
> |Gaussian|**0.7052** $\pm$ 0.0051|0.1395 $\pm$ 0.0019|0.7778 $\pm$ 0.0125|**0.8740** $\pm$ 0.0145|
> |SNR: 5dB|**0.6536** $\pm$ 0.0162|0.1323 $\pm$ 0.0048|**0.8644** $\pm$ 0.0089|0.8362 $\pm$ 0.0227|
> |SNR: 10dB|**0.9069** $\pm$ 0.0038|0.2416 $\pm$ 0.0052|**0.9558** $\pm$ 0.0061|0.9385 $\pm$ 0.0123|

---

> ### Author Response · Authors · 2025-11-21
> **Response to Reviewer c7uQ (6)**
>
> > [W3] The authors present extensive experiments to demonstrate the stability of NERVE’s performance in intra-subject settings. However, in my own opinion, this emphasis appears misaligned with the intended role of a foundation model for EEG. Such models are expected to prioritize generalization across diverse scenarios, tasks, and multicenter environments, rather than focusing on individual-specific adaptations. Robustness in both intra-subject and inter-subject contexts has already been extensively studied in dedicated literature.
>
> - Thank you for your valuable feedback. Foundation models aim to learn a universal and transferable representation space from large-scale data, generalizing effectively across diverse scenarios, datasets, and tasks. To achieve this in the EEG domain, a foundation model must be seamlessly generalizable to unseen subjects without requiring individual-specific adaptation, since EEG signals exhibit substantial variability both within and across subjects. Consequently, robustness to intra- and inter-subject variabilities is essential. These variabilities represent the significant hurdles to EEG generalization; thus, models that effectively mitigate them inherently possess superior generalization capacity [1].
>
> - While existing EEG foundation models have primarily addressed format mismatches across datasets, they often overlook robustness to intra- and inter-subject variabilities, which degrades their generalization performance.
> In our visualization analysis on pre-trained embeddings (Figure 16 in the revised manuscript), existing foundation models failed to distinguish embeddings between subjects. To address this gap, we propose NERVE, a variability-robust EEG foundation model that also tackles acquisition-related challenges such as low signal-to-noise ratios and spatial dependencies between electrodes. Although subject-wise robustness has been extensively studied in the BCI literature, it has not been explicitly addressed within the context of EEG foundation models. To the best of our knowledge, NERVE is the first EEG foundation model to specifically enhance robustness against these variabilities.
>
> [1] Cheng, J. Y., Goh, H., Dogrusoz, K., Tuzel, O., & Azemi, E. (2020). Subject-aware contrastive learning for biosignals. arXiv preprint arXiv:2007.04871.

---

> ### Author Response · Authors · 2025-11-21
> **Response to Reviewer c7uQ (7)**
>
> > [Q1] How did the authors verify that the Neural Tokenizer had been adequately trained for noise robustness prior to the masked reconstruction pre-training?
>
> - Thank you for your thorough comment. Throughout the training process, we closely monitored the loss trajectories for the denoising temporal-spectral prediction. The steady decrease and eventual convergence of both temporal reconstruction and normalized frequency prediction losses indicate that the neural tokenizer effectively learns to denoise injected artifacts and achieve noise robustness. As illustrated in Figure 6 of the manuscript, both losses exhibit a steady decline and converge by the final epoch of training.
>
> > [Q2] Could the authors provide an ablation study on the Neural Tokenizer? Since the noise robustness of NERVE appears to stem primarily from the pre-trained Neural Tokenizer, would the anti-noise performance deteriorate significantly if this component were removed?
>
> Thank you for the valuable suggestion. Following your recommendation, we compared NERVE trained with the noise-robust neural tokenizer against NERVE trained with a standard neural tokenizer (without noise augmentation), which we term as NERVE (standard). As shown in Table 4, NERVE utilizing the noise-robust neural tokenizer outperforms the model trained without noise augmentation on SEED-V and TUSL. Moreover, it exhibits superior robustness to noise, characterized by small accuracy degradation and predictive entropy changes (see Table 5). The ablation study underscores that noise-robust tokenizer training is essential for achieving generalizability and robustness. More experimental results are presented in Section I.1 in the Appendix of the revised manuscript.
>
> (Table 4) The ablation results on noise-robust training of neural-tokenizer
>
> |Methods|  |SEED-V| | |TUSL| |
> |:---:|:---:|:---:|:---:|:---:|:---:|:---:|
> ||Balanced Acc|Cohen's Kappa|Weighted F1|Balanced Acc|Cohen's Kappa|Weighted F1|
> |NERVE|**0.3788** $\pm$ 0.0031|**0.2305** $\pm$ 0.0034|**0.3908** $\pm$ 0.0028|**0.7000** $\pm$ 0.0282|**0.5327** $\pm$ 0.0411|**0.6507** $\pm$ 0.0329|
> |NERVE (standard)|0.3764 $\pm$ 0.0014|0.2291 $\pm$ 0.0016|0.3096 $\pm$ 0.0012|0.6050 $\pm$ 0.0452|0.3748 $\pm$ 0.0736|0.5742 $\pm$ 0.0320|
>
> (Table 5) Analysis on robustness to noises of NERVE trained with noise-robust and standard neural tokenizers
>
> |High_Gamma||$\triangle$ Acc &emsp;&emsp;&emsp;&emsp;&emsp;|| \|$\triangle$ Entropy\|&emsp;&emsp;&emsp; | DEAP||$\triangle$ Acc &emsp;&emsp;&emsp;&emsp;&emsp;||\|$\triangle$ Entropy\|&emsp;&emsp;&emsp;|
> |:---:|:---:|:---:|:---:|:---:|:---:|:---:|:---:|:---:|:---:|
> |Noise|NERVE|NERVE (standard)|NERVE|NERVE (standard)|Noise|NERVE|NERVE (standard)|NERVE|NERVE (standard)|
> |EMG|-0.014|**-0.011**|**0.014**|0.017|EMG|**0.000**|-0.001|**0.001**|0.002|
> |EOG|**0.000**|-0.017|**0.001**|0.015|EOG|**-0.001**|-0.002|**0.006**|0.008|
> |Environment|**0.000**|-0.011|**0.001**|0.017|Environment|**0.000**|**0.000**|**0.001**|**0.001**|
> |Gaussian|**-0.131**|-0.147|**0.276**|0.345|Gaussian|**-0.025**|-0.029|**0.035**|0.061|
> |SNR 5|-0.175|**-0.169**|**0.374**|0.394|SNR 5|**-0.019**|-0.026|**0.030**|0.060|
> |SNR 10|**-0.003**|-0.083|**0.040**|0.188|SNR 10|**-0.008**|-0.028|**0.020**|0.048|
> |SNR 20|**-0.001**|-0.005|**0.001**|0.0015|SNR 20|**-0.002**|-0.007|**0.005**|0.012|

---

> > ### Comment · Reviewer_c7uQ · 2025-11-24
> >
> > I appreciate the authors' detailed response, and most of my concerns have been addressed. After carefully evaluating the authors' replies and the revised manuscript, I have decided to maintain my initial positive score and update the confidence to 4.

---

> > > ### Author Response · Authors · 2025-11-24
> > >
> > > We appreciate your positive response. We are glad to have addressed most of your concerns and would be happy to engage further if there are additional questions.
> > >
> > > Following other reviewers’ feedback, we have also included **formal connection between EPA attention and criss-cross attention in CBraMod** and demonstrated that EPA transformer is a sequential generalization of criss-cross transformer (Section F). We have also added **interpretability analysis for channel embeddings and class topology comparison** (Section H). **Additional ablation studies** were conducted to further validate the effectiveness of EPA attention (Section I.4 and I.5). Lastly, we have **increased the number of repetition for evaluation** to strengthen the reliability of experimental results.
> > >
> > > We hope that these comprehensive revisions, which reflect our dedication to addressing every point, now provide the necessary clarity and technical detail for you to reconsider the overall score.
> > >
> > > Once again, we sincerely thank you for your valuable and insightful feedback.
> > >
> > > Best regards,
> > >
> > > The Authors

---

> > > > ### Author Response · Authors · 2025-11-30
> > > >
> > > > Dear Reviewer c7uQ,
> > > >
> > > > We appreciate your positive comments and insightful feedback, which led to constructive discussions and meaningful improvements in the revised manuscript.
> > > >
> > > > Thanks to your feedback, we were able to **enhance the presentation of our technical contribution** (W1), **demonstrate the noise robustness of NERVE** through additional robustness analyses, model behavior analysis on noisy inputs (W2, Q2). Lastly, we were able to **clarify the necessity of variability robustness for foundation model** (W3).
> > > >
> > > > We sincerely thank you for helping us strengthen the technical contribution of our work in various aspects. We have uploaded our meta responses for this rebuttal phase, summarizing the key improvements resulting from our discussions with you and other reviewers. Please review our meta responses, and we would be happy to engage further should you have additional questions.
> > > >
> > > > Sincerely,
> > > >
> > > > The authors

---

### Official Review · Reviewer_ejCD · 2025-10-29

**Soundness:** 2
**Presentation:** 2
**Contribution:** 3
**Rating:** 4
**Confidence:** 5

**Summary:**

This paper highlights the importance of robustness to noise and intra-subject variability in EEG foundation models. To address these challenges, the authors designed specialized modules—such as the EAP and the noise-robust tokenizer—as well as tailored learning objectives, including masked codebook reconstruction with KoLeo regularization, to enhance model robustness. Their robustness analysis reveals that existing EEG foundation models often produce unstable representations for the same class and struggle to disentangle subject-specific from class-specific information. In contrast, the proposed approach demonstrates improved stability and resilience to variability. Overall, the paper raises important awareness of the diverse sources of noise, variability, and artifacts that EEG foundation models must effectively account for.

**Strengths:**

1. The authors rigorously quantified noise and variability (both inter- and intra-subject) and evaluated the proposed models with corresponding analyses. This provides evidence of NERVE’s robustness to noise and individual differences. Such analysis is novel in the literature of EEG foundation models.
2. The systematic comparison of various attention mechanisms in Section F.3 provides valuable insights for future research on the architectural design of transformer blocks in EEG modeling.
3. The proposal to adapt the denoising autoencoder framework for EEG foundation models is a novel and valuable contribution. It highlights an important issue in current practice—the inadequacy of using mean squared error (MSE) loss for training on noisy data—and could raise broader awareness within the community regarding the need for more robust training objectives.
4. The curated pre-training corpus, comprising 27 public datasets encompassing EEG recordings from a wide range of task paradigms, represents a substantial advancement in data diversity and scale. It is notably more comprehensive than the corpora employed in previous studies, providing a stronger foundation for pre-training and evaluation.

**Weaknesses:**

1. In Section 3.2, comparisons with previous models are conducted using three random seeds. Although this number aligns with some prior studies, increasing the number of random repetitions to five or more would strengthen the statistical reliability of the results and more convincingly demonstrate the superiority of NERVE.
2. Line 407 of Section 3.3 attributes the superior performance on the High-Gamma dataset to the EPA attention mechanism; however, this claim is not substantiated by appropriate ablation studies. Moreover, several foundation models that do not incorporate EPA attention (e.g., LaBraM, CBraMod, NeuroLM) also exhibit comparatively strong performance on this dataset, further questioning the causal link between EPA attention and the observed results.
3. The effectiveness of KoLeo in enhancing robustness to variability is not consistently assessed across all downstream datasets. Moreover, as shown in Figure 8, its impact appears to vary substantially between datasets. A more detailed analysis explaining this phenomenon—such as identifying which characteristics of the datasets (e.g., task type, signal-to-noise ratio, temporal complexity, or inter-subject variability) contribute to these differences—would provide deeper insight into when and why KoLeo is most beneficial.
4. The position router introduced in Line 208 of Section 2.1 is claimed to model cortical regions; however, no explicit physical prior is incorporated into its design. Since both the router $P$ and weight matrices $W^Q$, $W^K$ are both learnable linear operators, their product during back-propagation is equivalent to a single transformation matrix. Consequently, the Q-K product defined in equation (4) can be reformulated to $(HE)(HE)^T$ where $E \in \mathbb{R}^{C \times R}$. Given that $R = 6$ in this paper -- typically smaller than the number of EEG channels $C$ -- the router usually smaller than number of EEG channels $C$, this router essentially performs a dimension reduction operation analogous to the Linformer (Wang et al, 2020), compressing spatial dimension to reduce computational cost. Therefore, apart from converting the parallel spatial-temporal attention blocks into sequential order and lowering attention complexity, the proposed EPA Transformer is not fundamentally novel mechanism compared to the Criss-cross attention method proposed in CBraMod. As such, the claimed functionality of this module as a cortical-region modeler appears unsubstantiated.
5. Artifacts in scalp EEG—particularly biological artifacts—are inherently non-stationary stochastic processes that contaminate the signal across spatial, temporal, and spectral domains (see Figure 2 in Agounad, 2025). In the pre-training stage, the use of Gaussian noise as data augmentation represents an overly simplified assumption of the complex noise and artifact characteristics typically encountered in EEG recordings. Moreover, in the robustness-to-noise analysis, additive Gaussian noise is injected into specific frequency bands. This approach effectively introduces continuous sinusoidal oscillations in the time domain, which does not accurately reflect the characteristics of EMG and EOG artifacts that the study aims to simulate. Consequently, the conclusions drawn from this robustness analysis may be limited in their validity and generalisability to real-world EEG noise conditions.

Wang, S., Li, B. Z., Khabsa, M., Fang, H. & Ma, H. Linformer: Self-Attention with Linear Complexity. arXiv preprint arXiv:2006.04768 (2020).

Agounad, S., Tarahi, O., Moufassih, M. et al. Advanced Signal Processing and Machine/Deep Learning Approaches on a Preprocessing Block for EEG Artifact Removal: A Comprehensive Review. Circuits Syst Signal Process 44, 3112–3160 (2025). https://doi.org/10.1007/s00034-024-02936-3

**Questions:**

1. The selected fine-tuning datasets differ substantially in trial duration and temporal dynamics. For instance, the DEAP dataset’s video-watching task involves long, continuous time courses, whereas the High Gamma dataset’s movement task exhibits rapidly evolving dynamics, often within sub-second timescales. How does NERVE accommodate these diverse temporal characteristics using a unified patch length of 1 second (200 samples)?
2. The phase information in higher-frequency EEG bands is typically highly sensitive to temporal jitters and tends to become unreliable for frequencies above approximately 6 Hz. Could the authors elaborate on why capturing phase information is claimed to be important across all frequency bands, as stated in Lines 258–263 of Section 2.2? Furthermore, the temporal reconstruction loss and frequency reconstruction loss defined in Equation (9) are strongly correlated, since the frequency-domain representations are derived from the temporal-domain signals via the Discrete Fourier Transform (DFT). Please clarify the rationale for including both loss terms—specifically, what distinct information or complementary benefits each contributes to model optimisation.

Some imprecise parts observed in the paper:
1. The cited material (Miah et al., 2021) in Line 96 does not appear to focus on modeling spatial structure. It would be helpful if the authors could clarify how this reference supports the stated argument regarding the importance of incorporating spatial information. Additionally, Line 81 of Section 1 mentions that “quantized embeddings do not necessarily ensure robustness to noise” as a limitation of LaBraM; however, this specific limitation is not discussed in the cited source (Hu et al., 2023). Please provide further justification or an alternative reference that substantiates this claim.
2. Line 80 mentions “low SNR makes reconstruction challenging”, a better expression would be making reconstruction “inappropriate” as the model learns to reconstruct noises as well as semantic information.
3. The dot operation in Equations (4) and (5) could mislead the readers, as it is often preserved for dot product instead of matrix multiplications.
4. The loss term in Equation (10) seems to be probability logits rather than neural code reconstruction loss.

---

> ### Author Response · Authors · 2025-11-21
> **Response to Reviewer ejCD (1)**
>
> > [w1] In Section 3.2, comparisons with previous models are conducted using three random seeds. Although this number aligns with some prior studies, increasing the number of random repetitions to five or more would strengthen the statistical reliability of the results and more convincingly demonstrate the superiority of NERVE.
>
> - We appreciate your constructive suggestion. Following your advice, we have increased the number of random seeds from three to five to ensure experimental robustness. The updated results are presented in Tables 3, 4, 10, and 11 of the revised manuscript. As shown, NERVE consistently outperforms the baselines, aligning with our initial observations.
>
> - We are currently in the process of extending all remaining experiments from three to five seeds. However, due to limited computational resources, this process is taking some time. You have confirmed that all evaluation results in the main paper and the Appendix have been replaced in the current revised manuscript, except for Table 16 and Table 18. We are prioritizing remaining updates and will revise the manuscript with the new results as soon as each experiment is completed.

---

> ### Author Response · Authors · 2025-11-21
> **Response to Reviewer ejCD (2)**
>
> > [W2] Line 407 of Section 3.3 attributes the superior performance on the High-Gamma dataset to the EPA attention mechanism; however, this claim is not substantiated by appropriate ablation studies. Moreover, several foundation models that do not incorporate EPA attention (e.g., LaBraM, CBraMod, NeuroLM) also exhibit comparatively strong performance on this dataset, further questioning the causal link between EPA attention and the observed results.
>
> - Thank you for your thorough feedback. While other foundation models demonstrate competitive performance on the High-Gamma dataset, NERVE further improves performance by approximately 1% over the second-best baseline with the lowest standard deviation. Furthermore, following your advice, we have conducted various analyses to substantiate the link between EPA attention and the observed superior performance.
>
> - First, we conducted a visualization analysis to compare the topography of the raw sample with class activation topography of NERVE on the High-Gamma dataset. The visualization results and the setup of visualization analysis are provided in Figure 13 and Section H.2 of the Appendix in the revised manuscript. We observed that electrodes associated with the left hand exhibit patterns relatively symmetric to those of the right hand, while both cortical regions are activated for the 'both feet' class. In contrast, minimal activation is observed during the resting state. Although NERVE's class activation topography is not identical to the raw EEG signals, its representations capture broadly similar spatial patterns, demonstrating distinct differences across classes.
>
> - Second, we compared the topography of raw samples from the pre-training datasets with the channel embedding topographies learned by different attention architectures. Specifically, we analyzed NERVE (using EPA attention) against variants equipped with temporal attention (attention across the time dimension), temporal-channel attention (sequential blocks of attention in time and channel dimensions), and criss-cross attention (the mechanism proposed by CBraMod). The visualization results are presented in Figure 15 of the revised manuscript. We observed that the channel embedding topography of NERVE with EPA attention resembles the raw signal topography more closely than those of other architectures. This indicates that EPA attention effectively captures the inherent topological structure of the electrodes.
>
> - Third, we visualized the inter-channel cosine similarity heatmap of NERVE embeddings using the BCIC2020-3 dataset from our pre-training corpus. As shown in Figure 12 of the revised manuscript, we observe high similarity among specific groups of electrodes, forming a distinct block-wise similarity structure. This observation underscores the necessity of region-wise channel modeling for EEG signals and supports the validity of EPA attention.
>
> - Lastly, we conducted an ablation study to evaluate various attention mechanisms within the NERVE backbone. We replaced EPA attention with temporal, temporal-channel, and criss-cross attention modules. As shown in Table 1 below, NERVE with EPA attention outperforms other attention-based architectures on the TUEV and DEAP datasets, highlighting the effectiveness of EPA attention for EEG modeling. Additional experimental results are provided in Table 17 and Section I.4 of the revised manuscript.
>
> (Table 1) The ablation results on attention mechanism for NERVE backbone.
> |||TUEV|||DEAP||
> |:---:|:---:|:---:|:---:|:---:|:---:|:---:|
> ||Acc|Kappa|F1|Acc|AUPRC|AUROC|
> |Temporal|0.5386 $\pm$ 0.0084|0.4570 $\pm$ 0.0110 | 0.7112 $\pm$ 0.0075 |0.4939 $\pm$ 0.0188|0.6223 $\pm$ 0.0088|0.4971 $\pm$ 0.0214|
> |Temporal-channel|0.5503 $\pm$ 0.0169|0.4753 $\pm$ 0.0043|0.7235 $\pm$ 0.0049|0.5086 $\pm$ 0.0075|0.6314 $\pm$ 0.0216|0.5044 $\pm$ 0.0277|
> |Criss-cross|0.5270 $\pm$ 0.0346|0.4615 $\pm$ 0.0587|0.7227 $\pm$ 0.0315|0.4967 $\pm$ 0.0081|0.6267 $\pm$ 0.0145|0.4982 $\pm$ 0.0217|
> |EPA|**0.5595** $\pm$ 0.0106|**0.4810** $\pm$ 0.0146|**0.7249** $\pm$ 0.0095|**0.5167** $\pm$ 0.0131|**0.6553** $\pm$ 0.0156|**0.5406** $\pm$ 0.0267|

---

> ### Author Response · Authors · 2025-11-21
> **Response to Reviewer ejCD (3)**
>
> > [W3] The effectiveness of KoLeo in enhancing robustness to variability is not consistently assessed across all downstream datasets. Moreover, as shown in Figure 8, its impact appears to vary substantially between datasets. A more detailed analysis explaining this phenomenon—such as identifying which characteristics of the datasets (e.g., task type, signal-to-noise ratio, temporal complexity, or inter-subject variability) contribute to these differences—would provide deeper insight into when and why KoLeo is most beneficial.
>
> - We appreciate your insightful comment. First, among the eight downstream datasets, the analysis on robustness to variability was conducted on HCI-Tagging Emotion, TUSL, High-Gamma, and SEED-V. The remaining four datasets were unsuitable for this evaluation for the following reasons: TUEV lacks subject identifiers; SEED-VIG is a regression dataset, making it impossible to calculate the logits; and on DEAP and BCI-NER-Challenge, the LaBraM baseline exhibited degenerate predictions across certain seeds, which made reliable variability measurement impossible. We have explicitly stated these experimental constraints in the revised manuscript (see Section G.2 in the Appendix).
>
> - On Figure 8 (Figure 10 in the revised manuscript), LaBraM and NERVE show the similar intra-subject distance on High-Gamma. This is attributed to the small number of test subjects on High-Gamma, which is two. The small number of subjects accidently suggests comparable robustness to intra-subject variability, but NERVE exhibits superior robustness to intra-subject variability (the lowest intra-subject distance) on other datasets over three test subjects.
>
> - We extended our ablation study on variability-robust pre-training to include additional downstream datasets (see Table 14 and 15 in the revised manuscript). As shown in Table 2 below, we compared NERVE with and without KoLeo loss on High-Gamma and SEED-VIG. On High-Gamma, the effect of variability-robust pre-training was marginal. This aligns with the findings in Figure 8, which indicate that robustness to intra-subject variability is less critical given the dataset's limited number of test subjects (only two). For SEED-VIG, which is a regression dataset, NERVE trained without KoLeo loss outperforms the standard NERVE. This suggests that enhancing the uniformity of EEG samples in the embedding space does not necessarily enhance downstream performance for regression tasks. We have also included this insight in the revised manuscript (see Section I.3 in the Appendix).
>
> (Table 2) The ablation results on variability-robust pre-training (KoLeo loss)
>
> |||High-Gamma|||SEED-VIG||
> |:---:|:---:|:---:|:---:|:---:|:---:|:---:|
> ||Acc|Kappa|F1|Corr|R2 score|RMSE|
> |NERVE|**0.9906** $\pm$ 0.0011|**0.9875** $\pm$ 0.0015|**0.9906** $\pm$ 0.0011|0.4158 $\pm$ 0.0177|0.1591 $\pm$ 0.0142|0.2865 $\pm$ 0.0024|
> |NERVE (w/o KoLeo)|0.9904 $\pm$ 0.0013|0.9871 $\pm$ 0.0017|0.9904 $\pm$ 0.0013|**0.4427** $\pm$ 0.0198|**0.1694** $\pm$ 0.0251|**0.2847** $\pm$ 0.0044|

---

> ### Author Response · Authors · 2025-11-21
> **Response to Reviewer ejCD (4)**
>
> > [W4] The position router introduced in Line 208 of Section 2.1 is claimed to model cortical regions; however, no explicit physical prior is incorporated into its design. Since both the router $P$ and weight matrices $W^Q$, $W^K$  are both learnable linear operators, their product during back-propagation is equivalent to a single transformation matrix. Consequently, the Q-K product defined in equation (4) can be reformulated to $(HE)(HE)^T$ where $E\in\mathbb{R}^{C \times R}$ . Given that $R$ in this paper -- typically smaller than the number of EEG channels $C$ -- the router usually smaller than number of EEG channels $C$, this router essentially performs a dimension reduction operation analogous to the Linformer (Wang et al, 2020), compressing spatial dimension to reduce computational cost. Therefore, apart from converting the parallel spatial-temporal attention blocks into sequential order and lowering attention complexity, the proposed EPA Transformer is not fundamentally novel mechanism compared to the Criss-cross attention method proposed in CBraMod. As such, the claimed functionality of this module as a cortical-region modeler appears unsubstantiated.
>
> 1) The position router introduced in Line 208 of Section 2.1 is claimed to model cortical regions; however, no explicit physical prior is incorporated into its design.
>
> - We appreciate your valuable feedback. We acknowledge that explicit physical priors are not strictly incorporated into the position router. By refraining from imposing constraints, NERVE aims to learn signal-driven soft cortical groupings. While electrodes are placed on physical brain regions, neural interactions are not exclusively bound by anatomical proximity due to volume conduction and functional networks [1, 2]. Thus, without structural regularization imposed on the position router, NERVE gains the flexibility to learn latent spatial patterns and interactions that are optimal for the task. We recognize that while removing explicit anatomical priors enhances flexibility, it may pose challenges regarding interpretability. We have added a discussion on this trade-off to the Limitations and Future Works section (refer to Section J in the revised manuscript).
>
> [1] Winter, W. R., Nunez, P. L., Ding, J., & Srinivasan, R. (2007). Comparison of the effect of volume conduction on EEG coherence with the effect of field spread on MEG coherence. Statistics in medicine, 26(21), 3946-3957.
>
> [2] Hipp, J. F., Hawellek, D. J., Corbetta, M., Siegel, M., & Engel, A. K. (2012). Large-scale cortical correlation structure of spontaneous oscillatory activity. Nature neuroscience, 15(6), 884-890.
>
> 2) Given that $R$ in this paper -- typically smaller than the number of EEG channels $C$ -- the router usually smaller than number of EEG channels $C$, this router essentially performs a dimension reduction operation analogous to the Linformer (Wang et al, 2020), compressing spatial dimension to reduce computational cost.
>
> - While EPA attention shares similarities with Linformer attention in utilizing linear layers, their objectives and detailed mechanisms are fundamentally distinct (see Table 3 below). The primary objective of Linformer is to utilize linear layers to reduce the computational complexity of the self-attention mechanism. In contrast, EPA attention defines the position router as learnable parameters to explicitly model the interaction between EEG channels and their spatial positions. As a result, although EPA attention structurally resembles an attention operation with linear layers, its fundamental purpose is spatial modeling rather than complexity reduction. In Linformer attention, linear layers are applied to the key and value embeddings to reduce dimensionality. In contrast, EPA attention applies linear transformations (via the position router) to the query and key embeddings. Due to this structural difference, Linformer operates with $O(C)$ complexity, whereas EPA attention maintains an $O(C^2)$ complexity.
>
> (Table 3) Comparison of Linformer attention and EPA attention
>
> ||Linformer attention|EPA attention|
> |:---:|:---:|:---:|
> |Objective|Complexity reduction|Electrode topology modeling|
> |Linear layer location|Key, value | Query, key|
> |Complexity| $O(C)$ | $O(C^2)$ |

---

> ### Author Response · Authors · 2025-11-21
> **Response to Reviewer ejCD (5)**
>
> > [W4] The position router introduced in Line 208 of Section 2.1 is claimed to model cortical regions; however, no explicit physical prior is incorporated into its design. Since both the router $P$ and weight matrices $W^Q$, $W^K$  are both learnable linear operators, their product during back-propagation is equivalent to a single transformation matrix. Consequently, the Q-K product defined in equation (4) can be reformulated to $(HE)(HE)^T$ where $E\in\mathbb{R}^{C \times R}$ . Given that $R$ in this paper -- typically smaller than the number of EEG channels $C$ -- the router usually smaller than number of EEG channels $C$, this router essentially performs a dimension reduction operation analogous to the Linformer (Wang et al, 2020), compressing spatial dimension to reduce computational cost. Therefore, apart from converting the parallel spatial-temporal attention blocks into sequential order and lowering attention complexity, the proposed EPA Transformer is not fundamentally novel mechanism compared to the Criss-cross attention method proposed in CBraMod. As such, the claimed functionality of this module as a cortical-region modeler appears unsubstantiated.
>
> 3) Therefore, apart from converting the parallel spatial-temporal attention blocks into sequential order and lowering attention complexity, the proposed EPA Transformer is not fundamentally novel mechanism compared to the Criss-cross attention method proposed in CBraMod.
>
> - EPA transformer is not merely a sequential adaptation of criss-cross transformer; rather, it constitutes a sequential formulation of 'generalized' criss-cross transformer. We denote $S_j=(Q_H M_j K_H^T)/\sqrt{d}$ and $M_j=Q_P^T K_P=(PW_Q)^T(PW_K)\in\mathbb{R}^{d\times d}$. If $M_j\equiv I$, EPA transformer is reduced to the sequential conversion of criss-cross transformer. Thus, criss-cross attention can be regarded as a special case of EPA attention.
>
> - Furthermore, given that $PW_Q \in \mathbb{R}^{d\times R}$, the rank of $M_j$ is inherently bounded by $R$. Since $rank(M_j)\leq R$, so EPA attention constrains the channel-wise score matrix $S_j=(Q_H M_j K_H^T)/\sqrt{d}$ to lie in a low-dimensional, router-defined subspace. This low-rank and topology-aligned regularization encourages robust electrode–region–electrode dependencies while suppressing spurious correlations among channels.
>
>
> 4) Since both the router $P$ and weight matrices $W^Q$, $W^K$ are both learnable linear operators, their product during back-propagation is equivalent to a single transformation matrix.
>
> $$\frac{\partial L}{\partial S_j} = \frac{\partial L}{\partial Q_H} \left(\frac{\partial S_j}{\partial Q_H}\right)^{-1}=\frac{\partial L}{\partial Q_H}(M_jK_H^T)^{-1}=\frac{\partial L}{\partial Q_H}(K_H)^{-1} (M_j^T)^{-1}$$
>
> - When deriving the gradient of the loss function $L$ with respect to the attention score $S_j$ (see the equation above), $M_j^T$ acts as a multiplier to the gradient of the standard channel attention. Given that $rank(M_j)\leq R$, this operation constrains the gradient flow, imposing an implicit regularization effect on the learned parameters.
>
> In summary, the EPA transformer leverages the position router to calculate attention scores, thereby learning electrode-region-electrode dependencies and signal-driven soft cortical groupings. While computationally simple because it operates with linear layers, it effectively generalizes the criss-cross transformer and successfully models EEG signal topographies (please refer to the response on W2). We sincerely appreciate your constructive feedback, as it has enabled us to solidify the methodological contribution of our work. We have incorporated this discussion into Section F of the revised manuscript.

---

> ### Author Response · Authors · 2025-11-21
> **Response to Reviewer ejCD (6)**
>
> > [W5] Artifacts in scalp EEG—particularly biological artifacts—are inherently non-stationary stochastic processes that contaminate the signal across spatial, temporal, and spectral domains (see Figure 2 in Agounad, 2025). In the pre-training stage, the use of Gaussian noise as data augmentation represents an overly simplified assumption of the complex noise and artifact characteristics typically encountered in EEG recordings. Moreover, in the robustness-to-noise analysis, additive Gaussian noise is injected into specific frequency bands. This approach effectively introduces continuous sinusoidal oscillations in the time domain, which does not accurately reflect the characteristics of EMG and EOG artifacts that the study aims to simulate. Consequently, the conclusions drawn from this robustness analysis may be limited in their validity and generalisability to real-world EEG noise conditions.
>
> 2) Moreover, in the robustness-to-noise analysis, additive Gaussian noise is injected into specific frequency bands. This approach effectively introduces continuous sinusoidal oscillations in the time domain, which does not accurately reflect the characteristics of EMG and EOG artifacts that the study aims to simulate. Consequently, the conclusions drawn from this robustness analysis may be limited in their validity and generalisability to real-world EEG noise conditions.
>
> - We appreciate your valuable comment. We acknowledge the reviewer's observation that injecting continuous Gaussian noise might not fully capture the non-Gaussian structure of real-world EMG and EOG artifacts. However, we adhered to the conventional methodology employed in the BCI literature for several reasons. Although the structural characteristics of EOG and EMG are not Gaussian, many prior studies on EEG artifact removal and noise robustness have simulated these physiological effects by injecting random white noise into their corresponding frequency bands [5, 6, 7]. For instance, these studies have simulated physiological artifacts, such as eye blinks and muscle activity, which are common disturbances during EEG signal acquisition, by injecting Gaussian noise into specific spectral ranges. This approach provides a standardized proxy for evaluating robustness within defined spectral ranges, allowing for a benchmark comparison with traditional BCI background methods. Therefore, while we recognize the structural simplification, our experimental setup adheres to this established methodology, thereby guaranteeing consistency with established practices in the field of EEG robustness.
>
> [5] Yong, X., Fatourechi, M., Ward, R. K., & Birch, G. E. (2012). Automatic artefact removal in a self-paced hybrid brain-computer interface system. Journal of neuroengineering and rehabilitation, 9(1), 50.
>
> [6] Zangeneh Soroush, M., Tahvilian, P., Nasirpour, M. H., Maghooli, K., Sadeghniiat-Haghighi, K., Vahid Harandi, S., ... & Jafarnia Dabanloo, N. (2022). EEG artifact removal using sub-space decomposition, nonlinear dynamics, stationary wavelet transform and machine learning algorithms. Frontiers in Physiology, 13, 910368.
>
> [7] Delorme, A., Sejnowski, T., & Makeig, S. (2007). Enhanced detection of artifacts in EEG data using higher-order statistics and independent component analysis. Neuroimage, 34(4), 1443-1449.
>
> - We deeply understand the concerns regarding the practical validity of noise robustness analysis. To better validate the robustness to complex and unstructured noises, we control the signal-to-noise ratio (SNR) of the input signal, which is defined as $SNR_{DB}=10log_{10}(\frac{{P_{signal}}}{P_{noise}})$, where the power $P=lim_{T\rightarrow \infty} \frac{1}{T} \int_{-\frac{T}{2}}^{\frac{T}{2}}s^2(t)dt$. SNR serves as the standard metric for quantifying signal quality and evaluating model robustness across varying noise levels [8, 9]; low SNR of EEG signals represents the most critical acquisition-related challenge we aim to address. We define several noise levels using SNR values of 20, 10, 5 dB (ranging from clean to noisy signals) and corrupt the input signal with these artificial noises. As shown in Table 4 below, NERVE demonstrates the superior robustness on these corrupted signals, achieving smallest accuracy drop and prediction entropy change compared to existing EEG foundation models. These analyses demonstrates the robustness of NERVE to complex and unstructured noises as well as physiological noises, emphasizing the practical relevance and generalizability of our work. More analysis results are provided in Section G.1 in the revised manuscript.

---

> ### Author Response · Authors · 2025-11-21
> **Response to Reviewer ejCD (7)**
>
> (Table 4) Noise robustness analysis on corrupted EEG samples from High-Gamma and HCI-Tagging Emotion
>
> |High_Gamma|  | $\triangle$ Acc |  |  | \|$\triangle$ Entropy\| | |HCI-Tagging Emotion|  | $\triangle$ AUROC |  |  | \|$\triangle$ Entropy\| | |
> |:---:|:---:|:---:|:---:|:---:|:---:|:---:|:---:|:---:|:---:|:---:|:---:|:---:|:---:|
> ||LaBraM|CBraMod|NERVE|LaBraM|CBraMod|NERVE||LaBraM|CBraMod|NERVE|LaBraM|CBraMod|NERVE|
> |SNR 5|-0.686|-0.381|**-0.208**|1.197|0.581|**0.245**|SNR 5|-0.058|-0.046|**-0.028**|0.066|0.075|**0.005**|
> |SNR 10|-0.444|-0.111|**-0.042**|0.608|0.133|**0.045**|SNR 10|-0.047|-0.015|**-0.014**|0.080|0.035|**0.003**|
> |SNR 20|-0.061|-0.006|**-0.001**|0.065|0.007|**0.001**|SNR 20|-0.039|0.001|**-0.002**|0.068|0.007|**0.001**|
>
> [8] Braun, S., Neil, D., & Liu, S. C. (2017, August). A curriculum learning method for improved noise robustness in automatic speech recognition. In 2017 25th European Signal Processing Conference (EUSIPCO) (pp. 548-552). IEEE.
>
> [9] Galeotti, L., & Scully, C. G. (2018). A method to extract realistic artifacts from electrocardiogram recordings for robust algorithm testing. Journal of electrocardiology, 51(6), S56-S60.

---

> ### Author Response · Authors · 2025-11-21
> **Response to Reviewer ejCD (8)**
>
> > [Q1] The selected fine-tuning datasets differ substantially in trial duration and temporal dynamics. For instance, the DEAP dataset’s video-watching task involves long, continuous time courses, whereas the High Gamma dataset’s movement task exhibits rapidly evolving dynamics, often within sub-second timescales. How does NERVE accommodate these diverse temporal characteristics using a unified patch length of 1 second (200 samples)?
>
> - Thank you for your in-depth comment. We consider 1 second to be the fundamental unit for many EEG tasks; thus, we suggest it is an ideal patch duration for ensuring the transferability of an EEG foundation model to various downstream tasks. This setting aligns with existing EEG foundation models [10, 11]. While the DEAP and High-Gamma datasets involve different temporal dynamics, both contain signals exceeding 1 second (1 minute for DEAP and 4 seconds for High-Gamma). Furthermore, since our EPA transformer layer consists of sequential temporal and channel blocks, NERVE effectively captures the dynamic temporal patterns of EEG samples through the temporal block.
>
> [10] Weibang Jiang, Liming Zhao, and Bao liang Lu. (2024). Large brain model for learning generic represen- tations with tremendous EEG data in BCI. In The Twelfth International Conference on Learning Representations,
>
> [11] Wang, J. et al., (2025). CBraMod: A Criss-Cross Brain Foundation Model for EEG Decoding. In The Thirteenth International Conference on Learning Representations.

---

> ### Author Response · Authors · 2025-11-21
> **Response to Reviewer ejCD (9)**
>
> > [Q2] The phase information in higher-frequency EEG bands is typically highly sensitive to temporal jitters and tends to become unreliable for frequencies above approximately 6 Hz. Could the authors elaborate on why capturing phase information is claimed to be important across all frequency bands, as stated in Lines 258–263 of Section 2.2? Furthermore, the temporal reconstruction loss and frequency reconstruction loss defined in Equation (9) are strongly correlated, since the frequency-domain representations are derived from the temporal-domain signals via the Discrete Fourier Transform (DFT). Please clarify the rationale for including both loss terms—specifically, what distinct information or complementary benefits each contributes to model optimisation.
>
> 1. Could the authors elaborate on why capturing phase information is claimed to be important across all frequency bands, as stated in Lines 258–263 of Section 2.2?
>
> - We appreciate your insightful feedback. While high-frequency bands are indeed sensitive to noise artifacts and can be unreliable for certain BCI tasks, they are frequently corrupted by physiological noises such as EMG (affecting beta and gamma bands, >13 Hz) and environmental noises like power line interference (50-60 Hz). To ensure robustness against such diverse artifacts, we argue that the model must capture phase information across all frequency bands to effectively distinguish and remove these noises. Furthermore, we employed Gaussian noise to train our noise-robust neural tokenizer. Since Gaussian noise distorts the entire spectrum, to accurately disentangle these artifacts, it is essential to predict the full frequency spectrum via denoising temporal-spectral prediction.
>
> 2. Furthermore, the temporal reconstruction loss and frequency reconstruction loss defined in Equation (9) are strongly correlated, since the frequency-domain representations are derived from the temporal-domain signals via the Discrete Fourier Transform (DFT). Please clarify the rationale for including both loss terms—specifically, what distinct information or complementary benefits each contributes to model optimisation.
>
> - While timestamps within a patch are highly correlated with their preceding values, temporal reconstruction via a linear head implicitly assumes timestamp-wise independence, thereby overlooking the inherent autocorrelation. To mitigate the bias arising from neglecting this autocorrelation, we apply a frequency prediction loss to the Fourier transform of the reconstructed patch. As the frequency-domain representation represents reconstructed patch signal as a linear combination of orthogonal frequency components, calculating loss in the frequency domain effectively circumvents the issue of time-domain autocorrelation [12]. By utilizing objective functions on both the reconstructed time-domain signals and their spectral representations, we mitigate the bias of temporal autocorrelation, thus enhancing the denoising efficacy. The validity of this formulation has been demonstrated in the context of time series forecasting [12], and similar studies have reported improved performance using dual-domain losses [13]. Furthermore, this approach reduces the parameter count for the frequency prediction head, thereby improving the computational efficiency of the neural tokenizer training.
>
> [12] Wang, H. et al. (2025) FreDF: Learning to Forecast in the Frequency Domain. In The Thirteenth International Conference on Learning Representations.
>
> [13] Li, Z. et al. (2025). FTMixer: Frequency and Time Domain Representations Fusion for Time Series Forecasting. IEEE Signal Processing Letters.

---

> ### Author Response · Authors · 2025-11-21
> **Response to Reviewer ejCD (10)**
>
> > [M1] The cited material (Miah et al., 2021) in Line 96 does not appear to focus on modeling spatial structure. It would be helpful if the authors could clarify how this reference supports the stated argument regarding the importance of incorporating spatial information. Additionally, Line 81 of Section 1 mentions that “quantized embeddings do not necessarily ensure robustness to noise” as a limitation of LaBraM; however, this specific limitation is not discussed in the cited source (Hu et al., 2023). Please provide further justification or an alternative reference that substantiates this claim.
>
> - Thank you for your detailed comment. We apologize for the oversight regarding the incorrect references. Following your feedback, we have replaced '(Miah et al., 2021)' in Line 96 with the correct reference advocating for topology-aware modeling in EEG signals [14]. Similarly, we have replaced '(Hu et al., 2023)' with the appropriate citation discussing the limited noise robustness of quantized models [15]. Please refer to Lines 83 and 97 in the revised manuscript.
>
> [14] Shen, C., & Namiki, A. (2025). A Topology-Aware Multiscale Feature Fusion Network for EEG-Based Motor Imagery Decoding. Knowledge-Based Systems, 114540.
>
> [15] Xiao, Y. et al. (2023). Robustmq: benchmarking robustness of quantized models. Visual Intelligence, 1(1), 30.
>
> > [M2] Line 80 mentions “low SNR makes reconstruction challenging”, a better expression would be making reconstruction “inappropriate” as the model learns to reconstruct noises as well as semantic information.
>
> - Thank you for your detailed comment. Following your advice, we have revised the sentence to read: 'but the low SNR of EEG makes it difficult to reconstruct the original signals accurately.' We believe this change enhances the clarity and presentation of our work. Please refer to line 80 in the revised manuscript.
>
> > [M3] The dot operation in Equations (4) and (5) could mislead the readers, as it is often preserved for dot product instead of matrix multiplications.
>
> - Thank you for your constructive comment. To improve the clarity of the EPA attention formulation, we have revised the notation by converting superscripts on patch embeddings to subscripts and removing explicit dot product symbols for matrix multiplication. For example, $H^{i,:}$ has been replaced with $H_{:,j}$ and Equation 5 is now expressed as $(H_{:,j}W^Q P^T_{:,j})(P_{:,j}{W^K}^T H^T_{:,j})$. Please refer to Section 2.2 of the revised manuscript for the improved formulation.
>
>
>
> > [M4] The loss term in Equation (10) seems to be probability logits rather than neural code reconstruction loss.
>
> - We appreciate your detailed comment. Given a masked EEG signal, the NERVE model is trained to predict the codebook indices of the masked patches derived from the codebook of a noise-robust neural tokenizer. Consequently, the objective for masked EEG modeling (Equation 10) is defined as the cross-entropy loss over these codebook indices.

---

> > ### Author Response · Authors · 2025-11-30
> >
> > Dear Reviewer ejCD,
> >
> > We appreciate your positive comments and insightful feedback, which led to constructive discussions and meaningful improvements in the revised manuscript.
> >
> > Thanks to your feedback, we were able to enhance the technical contributions of all our architectures: we could **clarify the validity of denoising temporal-spectral prediction and Gaussian noise augmentation for noise-robust training** (W5, Q1); we could **investigate the EPA attention formally and experimentally through the rigorous formulation and ablation studies** (W2, W4); we could **investigate the effectiveness of variability-robust learning** across different task types and test subjects (W3). Also, we have **strengthened the statistical reliability of the results and enhanced the presentation of the work** (W1, M1-4).
> >
> > We sincerely thank you for helping us strengthen the technical contribution and novelty of our work in various aspects. We have uploaded our meta responses for this rebuttal phase, summarizing the key improvements resulting from our discussions with you and other reviewers. Please review our meta responses, and we would be happy to engage further should you have additional questions.
> >
> > Sincerely,
> >
> > The authors

---

### Official Review · Reviewer_S8KX · 2025-10-30

**Soundness:** 2
**Presentation:** 2
**Contribution:** 2
**Rating:** 4
**Confidence:** 5

**Summary:**

This paper proposes NERVE, a noise- and variability-robust EEG foundation model that explicitly addresses three acquisition-related challenges: low signal-to-noise ratio, high inter- and intra-subject variability, and spatial dependencies among electrodes. NERVE introduces a noise-robust neural tokenizer trained via denoising temporal–spectral prediction, a variability-robust pre-training objective using KoLeo regularization, and an electrode-position-aware (EPA) transformer to capture spatial structure. Evaluated on multiple downstream BCI tasks, NERVE demonstrates competitive performance and improved robustness to noise and variability compared to existing EEG foundation models.

**Strengths:**

- NERVE explicitly tackles three real-world acquisition-related issues—low SNR, high inter/intra-subject variability, and spatial electrode dependencies—often overlooked by prior EEG foundation models.
- The combination of a noise-robust neural tokenizer, KoLeo-based variability regularization, and the electrode-position-aware (EPA) transformer provides a coherent framework for robust representation learning.
- Experiments across diverse BCI tasks and noise conditions demonstrate consistent performance gains and improved robustness over existing baselines.

**Weaknesses:**

- While NERVE introduces several modifications, its core pre-training framework closely resembles that of LaBraM—relying on masked EEG modeling with a vector-quantized neural tokenizer. The primary differences (e.g., Gaussian noise augmentation and the electrode-position-aware attention) are incremental rather than fundamentally novel, weakening the paper’s claim to significant architectural innovation.

- The motivation for noise robustness hinges on training the neural tokenizer with synthetic Gaussian noise. However, real-world EEG noise arises from diverse physiological (e.g., EMG, EOG) and environmental sources with complex, non-stationary characteristics. Relying solely on Gaussian noise may not adequately capture this complexity, raising concerns about the practical relevance and generalizability of the proposed noise-robustness strategy.

- The paper demonstrates that NERVE outperforms baselines even on clean data, yet it does not convincingly explain why the noise-augmented tokenizer leads to better representations in the absence of synthetic noise. This suggests that performance gains may stem from factors other than noise robustness (e.g., regularization effects), but the contribution of the noise-robust tokenizer to representation quality remains ambiguous.

- A critical experiment is absent: a direct comparison between NERVE trained with a standard (non-noise-robust) neural tokenizer and the proposed noise-robust variant. Without this control, it is difficult to isolate and validate the actual benefit of the denoising temporal–spectral prediction objective, undermining the evidence for its necessity.

**Questions:**

- **Why is Gaussian noise specifically chosen in the Noise-robust Neural Tokenizer Training?**
  The paper justifies the use of Gaussian noise by noting its statistical independence and resistance to rule-based filtering. However, it does not provide a thorough comparison with other noise types (e.g., EMG, EOG, or real-world artifacts) or explain why Gaussian noise is a suitable proxy for the complex, structured noise commonly present in real EEG recordings.

- **How does training with Gaussian noise improve robustness to other types of noise (e.g., environmental, EMG, or EOG)?**
  The results suggest that NERVE trained with Gaussian-augmented inputs shows improved robustness across diverse noise conditions. Yet, the mechanism behind this cross-noise generalization remains unclear. Since Gaussian noise differs significantly in spectral and physiological characteristics from structured artifacts like eye blinks or muscle activity, it is uncertain why robustness to such diverse noise sources would emerge from Gaussian-only augmentation.

---

> ### Author Response · Authors · 2025-11-21
> **Response to Reviewer S8KX (1)**
>
> > [W1] While NERVE introduces several modifications, its core pre-training framework closely resembles that of LaBraM—relying on masked EEG modeling with a vector-quantized neural tokenizer. The primary differences (e.g., Gaussian noise augmentation and the electrode-position-aware attention) are incremental rather than fundamentally novel, weakening the paper’s claim to significant architectural innovation.
>
> - Thank you for your valuable comment. NERVE is the first foundation model that addresses three key challenges arising from EEG signal acquisition; low signal-to-noise ratio, high variability between EEG samples, and spatial dependency from electrode configuration, which should be considered since EEG is collected through the physical sensors on the human body. To systematically takle these challenges, we propose a noise-robust neural tokenizer training, variability-robust NERVE pre-training methods, and electrode-position-aware (EPA) attention.
>
> - While NERVE also follows vector-quantized variational autoencoder (VQ-VAE) and masked EEG modeling, we enhance the pre-training architecture by proposing a new neural tokenizer pre-training task, denoising temporal-spectral prediction, variability-robust learning task, and an EPA attention (see Table 1 below).
>     - In particular, denoising temporal-spectral prediction for the neural tokenizer mitigates the failture of phase prediction in LaBraM and facilitates learning the full spectrum of the original sample from the noisy contexts. By applying both temporal reconstruction and frequency prediction losses to the reconstructed EEG signals without an additional head, we can simultaneously achieve enhanced denoising efficacy and improved computational efficiency.
>     - While existing foundation models do not consider the topological structure of electrode placements for encoding EEG samples, EPA attention incorporates electrode positions and cortical region groupings into calculation of the attention score (see Figures 1 and 3 in the manuscript). EPA attention can be regarded as the sequential generalization of criss-cross attention in CBraMod, which suggests the architectural contribution of NERVE (see Section F in the Appendix of the revised manuscript).
>
>
> (Table 1) EEG foundation model comparison
>
> || BIOT | LaBraM | CBraMod | NERVE |
> |---|:---:|:---:|:---:|:---:|
> |1. Diverse sample format| O | O | O | O |
> |2. Acquisition-related challenges | X | X | X | O |
> |2.1. Noise robustness| X | X | X | O |
> |2.2. Variability robustness | X | X | X | O |
> |2.3. Spatial dependency between electrodes | X | X | $\triangle$ | O |

---

> ### Author Response · Authors · 2025-11-21
> **Response to Reviewer S8KX (2)**
>
> > [w2] The motivation for noise robustness hinges on training the neural tokenizer with synthetic Gaussian noise. However, real-world EEG noise arises from diverse physiological (e.g., EMG, EOG) and environmental sources with complex, non-stationary characteristics. Relying solely on Gaussian noise may not adequately capture this complexity, raising concerns about the practical relevance and generalizability of the proposed noise-robustness strategy.
>
> - Thank you for your thorough feedback. We chose Guassian noise to achieve artifact-agnoistic robustness as a foundation model and enhancing the generalizability of NERVE to various noise scenarios. The objective of noise-robust training was not to become robust to specific noise artifacts. Instead, our goal was to learn a universal denoising mechanism for building the foundation model that is generalizable to diverse acqusition process and BCI tasks. Gaussian noise perturbs the signal shapes in the time domain, while distorting the proportions of all frequencies in the signal spectrum [1]. Physiological noise artifacts modify the specific frequency bands, for example, EOG noise generally corrupts the low frequency bands below 12 Hz and EMG noise is added to the high frequency bands over 20 Hz. Our denoising temporal-spectral prediction on broadband disturbance signals reconstructs the original signal across both the time domain and the full spectrum. This enables the model to learn EEG embeddings that are robust even against noise localized to specific ranges.
>
> - In BCI research, Gaussian noise is widely adopted as a representative proxy for EEG noise [1]. Existing studies have shown that models trained with Gaussian noise exhibit robustness to real-world physiological noises [2]. We have also demonstrated NERVE achieves superior robustness over existing foundation models on synthetic EOG, EMG, and environmental noises (see Figure 4 and 9 in the revised manuscript).
>
> - We deeply understand the concerns regarding the practical validity of noise robustness analysis. To better validate the robustness to complex and unstructured noises, we controlled the signal-to-noise ratio (SNR) of the input signal, which is defined as
> $SNR_{DB}=10log_{10}(\frac{{P_{signal}}}{P_{noise}})$, where the power $P=lim_{T\rightarrow \infty} \frac{1}{T} \int_{-\frac{T}{2}}^{\frac{T}{2}}s^2(t)dt$. SNR serves as the standard metric for quantifying signal quality and evaluating model robustness across varying noise levels [3, 4]; low SNR of EEG signals represents the most critical acquisition-related challenge we aim to address. We defined several noise levels using SNR values of 20, 10, 5 dB (ranging from clean to noisy signals) and corrupted the input signal with these artificial noises.
>
> - As shown in Table 2 below, NERVE demonstrates the superior robustness on these corrupted signals, achieving smallest accuracy drop and prediction entropy change compared to existing EEG foundation models. These analyses demonstrates the robustness of NERVE to complex and unstructured noises as well as physiological noises, emphasizing the practical relevance and generalizability of our work. To faciliate your understanding, we provided a case visualization of synthetic noise scenarios in Figure 8 of the revised manuscript. Additional analysis results can also be found in Figure 9 and Section G.1 in the Appendix.
>
> (Table 2) Noise robustness analysis on corrupted EEG samples from High-Gamma and HCI-Tagging Emotion
>
> |High_Gamma|  | $\triangle$ Acc |  |  | \|$\triangle$ Entropy\| | |HCI-Tagging Emotion|  | $\triangle$ AUROC |  |  | \|$\triangle$ Entropy\| | |
> |:---:|:---:|:---:|:---:|:---:|:---:|:---:|:---:|:---:|:---:|:---:|:---:|:---:|:---:|
> ||LaBraM|CBraMod|NERVE|LaBraM|CBraMod|NERVE||LaBraM|CBraMod|NERVE|LaBraM|CBraMod|NERVE|
> |SNR 5|-0.686|-0.381|**-0.208**|1.197|0.581|**0.245**|SNR 5|-0.058|-0.046|**-0.028**|0.066|0.075|**0.005**|
> |SNR 10|-0.444|-0.111|**-0.042**|0.608|0.133|**0.045**|SNR 10|-0.047|-0.015|**-0.014**|0.080|0.035|**0.003**|
> |SNR 20|-0.061|-0.006|**-0.001**|0.065|0.007|**0.001**|SNR 20|-0.039|0.001|**-0.002**|0.068|0.007|**0.001**|
>
>
> [1] Rommel, C., Moreau, T., Paillard, J., & Gramfort, A. CADDA: Class-wise Automatic Differentiable Data Augmentation for EEG Signals. In International Conference on Learning Representations 2022.
>
> [2] Gordienko, Y., Gordienko, N., Taran, V., Rojbi, A., Telenyk, S., & Stirenko, S. (2025). Effect of natural and synthetic noise data augmentation on physical action classification by brain–computer interface and deep learning. Frontiers in Neuroinformatics.
>
> [3] Braun, S., Neil, D., & Liu, S. C. (2017, August). A curriculum learning method for improved noise robustness in automatic speech recognition. In 2017 25th European Signal Processing Conference (EUSIPCO). IEEE.
>
> [4] Galeotti, L., & Scully, C. G. (2018). A method to extract realistic artifacts from electrocardiogram recordings for robust algorithm testing. Journal of electrocardiology, 51(6).

---

> ### Author Response · Authors · 2025-11-21
> **Response to Reviewer S8KX (3)**
>
> > [w3] The paper demonstrates that NERVE outperforms baselines even on clean data, yet it does not convincingly explain why the noise-augmented tokenizer leads to better representations in the absence of synthetic noise. This suggests that performance gains may stem from factors other than noise robustness (e.g., regularization effects), but the contribution of the noise-robust tokenizer to representation quality remains ambiguous.
>
> - Thank you for the valuable comment. Training with noise improves generalization performance by acting as training with generalized Tikhonov regularization [5], explaining why NERVE achieves better performance in downstream BCI tasks, even with clean data. Our noise-robust neural tokenizer training closely resembles a denoising autoencoder [6]. The goal of denoising autoencoder is not the task of denoising itself. Denoising serves as a training criterion for learning to extract useful representations that constitute better high-level semantic information [7]. By our denoising temporal-spectral prediction task, the neural tokenizer learns quantized codebook embeddings that contain high-level semantics of EEG signal, thereby providing a prediction target containing high-level information for NERVE pre-training.
>
> - The process of denoising can also be interpreted geometrically as mapping a corrupted sample back to an uncorrupted one under the manifold assumption [7]. In particular, when Gaussian noise is added, the denoising autoencoder training criterion is equivalent to score matching objective, which indicates that optimizing this criterion approximates the gradient of the log density [8]. Thus, our noise-robust training can be theoretically regarded as learning the useful representations that effectively approximate the data manifold.
>
> [5] Bishop, C. M. (1995). Training with noise is equivalent to Tikhonov regularization. Neural computation, 7(1), 108-116.
>
> [6] Vincent, P., Larochelle, H., Bengio, Y., & Manzagol, P. A. (2008, July). Extracting and composing robust features with denoising autoencoders. In Proceedings of the 25th international conference on Machine learning (pp. 1096-1103).
>
> [7] Vincent, P., Larochelle, H., Lajoie, I., Bengio, Y., Manzagol, P. A., & Bottou, L. (2010). Stacked denoising autoencoders: Learning useful representations in a deep network with a local denoising criterion. Journal of machine learning research, 11(12).
>
> [8] Vincent, P. (2011). A connection between score matching and denoising autoencoders. Neural computation, 23(7), 1661-1674.

---

> ### Author Response · Authors · 2025-11-21
> **Response to Reviewer S8KX (4)**
>
> > [W4] A critical experiment is absent: a direct comparison between NERVE trained with a standard (non-noise-robust) neural tokenizer and the proposed noise-robust variant. Without this control, it is difficult to isolate and validate the actual benefit of the denoising temporal–spectral prediction objective, undermining the evidence for its necessity.
>
> - Thank you for the valuable suggestion. Following your recommendation, we compared NERVE trained with the noise-robust neural tokenizer against NERVE trained with a standard neural tokenizer (without noise augmentation), which we term NERVE (standard). As shown in Table 3, NERVE utilizing the noise-robust neural tokenizer outperforms the model trained without noise augmentation in BCI tasks. Moreover, it exhibits superior robustness to noise, characterized by small accuracy degradation and predictive entropy changes (see Table 4). The ablation study underscores that noise-robust tokenizer training is essential for achieving generalizability and robustness. More experimental results are presented in Section I.1 in the Appendix of the revised manuscript.
>
> (Table 3) The ablation results on noise-robust training of neural-tokenizer
>
> |Methods|  |SEED-V| | |TUSL| |
> |:---:|:---:|:---:|:---:|:---:|:---:|:---:|
> ||Balanced Acc|Cohen's Kappa|Weighted F1|Balanced Acc|Cohen's Kappa|Weighted F1|
> |NERVE|**0.3788** $\pm$ 0.0031|**0.2305** $\pm$ 0.0034|**0.3908** $\pm$ 0.0028|**0.7000** $\pm$ 0.0282|**0.5327** $\pm$ 0.0411|**0.6507** $\pm$ 0.0329|
> |NERVE (standard)|0.3764 $\pm$ 0.0014|0.2291 $\pm$ 0.0016|0.3096 $\pm$ 0.0012|0.6050 $\pm$ 0.0452|0.3748 $\pm$ 0.0736|0.5742 $\pm$ 0.0320|
>
> (Table 4) Analysis on robustness to noises of NERVE trained with noise-robust and standard neural tokenizers
>
> |High_Gamma||$\triangle$ Acc &emsp;&emsp;&emsp;&emsp;&emsp;|| \|$\triangle$ Entropy\|&emsp;&emsp;&emsp; | DEAP||$\triangle$ Acc &emsp;&emsp;&emsp;&emsp;&emsp;||\|$\triangle$ Entropy\|&emsp;&emsp;&emsp;|
> |:---:|:---:|:---:|:---:|:---:|:---:|:---:|:---:|:---:|:---:|
> |Noise|NERVE|NERVE (standard)|NERVE|NERVE (standard)|Noise|NERVE|NERVE (standard)|NERVE|NERVE (standard)|
> |EMG|-0.014|**-0.011**|**0.014**|0.017|EMG|**0.000**|-0.001|**0.001**|0.002|
> |EOG|**0.000**|-0.017|**0.001**|0.015|EOG|**-0.001**|-0.002|**0.006**|0.008|
> |Environment|**0.000**|-0.011|**0.001**|0.017|Environment|**0.000**|**0.000**|**0.001**|**0.001**|
> |Gaussian|**-0.131**|-0.147|**0.276**|0.345|Gaussian|**-0.025**|-0.029|**0.035**|0.061|
> |SNR 5|-0.175|**-0.169**|**0.374**|0.394|SNR 5|**-0.019**|-0.026|**0.030**|0.060|
> |SNR 10|**-0.003**|-0.083|**0.040**|0.188|SNR 10|**-0.008**|-0.028|**0.020**|0.048|
> |SNR 20|**-0.001**|-0.005|**0.001**|0.0015|SNR 20|**-0.002**|-0.007|**0.005**|0.012|

---

> ### Author Response · Authors · 2025-11-21
> **Response to Reviewer S8KX (5)**
>
> > [Q1] Why is Gaussian noise specifically chosen in the Noise-robust Neural Tokenizer Training? The paper justifies the use of Gaussian noise by noting its statistical independence and resistance to rule-based filtering. However, it does not provide a thorough comparison with other noise types (e.g., EMG, EOG, or real-world artifacts) or explain why Gaussian noise is a suitable proxy for the complex, structured noise commonly present in real EEG recordings.
>
> - We sincerely appreciate your insightful feedback. Based on your feedback, we provide the figure comparing the patterns of various noise artifacts in the revised manuscript (see Figure 8 in the revised manuscript).
>
> - We employed Gaussian noise as the primary interference source to enhance the robustness of NERVE. Gaussian noise represents the most comprehensive form of noise widely addressed in EEG research [1, 2]. Unlike physiological artifacts such as EMG or EOG, which are localized to specific frequency bands, Gaussian noise perturbs the entire frequency spectrum. This broadband disturbance leads to a more complex distortion of temporal patterns, presenting a challenge for the model to recover the original signal. Furthermore, by training on this unstructured and random noise, we aim to prevent the model from becoming biased toward specific artifact shapes. This approach ensures that the model learns the intrinsic manifold of the EEG signals, thereby functioning as a 'foundation model' capable of generalizing across diverse and unseen noise conditions. Thus, we propose Gaussian noise as an ideal proxy for EEG artifacts in training a foundation model.
>
>
> > [Q2] How does training with Gaussian noise improve robustness to other types of noise (e.g., environmental, EMG, or EOG)? The results suggest that NERVE trained with Gaussian-augmented inputs shows improved robustness across diverse noise conditions. Yet, the mechanism behind this cross-noise generalization remains unclear. Since Gaussian noise differs significantly in spectral and physiological characteristics from structured artifacts like eye blinks or muscle activity, it is uncertain why robustness to such diverse noise sources would emerge from Gaussian-only augmentation
>
> - Thank you for your valuable comment. We chose Gaussian noise to achieve cross-noise generalization capacity as a foundation model. Gaussian noise alters the signal shapes in the time domain, while distorting the proportions of all frequencies in the signal spectrum [1]. Physiological noise artifacts modify the specific frequency bands, for example, EOG noise generally corrupts the low frequency bands below 12 Hz and EMG noise is added to the high frequency bands over 20 Hz. Our denoising temporal-spectral prediction predicts the original signal in time domain and the full spectrum, which allows to learn the EEG embeddings robust to noises injected to specific range.
>
> - As a foundation model, NERVE aims for broad robustness against diverse noise artifacts, rather than being tailored to specific ones. While artifact-specific augmentation may enhance stability against targeted artifacts, it often fails to achieve cross-noise generalization. In BCI research, Gaussian noise is widely adopted as a representative proxy for EEG noise. Existing studies have shown that models trained with Gaussian noise exhibit robustness to real-world physiological noises [2]. Theoretically, the generalizability of a Gaussian-augmented denoising autoencoder is supported by its connection to score matching, which approximates the gradient of the data manifold [8]. Thus, from both practical and theoretical perspectives, Gaussian augmentation ensures generalized robustness to various noise artifacts, including those unseen during pre-training.
>
> - We further validate cross-noise generalization by evaluating noise robustness on corrupted EEG signals with degraded SNR. We define several noise levels using SNR values of 20, 10, 5 dB (ranging from clean to noisy signals) and corrupt the input signal with these artificial noises. As shown in Table 2 below, NERVE exhibits the superior robustness on these corrupted signals, highlighting the stability of NERVE to complex noises.
>
> (Table 2) Noise robustness analysis on corrupted EEG samples from High-Gamma and HCI-Tagging Emotion (Repeated)
>
> |High_Gamma|  | $\triangle$ Acc |  |  | \|$\triangle$ Entropy\| | |HCI-Tagging Emotion|  | $\triangle$ AUROC |  |  | \|$\triangle$ Entropy\| | |
> |:---:|:---:|:---:|:---:|:---:|:---:|:---:|:---:|:---:|:---:|:---:|:---:|:---:|:---:|
> ||LaBraM|CBraMod|NERVE|LaBraM|CBraMod|NERVE||LaBraM|CBraMod|NERVE|LaBraM|CBraMod|NERVE|
> |SNR 5|-0.686|-0.381|**-0.208**|1.197|0.581|**0.245**|SNR 5|-0.058|-0.046|**-0.028**|0.066|0.075|**0.005**|
> |SNR 10|-0.444|-0.111|**-0.042**|0.608|0.133|**0.045**|SNR 10|-0.047|-0.015|**-0.014**|0.080|0.035|**0.003**|
> |SNR 20|-0.061|-0.006|**-0.001**|0.065|0.007|**0.001**|SNR 20|-0.039|0.001|**-0.002**|0.068|0.007|**0.001**|

---

> > ### Author Response · Authors · 2025-11-30
> >
> > Dear Reviewer S8KX,
> >
> > We appreciate your positive comments and insightful feedback, which led to constructive discussions and meaningful improvements in the revised manuscript.
> >
> > Your feedback has led us to **clarify the technical contribution of NERVE** by positioning our work in previous EEG foundation model research (W1). Also we were able to **demonstrate the effectiveness of the noise-robust neural tokenizer through additional noise-robustness analyses and ablation studies** (W2, W4, Q2). Lastly, we could **strengthen the logical rationale for using Gaussian noise as the augmentation for our noise-robust foundation model and its superior generalization performance** (W3, Q1).
> >
> > We have uploaded our meta responses for this rebuttal phase, summarizing the key improvements resulting from our discussions with you and other reviewers. Please review our meta responses, and we would be happy to engage further should you have additional questions.
> >
> > Once again, we sincerely thank you for helping us clarify our research motivation and strengthen the technical contribution of our work.
> >
> > Sincerely,
> >
> > The authors

---

### Author Response · Authors · 2025-11-28
**Response to All Reviewers**

## Dear Reviewers,

We thank the reviewers for their insightful feedback. We appreciate **the positive comments on the significance of NERVE in EEG foundation model research** (S8KX, c7uQ, 1XLF), **the novelty of its architecture** (ejCD), **the effectiveness of the proposed method** (S8KX, ejCD, c7uQ), **the quality of pre-training corpus** (ejCD), **its impact on future EEG research** (ejCD), **the comprehensive evaluation** (S8KX, ejCD, c7uQ), **the solid model analyses** (ejCD), and **the clear presentation** (c7uQ, 1XLF).

Based on the constructive feedback from the reviewers, we have made the following improvements: (1) clarified the novelty of our work regarding EEG foundation models; (2) clarified the validity of Gaussian augmentation for noise-robust neural tokenizer training; (3) enhanced the technical contribution of electrode-position-aware (EPA) attention; (4) strengthened the practical validity of the noise robustness analysis; (5) conducted the extended ablation studies and analyses to demonstrate the effectiveness of NERVE.

Despite the extensive computational resources and time required for model pre-training, we have dedicated every effort to sincerely address the reviewers' comments and enhanced the manuscript within the short timeframe provided for the rebuttal. We will now outline the specific revisions made in the revised manuscript:

----

1. In response to reviewer c7uQ, we refined the caption of Figure 1 to clearly compare EPA attention with spatial attentions of existing EEG foundation models. **(Figure 1 in page 2)**
2. In response to reviewer ejCD, we have clarified the description of acquisition-related challenges. **(line 80-83, 96-98 in page 2)**
3. In response to reviewer ejCD, we have revised the notation in the EPA attention mechenism to improve the clarity of the formulation. **(line 220-223 in page 5)**
4. In response to the concerns of multiple reviewers on noise robustness analysis, we have included two low SNR scenarios of SNR 5 dB and 10 dB to enhance the practical validity of the analysis results. **(line 416-418, 424-426 in page 8, Figure 4 in page 9, and line 1378-1388, 1446-1449 in page 26-27)**
5. In response to reviewer ejCD and 1XLF, we have added the new section "Connection Between EPA Attention and Existing Studies" to investigate the formal connection between the EPA attention and attention algorithms proposed in the previous EEG foundation model. **(line 1332-1372 in page 25-26)**
6. In response to the suggestion of reviewer c7uQ, we have provided the case visualization of synthetic noise generalization for robustness analysis. **(Figure 8 in page 26)**
7. In response to the suggestion of reviewer c7uQ, we have added the results on two additional datasets (TUSL and SEED-V) about noise robustness analysis. **(Figure 9 in page 27)**
8. In response to reviewer ejCD, we have explained the experimental constraints on the variability robustness analysis, detailing why we cannot provide analysis results on all downstream datasets. **(line 1465-1473 in page 28)**
9. In response to reviewer ejCD and 1XLF, we have added the new section "Interpretability Analysis" to demonstrate the effectiveness of the EPA attention. **(line 1507-1544 and Figure 12-13 in page 28-29)**
10. In response to multiple reviewers, we have included the ablation study on noise-robust neural tokenizer training to demonstrate the effectiveness of noise-robust neural tokenizer for downstream tasks and noise robustness. **(line 1549-1600, Figure 14, and Table 12 in page 29-30)**
11. In response to reviewer c7uQ, we have provided the representation analysis to investigate the model's behavior when the signal is corrupted with the noise. **(line 1601-1626 and Table 13 in page 30-31)**
12. In response to reviewer ejCD, we have added the additional ablation results on variability-robust pre-training to investigate the case that variability-robust pre-training is not effective. **(line 1649-1658 and Table 15 in page 31)**
13. In response to reviewer ejCD and 1XLF, we have added the ablation results on attention mechanisms for NERVE encoder to demonstrate the effectiveness of the EPA attention as a foundation model backbone. **(line 1708-1714 and Table 17 in page 32)**
14. In response to reviewer ejCD and 1XLF, we have added the topography visualization of different attention mechanisms to present the capacity of the EPA attention to capture the raw topography. **(line 1747-1760 and Figure 15 in page 33)**
15. In response to reviewer ejCD, we have included the discussion on explicit physical constraint on the position router as a limitation of our work. **(line 1819-1824 in page 34)**
16. Throughout the paper, we have updated the experimental results **from three random repetitions to five random repetitions** to enhance the statistical reliability of the findings.

---

> ### Author Response · Authors · 2025-11-28
> **Response to All Reviewers**
>
> We hope that our work contributes valuable insights and impacts to EEG research communities, addressing significant challenges for the practitioners and researchers.
>
> We will use the following space to summarize the key points raised by the reviewers and provide our detailed meta responses. These meta responses will address the five aspects mentioned above, aiming to offer a clearer understanding of the overall revisions made to our work.

---

> ### Author Response · Authors · 2025-11-28
> **Meta Response 1**
>
> ## Meta Response 1. Clarifying the novelty of our work for EEG foundation models
>
> NERVE is the first foundation model that **addresses three key challenges arising from EEG signal acquisition; low signal-to-noise ratio, high variability between EEG samples, and spatial dependency from electrode configuration, which should be considered since EEG is collected through the physical sensors on the human body.** While existing EEG foundation models primarily variability from diverse sample format, they often overlook the inherent characteristics of EEG signals arising from their unique collection environment. To systematically tackle these challenges, we propose a noise-robust neural tokenizer training, variability-robust NERVE pre-training methods, and electrode-position-aware (EPA) attention (see Table 1 below)
>
> (Table 1) EEG foundation model comparison
>
> || BIOT | LaBraM | CBraMod | NERVE |
> |---|:---:|:---:|:---:|:---:|
> |1. Diverse sample format| O | O | O | O |
> |2. Acquisition-related challenges | X | X | X | O |
> |2.1. Noise robustness| X | X | X | O |
> |2.2. Variability robustness | X | X | X | O |
> |2.3. Spatial dependency between electrodes | X | X | $\triangle$ | O |
>
> With the canonical vector-quantized variational autoencoder (VQ-VAE) and masked EEG modeling, we enhance the pre-training architecture by proposing a new neural tokenizer pre-training task, denoising temporal-spectral prediction, variability-robust learning task, and an EPA attention.
>
> - In particular, denoising temporal-spectral prediction for the neural tokenizer mitigates the failture of phase prediction in LaBraM and facilitates learning the full spectrum of the original sample from the noisy contexts. By applying both temporal reconstruction and frequency prediction losses to the reconstructed EEG signals without an additional head, we can simultaneously achieve enhanced denoising efficacy and improved computational efficiency.
>
> - We regard the variability between EEG samples as properties within the representation space. Variability-robust learning enhances the uniformity of the EEG representation space for inter-subject separability, coupled with masked EEG modeling to implicitly enhance alignment for intra-subject consistency.
>
> - While existing foundation models do not consider the topological structure of electrode placements for encoding EEG samples, EPA attention incorporates electrode positions and cortical region groupings into calculation of the attention score (see Figures 1 and 3 in the manuscript). EPA attention can be regarded as the sequential generalization of criss-cross attention in CBraMod, which suggests the architectural contribution of NERVE (see Meta Response 3 and Section F in the revised manuscript).

---

> > ### Author Response · Authors · 2025-11-28
> > **Meta Response 2**
> >
> > ## Meta Response 2. Clarifying the validity of Gaussian augmentation for noise-robust neural tokenizer training
> >
> > Gaussian augmentation was selected for training noise-robust neural tokenizer to **achieve artifact-agnostic robustness as a foundation model and enhancing the generalizability of NERVE to various noise scenarios.** The objective of noise-robust training was not to become robust to specific noise artifacts. Instead, our goal was to learn a universal denoising mechanism for building the foundation model that is generalizable to diverse acqusition process and BCI tasks. To achieve this goal, Gaussian noise has several advantages:
> >
> > - Gaussian noise perturbs the signal shapes in the time domain, while distorting the proportions of all frequencies in the signal spectrum. Since the representative noise artifacts during EEG signal acquisition modify the specific frequency bands, our noise-robust raining on broadband disturbance signals reconstructs the original signal across both the time domain and the full spectrum. This enables the model to learn EEG embeddings that are robust even against noise localized to specific ranges, thereby functioning as a 'foundation model' capable of generalizing across diverse and unseen noise conditions.
> >
> > - While training with a general noise improves generalization performance, denoising with Gaussian noise provides the theoretical basis for learning useful representations. Denoising serves as a training criterion for learning to extract useful representations that constitute high-level semantic information [1]. By our denoising temporal-spectral prediction task, the neural tokenizer learns quantized codebook embeddings that contain high-level semantics of EEG signal, thereby providing a prediction target containing high-level information for NERVE pre-training. In particular, when Gaussian noise is added in the denoising framework, the denoising training criterion is equivalent to score matching objective, which indicates that optimizing this criterion approximates the gradient of the log density [2]. Thus, our noise-robust training with Gaussian noise can be theoretically regarded as learning the useful representations that effectively approximate the data manifold.
> >
> > In BCI research, Gaussian noise is widely adopted as a representative proxy for EEG noise [3]. Existing studies have shown that models trained with Gaussian noise exhibit robustness to real-world physiological noises [4]. We have also demonstrated NERVE achieves superior robustness over existing foundation models on synthetic EOG, EMG, and environmental noises (see Section 4 and G.1 in the revised manuscript).
> >
> > To experimentally validate the effectiveness of noise-robust training, we compared NERVE with NERVE trained with a standard neural tokenizer (without Gaussian noise augmentation), which we term NERVE (standard). As shown in Table 2, NERVE utilizing the noise-robust neural tokenizer outperforms the model trained without noise augmentation in BCI tasks. Moreover, it exhibits superior robustness to noise, characterized by small accuracy degradation and predictive entropy changes (see Table 3). The ablation study underscores that noise-robust tokenizer training is essential for achieving generalizability and robustness. The validity of Gaussian noise augmentation and more experimental results are presented in Section I.1 in the revised manuscript.
> >
> > (Table 2) The ablation results on noise-robust training of neural-tokenizer
> >
> > |Methods|  |SEED-V| | |TUSL| |
> > |:---:|:---:|:---:|:---:|:---:|:---:|:---:|
> > |NERVE|**0.3788** $\pm$ 0.0031|**0.2305** $\pm$ 0.0034|**0.3908** $\pm$ 0.0028|**0.7000** $\pm$ 0.0282|**0.5327** $\pm$ 0.0411|**0.6507** $\pm$ 0.0329|
> > |NERVE (standard)|0.3764 $\pm$ 0.0014|0.2291 $\pm$ 0.0016|0.3096 $\pm$ 0.0012|0.6050 $\pm$ 0.0452|0.3748 $\pm$ 0.0736|0.5742 $\pm$ 0.0320|
> >
> > (Table 3) Analysis on robustness to noises of NERVE trained with noise-robust and standard neural tokenizers
> >
> > |High_Gamma||$\triangle$ Acc &emsp;&emsp;&emsp;&emsp;&emsp;|| \|$\triangle$ Entropy\|&emsp;&emsp;&emsp; | DEAP||$\triangle$ Acc &emsp;&emsp;&emsp;&emsp;&emsp;||\|$\triangle$ Entropy\|&emsp;&emsp;&emsp;|
> > |:---:|:---:|:---:|:---:|:---:|:---:|:---:|:---:|:---:|:---:|
> > |Noise|NERVE|NERVE (standard)|NERVE|NERVE (standard)|Noise|NERVE|NERVE (standard)|NERVE|NERVE (standard)|
> > |EMG|-0.014|**-0.011**|**0.014**|0.017|EMG|**0.000**|-0.001|**0.001**|0.002|
> > |EOG|**0.000**|-0.017|**0.001**|0.015|EOG|**-0.001**|-0.002|**0.006**|0.008|
> > |Environment|**0.000**|-0.011|**0.001**|0.017|Environment|**0.000**|**0.000**|**0.001**|**0.001**|
> > |Gaussian|**-0.131**|-0.147|**0.276**|0.345|Gaussian|**-0.025**|-0.029|**0.035**|0.061|
> > |SNR 5|-0.175|**-0.169**|**0.374**|0.394|SNR 5|**-0.019**|-0.026|**0.030**|0.060|
> > |SNR 10|**-0.003**|-0.083|**0.040**|0.188|SNR 10|**-0.008**|-0.028|**0.020**|0.048|
> > |SNR 20|**-0.001**|-0.005|**0.001**|0.0015|SNR 20|**-0.002**|-0.007|**0.005**|0.012|

---

> ### Author Response · Authors · 2025-11-28
> **Meta Response 3 (1)**
>
> ## Meta Response 3. Enhancing the technical contribution of electrode-position-aware (EPA) attention
>
> The EPA attention in the EPA transformer defines a position router $P\in \mathbb{R}^{R\times N\times d}$ as a learnable parameter, representing groups of electrodes organized by cortical regions. Then EPA attention leverages the position router to calculate attention score, thereby learning electrode-region-electrode dependencies and signal-driven soft cortical groupings. We enhance the technical contribution of EPA attention by (1) deriving the connection between the formulation of EPA attention and attention algorithms from exising EEG foundation model and (2) conducting solid ablation studies and analyses.
>
>
> ### (1) The connection between EPA attention and criss-cross attention in CBraMod
>
> We denote the EPA attention score $S_j=(Q_H M_j K_H^T)/\sqrt{d}$ and $M_j=Q_P^T K_P=(PW_Q)^T(PW_K)\in\mathbb{R}^{d\times d}$. From this formulation, we demonstrate that EPA transformer constitutes as a sequential formulation of 'generalized' criss-cross transformer in CBraMod [5]. Criss-cross attention is defined as parallel-head formulation of standard temporal and spatial attentions. If $M_j\equiv I$, the EPA attention score reduces to the standard spatial attention. Thus, EPA transformer reduces to the sequential conversion of criss-cross transformer. Thus, criss-cross attention can be regarded as a special case of EPA attention.
>
> Furthermore, given that $PW_Q \in \mathbb{R}^{d\times R}$, the rank of $M_j$ is inherently bounded by $R$. Since $rank(M_j)\leq R$, so EPA attention constrains the channel-wise score matrix $S_j=(Q_H M_j K_H^T)/\sqrt{d}$ to lie in a low-dimensional, router-defined subspace. This low-rank and topology-aligned regularization encourages robust electrode–region–electrode dependencies while suppressing spurious correlations among channels.
>
> When deriving the gradient of the loss function $L$ with respect to the attention score $S_j$ (see the equation below), $M_j^T$ acts as a multiplier to the gradient of the standard channel attention. Given that $rank(M_j)\leq R$, this operation constrains the gradient flow, imposing an implicit regularization effect on the learned parameters.
>
> $$\frac{\partial L}{\partial S_j} = \frac{\partial L}{\partial Q_H} \left(\frac{\partial S_j}{\partial Q_H}\right)^{-1}=\frac{\partial L}{\partial Q_H}(M_jK_H^T)^{-1}=\frac{\partial L}{\partial Q_H}(K_H)^{-1} (M_j^T)^{-1}$$

---

> ### Author Response · Authors · 2025-11-28
> **Meta Response 3 (2)**
>
> ### (2) Ablation studies and analyses for the EPA attention
>
> We have conducted several ablations and analyses to substantiate the link between EPA attention and the superior BCI task performance:
>
> - First, we conducted a visualization analysis to compare the topography of the raw sample with class activation topography of NERVE on the High-Gamma dataset. The visualization results and the setup of visualization analysis are provided in Figure 13 and Section H.2 of the Appendix in the revised manuscript. We observed that electrodes associated with the left hand exhibit patterns relatively symmetric to those of the right hand, while both cortical regions are activated for the 'both feet' class. In contrast, minimal activation is observed during the resting state. Although NERVE's class activation topography is not identical to the raw EEG signals, its representations capture broadly similar spatial patterns, demonstrating distinct differences across classes.
>
> - Second, we compared the topography of raw samples from the pre-training datasets with the channel embedding topographies learned by different attention architectures. Specifically, we analyzed NERVE (using EPA attention) against variants equipped with temporal attention (attention across the time dimension), temporal-channel attention (sequential blocks of attention in time and channel dimensions), and criss-cross attention (the mechanism proposed by CBraMod). The visualization results are presented in Figure 15 of the revised manuscript. We observed that the channel embedding topography of NERVE with EPA attention resembles the raw signal topography more closely than those of other architectures. This indicates that EPA attention effectively captures the inherent topological structure of the electrodes.
>
> - Third, we visualized the inter-channel cosine similarity heatmap of NERVE embeddings using the BCIC2020-3 dataset from our pre-training corpus. As shown in Figure 12 of the revised manuscript, we observed high similarity among specific groups of electrodes, forming a distinct block-wise similarity structure. This observation underscores the necessity of region-wise channel modeling for EEG signals and supports the validity of EPA attention.
>
> - Lastly, we conducted an ablation study to evaluate various attention mechanisms within the NERVE backbone. We replaced EPA attention with temporal, temporal-channel, and criss-cross attention modules. As shown in Table 1 below, NERVE with EPA attention outperforms other attention-based architectures on the TUEV and DEAP datasets, highlighting the effectiveness of EPA attention for EEG modeling. Additional experimental results are provided in Table 17 and Section I.4 of the revised manuscript.
>
> (Table 4) The ablation results on attention mechanism for NERVE backbone.
> |||TUEV|||DEAP||
> |:---:|:---:|:---:|:---:|:---:|:---:|:---:|
> ||Acc|Kappa|F1|Acc|AUPRC|AUROC|
> |Temporal|0.5386 $\pm$ 0.0084|0.4570 $\pm$ 0.0110 | 0.7112 $\pm$ 0.0075 |0.4939 $\pm$ 0.0188|0.6223 $\pm$ 0.0088|0.4971 $\pm$ 0.0214|
> |Temporal-channel|0.5503 $\pm$ 0.0169|0.4753 $\pm$ 0.0043|0.7235 $\pm$ 0.0049|0.5086 $\pm$ 0.0075|0.6314 $\pm$ 0.0216|0.5044 $\pm$ 0.0277|
> |Criss-cross|0.5270 $\pm$ 0.0346|0.4615 $\pm$ 0.0587|0.7227 $\pm$ 0.0315|0.4967 $\pm$ 0.0081|0.6267 $\pm$ 0.0145|0.4982 $\pm$ 0.0217|
> |EPA|**0.5595** $\pm$ 0.0106|**0.4810** $\pm$ 0.0146|**0.7249** $\pm$ 0.0095|**0.5167** $\pm$ 0.0131|**0.6553** $\pm$ 0.0156|**0.5406** $\pm$ 0.0267|

---

> > ### Author Response · Authors · 2025-11-28
> > **Meta Response 4**
> >
> > ## Meta Response 4. Strengthening the practical validity of the noise robustness analysis
> >
> > Some reviewers raised concerns regarding the practical validity of NERVE's superior performance specifically under the Gaussian noise condition. There were also concerns about the marginal difference observed between NERVE and the baseline models when evaluated on the High-Gamma dataset under EMG, EOG, and environmental noise conditions.
> >
> > While representative noise artifacts are often filtered through rule-based filtering techniques, residual noise inevitably remains in the processed signals [6, 7]. Because of that, the robustness to Gaussian noise is especially significant. Gaussian noise is statistically independent and corrupts all frequency bands, making its complete removal extremely difficult [8]. Baselines exhibited significant degradation in performance and prediction stability under the Gaussian condition. This indicates that residual Gaussian noise significantly hinders the reliable prediction of an EEG foundation model. Gaussian noise is the most commonly observed, unstructured pattern in many BCI tasks. An EEG foundation model must guarantee superior robustness to this fundamental noise type to ensure its practical utility. Therefore, our experimental results demonstrating superior robustness to the Gaussian condition critically highlight the practical contribution of NERVE.
> >
> > We acknowledge the point that the performance differences between NERVE and the baseline models on EMG, EOG, and environmental noise conditions might appear marginal. The High-Gamma dataset is a relatively easy dataset where most baselines already demonstrate high performance. Consequently, the designed noise simulation may not have been strong enough to induce a significant degradation in performance and stability across all noise types. However, as demonstrated in Figure 9 in the revised manuscript, the gap between NERVE and other baselines is not marginal and represents a meaningful improvement.
> >
> > To better validate the robustness to complex and unstructured noises, we generated two additional noise scenarios by controlling signal-to-noise ratio (SNR) of the input signal, which is defined as $SNR_{DB}=10log_{10}(\frac{{P_{signal}}}{P_{noise}})$, where the power $P=lim_{T\rightarrow \infty} \frac{1}{T} \int_{-\frac{T}{2}}^{\frac{T}{2}}s^2(t)dt$. SNR serves as the standard metric for quantifying signal quality and evaluating model robustness across varying noise levels [4, 5]; low SNR of EEG signals represents the most critical acquisition-related challenge we aim to address. We defined several noise levels using SNR values of 20, 10, 5 dB (ranging from clean to noisy signals) and corrupted the input signal with these artificial noises. As shown in Table 5 below, NERVE demonstrates the superior robustness on these corrupted signals, achieving smallest accuracy drop and prediction entropy change compared to existing EEG foundation models. These analyses demonstrates the robustness of NERVE to complex and unstructured noises as well as physiological noises, emphasizing the practical relevance and generalizability of our work. More analysis results are provided in Section G.1 in the revised manuscript.
> >
> >
> >
> > (Table 5) Noise robustness analysis on corrupted EEG samples from High-Gamma and HCI-Tagging Emotion
> >
> > |High_Gamma|  | $\triangle$ Acc |  |  | \|$\triangle$ Entropy\| | |HCI-Tagging Emotion|  | $\triangle$ AUROC |  |  | \|$\triangle$ Entropy\| | |
> > |:---:|:---:|:---:|:---:|:---:|:---:|:---:|:---:|:---:|:---:|:---:|:---:|:---:|:---:|
> > ||LaBraM|CBraMod|NERVE|LaBraM|CBraMod|NERVE||LaBraM|CBraMod|NERVE|LaBraM|CBraMod|NERVE|
> > |SNR 5|-0.686|-0.381|**-0.208**|1.197|0.581|**0.245**|SNR 5|-0.058|-0.046|**-0.028**|0.066|0.075|**0.005**|
> > |SNR 10|-0.444|-0.111|**-0.042**|0.608|0.133|**0.045**|SNR 10|-0.047|-0.015|**-0.014**|0.080|0.035|**0.003**|
> > |SNR 20|-0.061|-0.006|**-0.001**|0.065|0.007|**0.001**|SNR 20|-0.039|0.001|**-0.002**|0.068|0.007|**0.001**|

---

> > > ### Author Response · Authors · 2025-11-28
> > > **Meta Response 5**
> > >
> > > ## Meta Response 5. Conducting the extended ablation studies and analyses to demonstrate the effectiveness of NERVE.
> > >
> > > Apart from the ablation studies and analyses explained in the earlier meta responses, we have conducted further experiments to robustly demonstrate the effectiveness of the NERVE architecture.
> > >
> > > - We conducted the additional ablation study on variability-robust pre-training (pre-training without KoLeo loss) and analysed the cases when enhancing robustness to variability is not effective. As shown in Table 2 below, we compared NERVE with and without KoLeo loss on High-Gamma and SEED-VIG. On High-Gamma, the effect of variability-robust pre-training was marginal. This is attributed to the small number of test subjects on High-Gamma, which is two. For SEED-VIG, which is a regression dataset, NERVE trained without KoLeo loss outperforms the standard NERVE. This suggests that enhancing the uniformity of EEG samples in the embedding space does not necessarily enhance downstream performance for regression tasks.
> > >
> > >
> > > (Table 6) The ablation results on variability-robust pre-training (KoLeo loss)
> > >
> > > |||High-Gamma|||SEED-VIG||
> > > |:---:|:---:|:---:|:---:|:---:|:---:|:---:|
> > > ||Acc|Kappa|F1|Corr|R2 score|RMSE|
> > > |NERVE|**0.9906** $\pm$ 0.0011|**0.9875** $\pm$ 0.0015|**0.9906** $\pm$ 0.0011|0.4158 $\pm$ 0.0177|0.1591 $\pm$ 0.0142|0.2865 $\pm$ 0.0024|
> > > |NERVE (w/o KoLeo)|0.9904 $\pm$ 0.0013|0.9871 $\pm$ 0.0017|0.9904 $\pm$ 0.0013|**0.4427** $\pm$ 0.0198|**0.1694** $\pm$ 0.0251|**0.2847** $\pm$ 0.0044|
> > >
> > >
> > >
> > > - To better illustrate the behavior of NERVE on various noise scenarios, we conducted a representation analysis. We computed the Centered Kernel Alignment (CKA) similarity between embeddings of raw EEG signals and signals corrupted with synthetic noises for NERVE and NERVE trained with standard neural tokenizer (not noise-robust variant trained without noise augmentation), which we term as NERVE (standard). Table 7 below shows the representation analysis results on High-Gamma and TUSL. NERVE demonstrated higher CKA similarity between raw and corrupted EEG signals on most noise conditions. This result indicates that NERVE produces stable representations in the presence of noise artifacts, which accounts for the superior robustness reported in the noise robustness analysis. Through noise augmentation and denoising temporal-spectral prediction, the noise-robust neural tokenizer learns EEG codebooks that are resilient to noise, thereby providing a prediction target containing high-level semantic information for NERVE pre-training.
> > >
> > > (Table 7) Analysis of CKA similarity between embeddings of raw and noise-corrupted EEG signals
> > >
> > > |Noises||High-Gamma&emsp;&emsp;||TUSL&emsp;&emsp;&emsp;&emsp;&emsp;&emsp;&emsp;&emsp;&emsp;&emsp;|
> > > |:---:|:---:|:---:|:---:|:---:|
> > > ||NERVE|NERVE (standard)|NERVE| NERVE (standard)|
> > > |EOG|**0.9986** $\pm$ 0.0002|0.9941 $\pm$ 0.0004|**0.9980** $\pm$ 0.0008|0.9956 $\pm$ 0.0012|
> > > |EMG|**0.9996** $\pm$ 0.0001|0.9989 $\pm$ 0.0001|**0.9987** $\pm$ 0.0002|0.9967 $\pm$ 0.0006|
> > > |Environment|**0.9999** $\pm$ 0.0000|**0.9999** $\pm$ 0.0001|**0.9998** $\pm$ 0.0000|0.9993 $\pm$ 0.0002|
> > > |Gaussian|**0.7052** $\pm$ 0.0051|0.1395 $\pm$ 0.0019|0.7778 $\pm$ 0.0125|**0.8740** $\pm$ 0.0145|
> > > |SNR: 5dB|**0.6536** $\pm$ 0.0162|0.1323 $\pm$ 0.0048|**0.8644** $\pm$ 0.0089|0.8362 $\pm$ 0.0227|
> > > |SNR: 10dB|**0.9069** $\pm$ 0.0038|0.2416 $\pm$ 0.0052|**0.9558** $\pm$ 0.0061|0.9385 $\pm$ 0.0123|

---

> > > > ### Author Response · Authors · 2025-11-28
> > > > **Meta Response (References)**
> > > >
> > > > [1] Vincent, P., Larochelle, H., Lajoie, I., Bengio, Y., Manzagol, P. A., & Bottou, L. (2010). Stacked denoising autoencoders: Learning useful representations in a deep network with a local denoising criterion. Journal of machine learning research, 11(12).
> > > >
> > > > [2] Vincent, P. (2011). A connection between score matching and denoising autoencoders. Neural computation, 23(7), 1661-1674.
> > > >
> > > > [3] Rommel, C., Moreau, T., Paillard, J., & Gramfort, A. CADDA: Class-wise Automatic Differentiable Data Augmentation for EEG Signals. In International Conference on Learning Representations 2022.
> > > >
> > > > [4] Gordienko, Y., Gordienko, N., Taran, V., Rojbi, A., Telenyk, S., & Stirenko, S. (2025). Effect of natural and synthetic noise data augmentation on physical action classification by brain–computer interface and deep learning. Frontiers in Neuroinformatics, 19, 1521805.
> > > >
> > > > [5] Wang, J. et al., (2025). CBraMod: A Criss-Cross Brain Foundation Model for EEG Decoding. In The Thirteenth International Conference on Learning Representations.
> > > >
> > > > [6] Darren Tanner, Kara Morgan-Short, and Steven J Luck. (2015). How inappropriate high-pass filters can produce artifactual effects and incorrect conclusions in erp studies of language and cognition. Psychophysiology.
> > > >
> > > > [7] Whitham, E. M. et al. (2007). Scalp electrical recording during paralysis: quantitative evidence that EEG frequencies above 20 Hz are contaminated by EMG. Clinical neurophysiology : official journal of the International Federation of Clinical Neurophysiology.
> > > >
> > > > [8] Vaseghi, S. V. (2008). Advanced digital signal processing and noise reduction. John Wiley & Sons.

---

### Meta-Review · Area_Chair_E2BQ · 2026-01-08

**Summary:**

The paper proposes a EEG foundation model addressing three challenges- noise, variability, and electrode topology. Validation is done on 5 datasets.

**Strength**

(1) The paper explicitly tackles three challenges with clear motivation. Noise and variability are quantitatively analyzed.

(2) The method is thoughtfully designed.

(3) Experiments show consistent performance gains.

**Weakness**

(1) The core pre-training framework is similar to the existing LaBraM, and novelty is incremental (S8KX).

(2) Gaussian noise is added for training to achieve noise robustness, which seems to be far from real situations where noise in EEG has more complex characteristics (S8KX, ejCD, c7uQ).

(3) The effectiveness of the noise-robust tokenizer is not fully validated (S8KX)

(4) The effectiveness of EPA attention is not well justified (ejCD).

(5) The combination of the position router together with weight matrices seems to perform dimensionality reduction instead of modeling cortical regions, similarly to the existing criss-cross attention in CBraMod (ejCD).

**Reviewer Concerns:**

(1) The rebuttal mentions that the paper includes new components such as the tokenizer pre-training task, denoising temporal-spectral prediction, variability-robust learning task, and EPA attention. However, it may be questionable whether the new components are innovative enough.

(2) The rebuttal explains that Gaussian noise is representative and sufficient to make the tokenizer robust to various noise types and levels. Nevertheless, AC thinks that incorporating the characteristics of actual noise in EEG would have been more appealing.

(3) The rebuttal explains that training with noise acts as Tikhonov regularization and as score matching. To AC, this is somewhat different from the original motivation, which would need more clarification.

(4) The authors include additional results, which seems convincing.

(5) The rebuttal resolves the concern only partly. The role of the position router is still not 100% clear.

**Reviewer Scores:**

AC thinks that no reviewer would have changed their scores.

---

### Decision · Program_Chairs · 2026-01-26

Reject